# CAMS-TEMPO: global and European emission temporal profile maps for atmospheric chemistry modelling

Marc Guevara[1], Oriol Jorba[1], Carles Tena[1], Hugo Denier van der Gon[2], Jeroen Kuenen[2], Nellie Elguindi[3], Sabine Darras[3], Claire Granier[3,4], Carlos Pérez García-Pando[1,5]

[1]Barcelona Supercomputing Center, Barcelona, Spain
[2]TNO, Department of Climate, Air and Sustainability, Utrecht, the Netherlands
[3]Laboratoire d'Aérologie, CNRS-Université de Toulouse, Toulouse, France
[4]NOAA/Chemical Sciences Laboratory and CIRES/University of Colorado, Boulder, USA
[5]ICREA, Catalan Institution for Research and Advanced Studies, 08010 Barcelona, Spain

*Correspondence to*: Marc Guevara (marc.guevara@bsc.es)

**Abstract.** We present the Copernicus Atmosphere Monitoring Service TEMPOral profiles (CAMS-TEMPO), a dataset of global and European emission temporal profiles that provides gridded monthly, daily, weekly and hourly weight factors for atmospheric chemistry modelling. CAMS-TEMPO includes temporal profiles for the priority air pollutants ($NO_x$, $SO_x$, NMVOC, $NH_3$, CO, $PM_{10}$, $PM_{2.5}$) and the greenhouse gases ($CO_2$ and $CH_4$) for each of the following anthropogenic source
categories: energy industry (power plants), residential combustion, manufacturing industry, transport (road traffic and air traffic in airports) and agricultural activities (fertilizer use and livestock). The profiles are computed on a global 0.1x0.1 deg and regional European 0.1x0.05 deg grid following the domain and sector classification descriptions of the global and regional emission inventories developed under the CAMS program. The profiles account for the variability of the main emission drivers of each sector. Statistical information linked to emission variability (e.g. electricity production, traffic counts) at national and
local levels were collected and combined with existing meteorological-dependent parametrizations to account for the influences of sociodemographic factors and climatological conditions. Depending on the sector and the temporal resolution (i.e. monthly, weekly, daily, hourly) the resulting profiles are pollutant-dependent, yearly-dependent (i.e. time series from 2010 to 2017) and/or spatially-dependent (i.e. the temporal weights vary per country or region). We provide a complete description of the data and methods used to build the CAMS-TEMPO profiles and whenever possible, we evaluate the representativeness
of the proxies used to compute the temporal weights against existing observational data. We find important discrepancies when comparing the obtained temporal weights with other currently used datasets. The CAMS-TEMPO data product including the global (CAMS-GLOB-TEMPOv2.1, https://doi.org/10.24380/ks45-9147, Guevara et al., 2020a) and regional European (CAMS-REG-TEMPOv2.1, https://doi.org/10.24380/1cx4-zy68, Guevara et al., 2020b) temporal profiles are distributed from the Emissions of atmospheric Compounds and Compilation of Ancillary Data (ECCAD) system (https://eccad.aeris-data.fr/).

## 1    Introduction

Spatially and temporally resolved atmospheric emission inventories are key to investigate and predict the transport and chemical transformation of pollutants, as well as to develop effective mitigation strategies (e.g. Pouliot et al., 2015; Galmarini et al., 2017). During the last decade, global and regional inventories have substantially increased spatial resolution from ~50 x 50 km (e.g. MACCity; Granier et al., 2011; EMEP-50 km, Mareckova et al., 2013) to ~10 x 10 km or less (e.g. EMEP-0.1deg, Mareckova et al., 2017; TNO-MACC; Kuenen et al., 2014). Several datasets even provide emission maps for selected pollutants or study regions with resolutions as fine as 1 by 1km (e.g. ODIAC2016, Oda et al., 2018; Hestia-LA; Gurney et al., 2019; Super et al., 2020). This improvement is largely due to the emergence of new detailed, satellite-based and open-access spatial proxies such as the population maps at 1 x 1 km proposed by the Global Human Settlement Layer (GHSL) project (Florczyk et al., 2019), the global land cover maps at 300 x 300 m provided by the European Spatial Agency Climate Change Initiative (ESA CCI, https://www.esa-landcover-cci.org/) or the georeferenced road traffic network distributed by Open Street Maps (OSM, www.openstreetmap.org). While a clear evolution is observed in terms of spatial resolution, the improvement of the temporal representation in current state-of-the-art emission datasets has not been addressed much (Reis et al., 2011).

Using global and regional emission inventories in atmospheric chemistry models requires the original aggregated annual emissions to be broken down into fine temporal resolutions (ideally hourly) using emission temporal profiles (e.g. Borge et al., 2008; Bieser et al., 2011; Mues et al., 2014). In practice, temporal profiles are normalized weight factors for each hour of the day, day of the week and month of the year. At the global scale, the most commonly used emission temporal profiles are the monthly factors provided by the Air Pollutants and Greenhouse Gases Emission Database for Global Atmospheric Research inventory (EDGARv4.3.2; Janssens-Maenhout et al., 2019) and the Evaluating the Climate and Air Quality Impacts of Short-Lived Pollutants inventory (ECLIPSEv5.a; Klimont et al., 2017). Also at the global level, the Temporal Improvements for Modeling Emissions by Scaling (TIMES) dataset was produced to represent the weekly and hourly variability for global $CO_2$ emission inventories (Nassar et al., 2013). More recently, Crippa et al. (2020) developed a new set of high-resolution temporal profiles for the EDGAR inventory, which allows producing monthly and hourly emission time series and grid maps.

At European level, the temporal factors provided by the University of Stuttgart (IER) as part of the Generation of European Emission Data for Episodes (GENEMIS) project are still considered as the main reference (Ebel et al., 1997; Friedrich and Reis, 2004). The original GENEMIS profiles were later used as a basis to derive two independent datasets: (i) the EMEP temporal profiles, which provide monthly, weekly and hourly weight factors that vary per emission sector, country and pollutant (Simpson et al., 2012), and (ii) the Netherlands Organisation for Applied Scientific Research (TNO) temporal profiles, which provide monthly, weekly and hourly weight factors that vary per emission sector (Denier van der Gon et al., 2011). These two sets of profiles have become over time the reference datasets under the framework of several European air quality modelling activities, including the earlier Monitoring Atmospheric Composition and Climate (MACC) project and the

current Copernicus Atmosphere Monitoring Service (CAMS), among others. Other widely used regional temporal profile datasets include the North American profiles provided by the Environmental Protection Agency (EPA) Clearinghouse for Inventories and Emissions Factors (CHIEF) (US EPA, 2019a) and the monthly profiles provided by the Multiresolution Emission Inventory for China (MEIC; Li et al., 2017).

Our goal is to provide a new set of global and European temporal profiles. Current datasets typically use the same temporal profiles for certain sectors and/or regions. For example, ECLIPSE and EMEP share the same monthly profiles for the energy sector in Europe and Russia. Similarly, TNO and EDGAR share the same monthly profiles for residential combustion and road transport (Friedrich and Reis, 2004), as well as for the energy industry (Veldt, 1992) and agriculture (Asman, 1992). In these two datasets, temporal profiles are mostly assumed to be both country- and meteorology-independent. The only exceptions

are, in the case of EDGAR, for the residential and agricultural sectors, which are approximated as a function of the geographical zone: the seasonality assumed in the northern hemisphere is shifted by six months in the southern hemisphere, and a flat profile is assumed along the equator. In the case of EMEP, the reported monthly and weekly profiles do consider differences across countries but are primarily based on old sources of information from the 90s and beginning of the 00s, and subsequently neglect behavioural changes that may have happened over the last years. Similarly, road transport weekly and hourly factors reported

by TNO are based on long time series of Dutch data registering the traffic intensity between 1985 and 1998. Moreover, variable climate conditions and changes in meteorology that may cause differences in the temporal weight factors within a country are not accounted for. In order to overcome this limitation, the ECLIPSE monthly profiles for the residential combustion sector were computed using global gridded temperature data and provided as monthly shares for each grid cell (Klimont et al., 2017).

This work presents the Copernicus Atmosphere Monitoring Service TEMPOral profiles (CAMS-TEMPO), a new dataset of global and European emission temporal profiles for atmospheric chemistry modelling. The development of CAMPS-TEMPO comes from the need of overcoming the aforementioned limitations of current profiles (i.e. use of outdated source of information and neglection of the temporal variation of emissions across species and countries/regions) and improving the representation of the emission temporal variations, which was defined as a priority task within the Copernicus Global and

Regional emissions service (CAMS_81) directly supporting the CAMS production chains (https://atmosphere.copernicus.eu/). Multiple socio-economic, statistical and meteorological data were collected and processed to create the profiles. The CAMS-TEMPO dataset includes monthly, weekly, daily and hourly temporal profiles for the priority air pollutants ($NO_x$, $SO_x$, NMVOC, $NH_3$, CO, $PM_{10}$, $PM_{2.5}$) and the greenhouse gases ($CO_2$ and $CH_4$) and each of the following anthropogenic source categories: energy industry, residential combustion, manufacturing industry, transport (road traffic and air traffic in airports)

and agriculture. Depending on the sector and temporal resolution, the profiles are either fixed (spatially-constant) or vary spatially by country or region, and can be pollutant-dependent and/or year-dependent. The CAMS-TEMPO profiles introduce multiple novel aspects when compared to the current profiles used for air quality modelling, including: (i) pollutant-dependency, (ii) spatial variability and (iii) meteorological influence. Table 1 summarizes and compares the main

characteristics of the CAMS-TEMPO profiles with the ones reported in other datasets including TNO (Denier van det Gon et al., 2011), EMEP (Simpson et al., 2012), EDGARv4.3.2 (Janssens-Maenhout et al., 2019) and EDGARv5 (Crippa et al., 2020) regarding spatial coverage, temporal and spatial resolution and yearly/meteorology dependency.

The CAMS-TEMPO profiles were created following the domain descriptions (resolution and geographical area covered) and emission sector classification system defined in the CAMS GLOBal ANThropogenic inventory (CAMS-GLOB-ANT) and CAMS REGional inventory for Air Pollutants and GreenHouse Gases (CAMS-REG_AP/GHG) emission inventories, also developed under CAMS_81 (Granier et al., 2019).

The CAMS-GLOB_ANT dataset (Elguindi et al., 2020a) is a global emission inventory developed for the years 2000-2020 at a spatial resolution of 0.1x0.1 deg. in support of the CAMS global simulations. The data is based on the EDGARv4.3.2 annual emissions developed by the European Joint Center (JRC, Crippa et al., 2018) for the years 2000-2012. After 2012, the emissions are extrapolated to the current year using linear trends fit to the years 2011-2014 from the CEDS global inventory (Hoesly et al., 2018) which provide historical emissions for the 6th IPCC Assessment Report (AR6). Emissions are provided for the main pollutants and greenhouse gases, together with a speciation of NMVOCs based on Huang et al. (2017). A comparison of CAMS-GLOB-ANT emissions to the other inventories is presented in Elguindi et al (2020b).

The CAMS regional emissions are being prepared for air pollutants and greenhouse gases (CAMS-REG_AP/GHG), in support of the CAMS regional production systems and policy tools. The inventory is built up largely using the official reported emission data from individual countries in Europe for each source category, which has as main advantage that it takes into account country specific information on technologies, practices and associated emissions. Where these data were either unavailable or not fit for purpose, these were replaced with other estimates. Then, a consistent spatial distribution is applied across Europe at a resolution of 0.1x0.05 deg. by means of using proxies for each source category of emissions. These proxies include among others point source emissions from E-PRTR, road networks, land use and population density information. Shipping emissions are taken from the STEAM model (Johansson et al., 2017). By also providing speciation profiles for PM and VOC, as well as default height profiles, the dataset is fit-for-purpose for air quality modelling at the European scale. Different versions of the CAMS regional emissions are available, where the latest version (v4.2) covering the years 2000-2017 was produced in early 2020. The methodology used for an earlier version of this inventory is available (Kuenen et al., 2014) and a new manuscript is currently in preparation for the latest version (Kuenen et al., in preparation).

The paper is organized as follows. Section 2 describes, for each sector, the approaches and sources of information used to develop the CAMS-TEMPO profiles. Section 3 discusses the obtained temporal profiles and compares them to currently available datasets. Sections 4 provides a description of the data availability and finally Sect. 5 presents the main conclusions of this work.

## 2 Methodology

### 2.1 Overview

The following subsections describe the input data and methodologies used to compute the CAMS-TEMPO emission temporal profiles for each targeted sector: (i) energy industry (Sect. 2.2), (ii) residential/commercial combustion (Sect. 2.3), (iii) manufacturing industry (Sect. 2.4), (iv) road transport (Sect. 2.5), (v) aviation (Sect. 2.6) and (vi) agriculture (Sect. 2.7).

The CAMS-TEMPO dataset consists of a collection of global and regional temporal factors that follow the domain description
and sector classification reported by the CAMS-GLOB_ANT and CAMS-REG_AP/GHG emission inventories. In order to better distinguish between the two sets of profiles, we refer to them as CAMS-GLOB-TEMPO (https://doi.org/10.24380/ks45-9147, global temporal profiles associated to the CAMS-GLOB_ANT inventory) and CAMS-REG-TEMPO (https://doi.org/10.24380/1cx4-zy68, regional European temporal profiles associated to the CAMS-REG_AP/GHG inventory). Depending on the pollutant source and temporal resolution (i.e. monthly, weekly, daily, hourly), the resulting profiles are
reported as spatially invariant (i.e. a unique set of temporal weights for the entire domain, Tables A.1 to A.4 in Appendix A of this work) or gridded values (i.e. temporal weights vary per grid cell). Similarly, depending on the characteristics of the input data used and approaches to compute the profiles, these can be year dependent and/or pollutant dependent. The spatial resolution of the gridded profiles is 0.1x0.1 deg for CAMS-GLOB-TEMPO and 0.1x0.05 deg for CAMS-REG-TEMPO. In the case of CAMS-REG-TEMPO, the domain covered by the dataset is: 30° W – 60° E and 30° N – 72°N.


Table 2 and Table 3 summarise the characteristics of each temporal profile included in CAMS-GLOB-TEMPO and CAMS-REG-TEMPO datasets, respectively. The sector classification for each case corresponds to those used in CAMS-GLOB_ANT and CAMS-REG_AP/GHG. The specificity of the computed profiles depends upon the degree of sectoral disaggregation used in the original CAMS inventories. For example, the CAMS-GLOB_ANT dataset reports emissions from power/heat plants
and refineries under the same sector ("ene", see Table 2) and therefore a common set of temporal profiles had to be assumed for the two types of facilities. In contrast, the CAMS-REG_AP/GHG inventory reports power/heat plants under the GNFR_A category (Public power) and refineries under sector GNFR_B (Industry), together with all manufacturing industries (Table 3). All the assumptions made regarding this topic are clearly stated in each subsection.

For both CAMS-GLOB-TEMPO and CAMS-REG_AP/GHG , the sum of all temporal weight factors is equal to 12 for monthly profiles, 7 for weekly profiles, 365 or 366 (in case of a leap year) for daily profiles and 24 for hourly profiles. Note that the hourly temporal profiles in CAMS-TEMPO are provided in local standard time (LST). The conversion from LST to coordinated universal time (UTC) as a function of time zones is a process that needs to be performed by the final user. Time zone adjustments is a process typically performed by the emission processing systems/tools designed to adapt emission data
to the air quality modelling requirements (e.g. Guevara et al., 2019).

## 2.2 Energy industry

The temporal profiles computed for the energy industry are reported under the *ene* sector in CAMS-GLOB-TEMPO and the *GNFR_A* category in CAMS-REG-TEMPO. The temporal variability of emissions from this sector was estimated from electricity production statistics under the assumption that it largely depends upon the combustion of fossil fuels in power and heat plants. This approximation is consistent with the definition of the *GNFR_A* sector in the CAMS-REG_AP/GHG dataset. The representativeness of the computed profiles is likely lower in CAMS-GLOB_TEMPO because the *ene* sector also includes other facilities such as refineries.

As shown in Table 2 and Table 3, the profiles reported for this sector include pollutant and country dependent monthly, weekly and hourly factors. The electricity production dataset compiled to derive profiles for this sector were as follows:

- The ENTSO-E Transparency Platform (Hirth et al., 2018; ENTSO-E, 2018): The European Network of Transmission System Operators for Electricity (ENTSO-E) centralizes the collection and publication of the electricity generation per production type for each European Member State. The information published by the Transparency Platform is collected from data providers such as Transmission System Operators (TSOs), power exchanges or other qualified third parties. The information collected included production data (MW) per country and fuel type (i.e. lignite, hard coal, natural gas, oil, biomass) at monthly (years 2010 to 2014) and hourly (years 2015 – 2017) levels.

- The US EPA emission modelling platform (US EPA, 2019a): The Environmental Protection Agency (EPA) maintains an emission modelling platform that includes processed and clean hourly emission data derived from a Continuous Emission Monitoring System (CEMS). The information collected includes hourly $NO_x$, $SO_x$ and heat input data for individual power plants in years 2011 and 2014.

- The IEA electricity statistics (IEA, 2018): The International Energy Agency (IEA) provides consistent electricity statistics split by generation type (i.e. total fossil fuels, nuclear, hydro, and geothermal/other) and country. The information collected included monthly data for the years 2010 to 2017 for each Member country of the Organisation for Economic Co-operation and Development (OECD).

- The MBS Online (MBS, 2018): The Monthly Bulletin of Statistics (MBS) is a database of the United Nations with focus on national economic and social statistics. It provides monthly data of the total electricity gross production per country. The information collected included data for the year 2015.

Figure 1.a illustrates the spatial coverage of the compiled dataset by source of information (i.e. ENTSO-E, US EPA, IEA and MBS). Overall, main emission producers (e.g. China, India, Europe and North America) are covered, while most of the countries with no information available are located in South America and Africa. For those countries with no data, the TNO profiles reported under the energy sector (Denier van der Gon et al., 2011) are used.

The compiled data was first analysed to assess whether interannual variability is important for this sector. Seasonal cycles were computed for different years (2010 – 2017) and countries using the IEA statistics. In the majority of the countries analysed, the monthly profiles were found to be consistent through the different years and to present small interannual variations (Fig. S1). Although some studies have pointed out a temperature dependence of the monthly electricity generation in power plants (Thiruchittampalam, 2014), we neglected it at present. Consequently, we assume the monthly temporal profiles for this sector to be the average over all the available years of data.

### 2.2.1    Monthly profiles

For European countries, monthly profiles were derived using the ENTSO-E dataset. The analysis of the data showed that the seasonality of electricity production varies significantly by fuel type (Fig. S1). The different use of energy sources (i.e. lignite, hard coal, natural gas, biomass and oil) implies that temporal patterns will also vary from one pollutant to the other. For each month, country and pollutant, profiles were calculated following Eq. (1):

$$M_{m,c,p} = \sum_{f=1}^{n} M_{m,c,f} \frac{FS_{c,f}*EF_{f,p}}{\sum_{f=1}^{n} FS_{c,f}*EF_{f,p}} \qquad (1)$$

Where $M_{m,c,p}$ is the monthly factor for month $m$, country $c$ and pollutant $p$; $M_{m,c,f}$ is the is the monthly factor for month $m$, country $c$ and fuel $f$; $FS_{c,f}$ is the fuel share factor for country $c$ and fuel $f$ and $EF_{f,p}$ is the emission factor for fuel $f$ and pollutant $p$. Fuel share factors were obtained by averaging the ENTSO-E production data for years 2010 to 2017 per country, and the emission factors were taken from the EMEP/EEA 2016 emission inventory guidebook for the priority air pollutants (EMEP/EEA, 2016; 1.A.1 Energy industries, Table 3-2, 3-3, 3-4, 3-5 and 3-7) and from the IPPC guidelines (IPCC, 2006; Volume 2: Energy, Table 2.2) for GHGs (Table 4). We note that only fuels with shares larger than 10% were considered. For instance, in the case of Austria, only hard coal (25%) and natural gas (65%) were used, and the original shares were normalised so that their sum equalled 100%. This was done to avoid introducing errors due to residual fuels, which may be related to few (or even just one) power plants.

For other countries, monthly factors by pollutant could not be developed as both the IEA and the MSB datasets do not report electricity production split by fuel type. Hence, monthly factors were derived by averaging the available production data per month and relating them to the total production in the year. For the US, $NO_x$ and $SO_x$ monthly profiles were derived from the

corresponding hourly measured emissions reported by the EPA's CEMS data. Measurements from all individual plants were averaged at the monthly level and then normalised to sum 12. The seasonality for the other pollutants (i.e. NMVOC, $NH_3$, CO, $PM_{10}$, $PM_{2.5}$, $CO_2$, $CH_4$) was linked to the measured heat input, following Stella (2005).

### 2.2.2 Weekly profiles

Weekly profiles were developed for Europe using the hourly electricity production data reported by ENTSO-E. As in the case of the monthly profiles, weekly scale factors were found to significantly vary according to the type of fuel (Fig. S1). These results are in line with the conclusions of Adolph (1997), which identified three generic weekly profiles—base, medium and peak load—as a function of the type of power plant. Pollutant-related weekly profiles were developed following the same methodology applied for obtaining the monthly weight factors (Eq. (1)).

For the US, the CEMS data was used to compute pollutant-dependent profiles following the same procedure as described in Sect. 2.2.1. Measurements from all individual plants were averaged per day of the week and then normalised to sum 7. For countries with no information on daily electricity production data, we used the weekly profile reported in the TNO dataset for the energy sector.

### 2.2.3 Hourly profiles

Hourly profiles were developed for Europe and the US using the hourly electricity production data reported by ENTSO-E and the measured emissions reported by CEMS, respectively. As previously seen, large differences are observed between fuels. Profiles related to the so-called base peak load power plants (i.e. annual useful life of more than 4000 hours) present a rather flat distribution, whereas in other cases the change in energy production between day and night is relatively high (Fig. S1).

Pollutant-related hourly profiles were developed following the same procedure shown in Eq. (1). For countries with no information on hourly electricity production data, we assumed the hourly profile reported in the TNO dataset. Some studies have suggested that the hourly variation of power plant activities may vary according to the season of the year (Thiruchittampalam, 2014). This feature is not considered in the present version of the CAMS-TEMPO profiles, and will be 255 addressed in future releases.

## 2.3 Residential/commercial combustion

The temporal profiles computed for the residential/commercial sector are reported under the *res* sector in CAMS-GLOB-TEMPO and the *GNFR_C* category in CAMS-REG-TEMPO. The temporal variability of emissions for this sector is assumed to be dominated by the stationary combustion of fossil fuels in households and commercial/public service buildings. These 260 categories are also assumed to be the main contributors to the total emissions reported by CAMS-GLOB_ANT and CAMS-

REG_AP/GHG. Other combustion installations activities included under this sector (i.e. plants in agriculture/forestry/aquaculture and other stationary including military) are assumed to follow the same temporal profile.

The temporal weight factors developed for this sector include monthly, daily and hourly profiles. The monthly and daily profiles depend upon year and region, and were derived using meteorological parametrisations (Sect. 2.3.1). The hourly profiles depend upon pollutant and region (Sect. 2.3.2).

### 2.3.1 Monthly and daily profiles

Gridded daily temporal profiles were derived according to the heating Degree Day (HDD) concept, which is an indicator used as a proxy variable to reflect the daily energy demand for heating a building (Quayle and Diaz, 1980). This method has been proven to be successful in previous emission modelling works (e.g. Mues et al., 2014; Terrenoire et al., 2015).

The heating degree day factor ($HDD(x, d)$) for grid cell $x$ and day $d$ is defined relative to a threshold temperature ($T_b$) above which a building needs no heating (i.e. heating appliances will be switched off), following Eq. (2):

$$HDD(x, d) = \max(T_b - T_{2m}(x, d), 1) \tag{2}$$

Where $T_{2m}(x, d)$ is the daily mean 2 m outdoor temperature for grid cell $x$ and day $d$ [°C]. This information was obtained from the ERA5 reanalysis dataset for the period 2010 – 2017 (C3S, 2017). As shown in Eq. (2), $HDD(x, d)$ increases with the difference between the threshold and actual outdoor temperatures. A minimum value of 1 is assumed instead of 0 to avoid numerical problems when used in Eq. (3).

A challenge when using this method is to set the threshold or comfort temperature ($T_b$). The choice of $T_b$ depends on local climate and building characteristics, among other. When dealing with an extended area like Europe or even the whole world, it is difficult to choose a unique $T_b$. This value is usually set to 18 °C (e.g., Mues et al. 2014), 15.5 °C (e.g. Spinoni et al. 2015) or even 15 °C (e.g. Stohl et al., 2013). Following the work by Spinoni et al. (2015), which developed gridded European degree-day climatologies, we assumed that $T_b = 15.5$ °C, a value also suggested by the UK MET-Office. A first guess of the daily temporal factor ($FD(x, d)$) for grid cell $x$ and day $d$ is (Eq. 3):

$$FD(x, d) = \frac{HDD(x,d)}{\overline{HDD}(x)} \tag{3}$$

Where $\overline{HDD}(x)$ is the yearly average of the heating degree day factor per grid cell $x$ (Eq. 4)

$$\overline{HDD}(x) = \frac{\sum_1^N HDD(x,d)}{N} \qquad\qquad (4)$$

Where $N = 365$ or 366 days (leap or non-leap year). Considering that residential combustion processes are not only related to space heating but also to other activities that remain constant throughout the year such as water heating or cooking, a second term is introduced to Eq. 5 by means of a constant offset ($f$) (Eq. 5):

$$FD(x,d) = \frac{HDD(x,d) + f * \overline{HDD}(x)}{(1+f) * \overline{HDD}(x)} \qquad\qquad (5)$$

Where $f = 0.2$ based on the European household energy statistics reported by Eurostat (2018) (Table 5). As observed, this share may vary depending by fuel. In the case of biofuels or coal products, which dominate the contribution to total PM emissions, space heating represents 89.1% of the residential combustion processes ($f \sim 0.1$), whereas in the case of natural gas, which is the main contributor to $NO_x$, the share is 77.7% ($f \sim 0.2$). Significant differences between countries are also observed for specific fuels. For instance, in Norway 100% of solid fuels are used for space heating ($f \sim 0$), whereas in Greece this share is only 65% ($f \sim 0.35$), with the remainder being attribute to water heating (27%) and cooking (8%) (Eurostat, 2018). More significant differences can be found in developing regions (e.g. Tibetan Plateau) where the share of solid biofuels used for cooking can go up to 80%. Despite these variations, a generic value of $f = 0.2$ is assumed for all regions. To illustrate how current assumptions may impact the derived profiles in non-European regions, we computed daily factors for the residential sector over India and China for 2015 using a range of $f$ (0, 0.2 and 0.5) and $T_b$ (15.5, 18 °C) values (Fig. S2). We selected these two countries as they produce a large share of the global residential emissions (Hoesly et al., 2018). Differences between the daily factors of up to 55% were found during winter time when comparing the results computed with f = 0.0 and 0.5, indicating that daily factors are sensitive to these parameters. The investigation and proposal of different $f$ values (as well as different $T_b$ values) for different regions of the world will be addressed in future works.

Gridded daily temporal profiles were developed for eight years (2010 to 2017). A climatological daily profile based on the average of each day over all the available years was also produced. Monthly gridded factors were derived from the daily profiles for all the years available. We interpolated the estimated gridded daily factors from the ERA5 working domain (approx. 0.3x0.3 deg.) onto the CAMS-GLOB_ANT (0.1x0.1 deg.) and CAMS-REG_AP/GHG (0.05x0.1 deg.) grids applying a nearest neighbour approach.

### 2.3.2  Hourly profiles

The hourly distribution of residential/commercial combustion activities has typically been described following the profile A presented in Fig.2a, used in both EMEP and TNO datasets. This hourly distribution presents one peak in the morning and

another one in the afternoon, when energy consumption is supposedly higher due to increased space heating or cooking activities. We evaluated this profiles with real-world measurements of natural gas consumption for residential houses in the UK (Retrofit for the Future project, https://retrofit.innovateuk.org/), US Texas (Data Port – Pecan Street dataset, https://dataport.cloud/) and a house in Canada (Makonin et al., 2016). The measurements show the two peaks in all the profiles, although their occurrence and intensity vary due to the specific energy consumption behaviour of each house (Fig. S2). This comparison suggests that profile A is representative of emissions related to natural gas combustion.

We created a second hourly profile (Fig. 2a, profile B) linked to the combustion of residential wood for space-heating purposes using as a basis information derived from citizen interviews performed in Norway and Finland (Finstad et al., 2004 and Gröndahl et al., 2010) as well as from long-term measurements of the wood burning fraction of black carbon in Athens (Athanasopoulou et al., 2017). As shown in the Fig. 2a, the resulting profile B presents an intense peak during the evening hours, but not during the morning, in contrast to Profile A. It is actually a common practice in developed countries to use fireplaces and other types of wood-burning appliances mainly in the evening.

As reported by the World Health Organization (WHO), most of developing countries use wood not only for heating space purposes but also for cooking activities (Bonjour et al., 2013). We created a third profile that represents these activities (Fig. 2a, profile C) based on information derived from continuous indoor $PM_{2.5}$ measurements performed in households in the eastern Tibetan Plateau (Carter et al., 2016). The profile is influenced by heating and cooking practices and therefore presents three peaks that correspond to typical morning, midday, and evening mealtimes.

The results summarised in Fig. 2a indicate that the hourly behaviour of residential combustion emissions varies according not only to the fuel type but also to the type of end-use (i.e. space heating or cooking). Both the CAMS-GLOB_ANT and CAMS-REG_AP/GHG reports total residential/commercial emission as a unique sector, without discriminating by type of fuel or end-use. Therefore, several decisions were made in order to assign the three proposed profiles to different pollutants and regions:

- Profile A: $NO_x$, $SO_x$ and $CH_4$ emissions in urban/rural areas of developed and developing countries.
- Profile B: $PM_{10}$, $PM_{2.5}$, CO, $CO_2$, NMVOC and $NH_3$ in urban/rural areas of developed countries.
- Profile C: All pollutants in rural areas of developing countries

The assumptions made behind this assignment are:

- Natural gas/diesel heating combustion is the main contributor to total $NO_x$, $SO_x$ and $CH_4$ emissions
- Wood combustion is the main contributor to total $PM_{10}$, $PM_{2.5}$, CO, $CO_2$, NMVOC and $NH_3$ emissions
- In the urban and rural areas of developed countries wood is mainly used for heating purposes.
- In the urban areas of developing countries wood is mainly used for heating purposes.

- In the rural areas of developing countries all fuels are used both for heating and/or cooking purposes (i.e. the two activities occur at the same time).

The list of developing countries was obtained from the World Bank Country Classifications (World Bank, 2014). The discrimination of human settlements between urban and rural areas was derived from the Global Human Settlement Layer (GHSL) project (Florczyk et al., 2019; Pesaresi et al., 2019). The GHSL provides a global classification of human settlements on the base of the built-up and population density at a resolution of 1kmx1km corresponding to four epochs (2015, 2000, 1990, and 1975). The 2015 epoch was selected and the original raster was remapped onto the CAMS-GLOB_ANT (0.1x0.1 deg) and CAMS-REG_AP/GHG (0.1x0.05 deg) grids.

## 2.4 Manufacturing industry

The temporal variability of industrial emissions is reported under the sectors *indu* (CAMS-GLOB-TEMPO) and *GNFR_B* (CAMS-REG-TEMPO). Both in the CAMS-GLOB_ANT and the CAMS-REG_AP/GHG inventories, all industrial manufacturing emissions are reported under these single categories. Hence, the same temporal pattern has to be assumed for all types of facilities (e.g. cement plants, iron and steel plants, food and beverage). For this sector, only country-dependent monthly profiles were developed due to the lack of more detailed data.

### 2.4.1 Monthly profiles

Country-specific monthly profiles were estimated using the Industrial Production Index (IPI), which measures the monthly evolution of the productive activity of different industrial branches, including manufacturing activities. The IPI as a monthly surrogate for industrial emissions has been used in previous studies (e.g. Pham et al., 2008; Markakis et al., 2010).

The IPI data was obtained from the MBS database (MBS, 2018) which provides monthly information per country and general industrial branch (i.e. mining, manufacturing, electricity, gas and steam and water supply) for the year 2015. The manufacturing branch, which includes several divisions such as iron and steel industries, chemical industries and food and beverage products, was used to derive country-specific monthly profiles. Figure 1b shows the spatial coverage of the compiled dataset. As in the case of the energy industry sector (Fig. 1a), the lack of information mostly affects Africa and South America. For those countries without available information, the monthly profile reported in the TNO dataset under the industry sector was used (Denier van der Gon et al., 2011). In the case of China, the monthly profile reported in the MIX inventory under the *industry* sector was used (Li et al., 2017).

The time profiles are based on IPI information from 2015 and are assumed to be representative for other years. Our assumption is supported by the low inter-annual variability observed in the IPI values collected from different national statistical offices including Italy (ISTAT, 2018), Norway (SSB, 2018), Spain (INE, 2018) and the UK (ONS, 2018) (Fig. S3). Another implicit

assumption made is that the constructed monthly profiles can be equally applied to all the different industrial activities reported under the *ind* and *GNFR_B* sectors. The national IPI values collected for Italy and the US (Board, 2020) were used to compare the seasonality of individual industrial divisions against to the general manufacturing IPI monthly profile. For both countries, and up to a certain extent, it was found that all the industrial divisions (except food/beverages and petrochemical industry in the case of Italy) follow the seasonality of the general manufacturing profile (Fig. S4), which allow concluding that the assumption made is reasonable. A similar result is reported for Thailand in Pham et al. (2008).

### 2.4.2    Weekly and hourly profiles

Due to the lack of country-specific data, the fixed weekly and hourly temporal profiles provided in the TNO dataset for industry sector is used. The weekly profile assumes a flat distribution during the working days and a slight decrease during weekends (Table A.2). On the other hand, the hourly profile includes an increase of the activity during the central hours of the day (Tables A.3 and A.4).

### 2.5    Road transport

Temporal profiles for road transport emissions are reported under the *tro* sector in CAMS-GLOB-TEMPO and the *GNFR_F1* (exhaust gasoline), *GNFR_F2* (exhaust diesel), *GNFR_F3* (exhaust LPG gas) and *GNFR_F4* (non-exhaust) categories in CAMS-REG-TEMPO. The fact that CAMS-REG_AP/GHG traffic-related emissions are classified into four different categories (discriminated by fuel and type of process) allows considering specific temporal features associated to each one of them, including: temperature dependence of CO and NMVOC gasoline exhaust emissions (*GNFR_F1*), of $NO_x$ diesel exhaust emissions (*GNFR_F2*) and of NMVOC non-exhaust emissions (*GNFR_F4*). On the other hand, in CAMS-GLOB_ANT all traffic emissions are reported under a single sector and subsequently the approach used for the development of the temporal profiles is more simplistic.

The temporal weight factors developed for this sector include monthly, weekly and hourly profiles. As summarised in Table 2, the CAMS-GLOB-TEMPO monthly and weekly profiles constructed for this sector are region-dependent, whereas the hourly profiles vary per region and day of the week (i.e. weekday, Saturday and Sunday). In the case of CAMS-REG-TEMPO, the constructed profiles can vary by region, pollutant, day of the week and/or year, as a function of the source sector and temporal resolution (Table 3). Depending on the dataset and sector category, temporal emission variability is assumed to be either exclusively driven by the traffic activity data (e.g. CAMS-GLOB-TEMP, all cases) or by a combination of traffic activity data and changes in ambient temperature (i.e. CAMS-REG-TEMP, CO and NMVOC *GNFR_F1*, $NO_x$ *GNFR_F2* and NMVOC *GNFR_F3*). For the first case, temporal profiles were developed using traffic count data compiled from multiple sources of information (Sect. 2.5.1). As listed in Table 6, the information was obtained from local and national open data portals, publications or through personal communications. The spatial coverage of the compiled dataset is illustrated in Fig. 1c. For

the second group of profiles, the temporal variability of traffic activity was combined with meteorological parametrisations available in the literature (Sect. 2.5.2).

Considering that for each traffic count dataset the reference years are different (Table 6), we analysed the data in view of differences in the resulting profiles as a function of the year. We took the Paris city traffic data as an example since it covers a wide range of years (2013 to 2017). For each year, monthly, weekly and hourly (i.e. Wednesday and Saturday) profiles were constructed (Fig. S5). The results suggest that temporal patterns in vehicle activity do not change much over long time scales. Consequently, and following the assumptions made for the energy and manufacturing industry sectors, we assumed that the inter-annual variability can be negligible. Hence, all profiles developed only as a function of traffic count data were constructed by averaging the values (per month, day of the week or hour of the day) over all the available years.

Some of the compiled datasets (e.g. Germany and California) report the traffic counts classified by vehicle type (i.e. light-duty vehicles, LDV and heavy-duty vehicles, HDV). The monthly, weekly and hourly profiles as a function of the vehicle type showed significant differences, especially for weekly and hourly profiles (Fig. S6). HDV traffic presents a larger decrease on the weekend than LDV. Moreover, the hourly LDV profile exhibits two distinct (morning and evening) commuter-related peaks, whereas HDV shows a single midday peak. These results highlight the importance of applying separate temporal profiles to characterize traffic and associated emissions for LDV and HDV. However, in the present work this disaggregation was not considered since both the CAMS-GLOB_ANT and CAMS-REG_AP/GHG inventories report LDV and HDV-related emissions under the same pollutant sector. When only vehicle-type temporal profiles were available (i.e. California), the information reported for LDV is used, as this type of vehicle dominates the temporal distribution of total traffic flow.

### 2.5.1    Meteorology-independent monthly profiles

A comparison between monthly variation in traffic patterns at urban and rural locations (i.e. urban streets and highways) was performed for selected countries/regions including California, Germany, Spain and UK. For UK and California, the original traffic statistics were already discriminated by type of location. For the German and Spanish datasets, each traffic station was classified as urban or rural considering its geographical location and the GHSL human settlement classification dataset (Sect. 2.3.2). As shown in Fig. 3, while there is little seasonal variation in German urban locations, rural areas tend to exhibit a stronger seasonality, with a peak occurring during summertime, presumably due to increased recreational and vacation-related driving. The results derived from California, Spain and UK are consistent with these patterns (Fig. S7).

The datasets collected from several cities (i.e. Athens, Barcelona, Berlin, Copenhagen, London, Madrid, Mexico City, Milano, Oslo and Paris) were also used to construct monthly profiles (Fig. 4a). For comparison purposes, the TNO road transport profile is also included (Denier van der Gon et al., 2011). The three southern European cities (i.e. Athens, Madrid and Milano) together with Paris present a similar pattern, with a significant decrease of the activity during the month of August due to the

summer holidays. Similarly, Northern European cities (i.e. Copenhagen and Oslo) also present a decrease in summer, but of a lower intensity and during July. On the other hand, the seasonality observed in London, Berlin and Mexico City is rather flat and closer to the TNO profile.

Results in Fig. 3b and Fig.4a showed that: (i) monthly variations can significantly differ among countries and (ii) within a country, traffic regimes show differences according to the location (urban, rural). Considering all of the above, country and region (urban, rural) specific monthly profiles were constructed based on the traffic information compiled. For countries without any available temporal factors, assumptions were made considering geographical proximity. For instance, the urban profile for Scandinavian countries without data (i.e. Finland, Sweden and Iceland) was constructed by averaging the profiles of Oslo and Copenhagen. On the other hand, the rural profile constructed for Spain was assigned to other Southern European countries (i.e. Italy, Greece, Malta, Croatia, Bosnia and Herzegovina, Montenegro, Albania, Slovenia, Cyprus and Portugal). Similarly, the seasonality in Canada was assumed to be equal to the one observed in the US. For all the countries not listed in the table, the urban/rural profiles developed for Germany were assumed. This approach may be further improved as more traffic count data becomes available. In the case of China and India, profiles were derived from the MIX emission inventory (Li et al., 2017).

Two main assumptions underlie these profiles. First, the differences among cities within a country are assumed to be small and therefore we use a unique urban profile therein. The second hypothesis was to assume that all the streets/highways located in urban/rural areas present the same seasonality. While this is a reasonable assumption overall, individual traffic count stations can show particular features. For instance, on certain highways near city entrances or crossing urban areas, traffic intensity shows a flat distribution without the typical summer anomalies. This level of detail, which would require a specific temporal profile per road segment, is out of the scope of this dataset but may be explored in future works.

### 2.5.2 Meteorology-dependent monthly profiles

The seasonality of traffic emissions can be affected by temperature. As shown by Zheng et al. (2014), during winter months vehicles in China produce 19% more CO and 11% more NMVOC than in the summer due to the higher contribution of cold-start emissions. This study also showed that the monthly pattern of emissions differs remarkably by latitude, which is explained by the large contribution of cold-start emissions and the relationship between latitude and temperature. More recently, Keller et al. (2017) and Grange et al. (2019) identified strong temperature dependence for diesel vehicle $NO_x$ emissions. On the basis of measurements of real-world vehicle emissions, both studies concluded that light-duty diesel NOx emissions are highly dependent on ambient temperature, with low temperatures resulting in higher NOx emissions (up to +80% for temperatures under 0ºC in EURO5 diesel cars; Keller et al., 2017). Grange et al. (2019) also highlighted the spatial heterogeneity of the so-called "diesel low temperature NOx emission penalty" throughout Europe due to the different climate conditions.

We used available parametrizations (Eq. 6 to Eq. 8) to account for the meteorological drivers of the seasonality of CO and NMVOC gasoline-related emissions (*GNFR_F1*) (US EPA, 2015) and $NO_x$ diesel-related emissions (*GNFR_F2*) (Keller et al., 2017):

$$FM_{CO}(x,m) = \begin{cases} e^{(-0.038*(T_{2m}(x,m)-75)}, & T_{2m}(x,m) \leq 75\ °F \\ 1, & T_{2m}(x,m) > 75\ °F \end{cases} \qquad (6)$$

$$FM_{NMVOC}(x,m) = \begin{cases} e^{(-0.048*(T_{2m}(x,m)-75)}, & T_{2m}(x,m) \leq 75\ °F \\ 1, & T_{2m}(x,m) > 75\ °F \end{cases} \qquad (7)$$

$$FM_{NOx}(x,m) = \begin{cases} 1.64, & T_{2m}(x,m) \leq 0\ °C \\ -0.034*T_{2m}(x,m)+1.64, & 0\ °C < T_{2m}(x,m) < 18\ °C \\ 1, & T_{2m}(x,m) \geq 18\ °C \end{cases} \qquad (8)$$

Where $FM_{CO}(x,m)$, $FM_{NMVOC}(x,m)$ and $FM_{NOx}(x,m)$ are the CO, NMVOC, and $NO_x$ gridded monthly profiles for grid cell $x$ and month $m$; and $T_{2m}(x,m)$ is the monthly mean 2 m outdoor temperature for grid cell $x$ and month $m$. Note that $T_{2m}(x,m)$ is expressed in °F in Eq. 6 and Eq. 7 and in °C in Eq. 8. This difference is due to the fact that the first two expression are
500 derived from the North American MOtor Vehicle Emission Simulator (MOVES), while the last one is derived from the European Handbook Emission Factors for Road Transport (HBEFA) model. Monthly gridded 2 m temperature was taken from ERA5 for the period 2010 – 2017 (CS3, 2017). The obtained monthly profiles where normalised so that their total sum equals 12.

The meteorology-dependent monthly profiles were then combined with the meteorology-independent ones (Sect. 2.5.1) so that the resulting seasonality accounts for both temperature influences and traffic activity. For CO, we used a weight factor of 45% for the temperature-dependent profiles and of 55% for the traffic activity ones, following the UK National Atmospheric Emissions Inventory (NAEI), which reports road transport annual emissions and distinguishes between cold start and hot exhaust. Due to the lack of information, we assumed the same share for all countries. Likewise, for NMVOC profiles we
assumed a 33% weight for the temperature-dependent temporal factors (i.e. emissions related to cold-start) and 67% for the ones derived from traffic counts (i.e. emissions related to hot exhaust). Finally, for $NO_x$ we assumed a 50% and 50% split, since the "diesel low temperature NOx emission penalty" affects total exhaust diesel emissions (cold start and hot).

In addition to the meteorological-dependent profiles described above, we created a specific monthly profile for NMVOC
evaporative emissions (*GNFR_F4*) based on recent results obtained with the High-Elective Resolution Modelling Emission System (HERMESv3) (Guevara et al., 2020c). The HERMESv3 model computes hourly gasoline evaporative emissions from standing cars (diurnal losses) using the Tier 2 approach reported in the EMEP/EEA emission inventory guidelines 2016.

Summer and winter temperature-dependent emission factors are defined for each type of vehicle as a function of the 2 m outdoor temperature obtained from the ERA5 reanalysis. The HERMESv3 model was run over Spain for year 2016 at a spatial resolution of 4x4 km and a temporal resolution of 1 hour. The results aggregated by month and normalized (orange line in Fig. 4a and Table A.1) show a strong seasonality, with emissions increasing up to 100% during summer. We used this profile as a first step that reflects the different dynamics of exhaust and evaporative emissions. Future works may consider developing region-specific profiles for NMVOC evaporative-related emissions.

### 2.5.3    Weekly profiles

Country- and region- (urban, rural) specific weekly profiles were constructed based on the traffic information summarised in Table 6. In contrast to urban areas, rural traffic activity is lower during weekdays and decrease relatively less during the weekend, especially on Sundays (Fig. 3c and Fig. S7). Figure 4b shows how, depending on the location, the intensity of the weekend decrease is relatively higher (e.g. Madrid) or lower (e.g. Mexico City), which is likely due to different sociodemographic patterns. We used the weekly profile provided in the TNO dataset as the urban profile for the countries where local information is not available. Similarly, we used an average profile including data from Germany, Spain and UK as the rural profile for countries without data. The resulting profiles were assigned equally to all the traffic-related categories of both CAMS-GLOB-TEMPO and CAMS-REG-TEMPO, with the exception of NMVOC gasoline evaporation (*GNFR_F4*), for which a flat profile is proposed.

### 2.5.4    Hourly profiles

The analysis of hourly profiles constructed per day of the week for six cities (i.e. Berlin, Madrid, Milan, Oslo, Paris and Utrecht) clearly highlights the need to create hourly profiles per day type (Fig. S9). Weekdays (i.e. Monday to Friday) tend to exhibit strong similarities and reflect commuting patterns that are typically bimodal with morning and afternoon volume peaks. Saturday and Sunday generally show the traffic activity to plateau between late morning and early evening, typically due to a decrease in commuting activity. Some studies have developed distinct hourly traffic profiles for Monday to Thursday, Friday, Saturday and Sunday (e.g. McDonald et al., 2014), and others have discriminated between weekdays and weekends (e.g. Zheng et al., 2009). CAMS-TEMPO includes hourly profiles that vary among weekdays, Saturdays and Sundays, following other studies such as Menut et al. (2012).

Hourly variations during weekdays at urban and rural locations were compared for selected countries and regions (Fig. 3d and Fig. S8). Morning traffic peaks associated with commuting are found in and near cities, but not in rural areas (see California and Guangzhou in Fig. S8). Lunch time peaks tend to be higher in rural areas, mainly due to the activity of HDV (Spain and Germany). In contrast, hourly variations on Saturday and Sunday were found to be very similar in urban and rural areas for all the available datasets (not shown). Consequently, only for weekdays we differentiated the hourly profiles of urban and rural areas. Also, in this case, the GHSL dataset was used to assign the respective profiles to either urban or rural grid cells.

Weekday hourly profiles constructed for different cities are shown in Fig. 4c. Two groups of profiles (showed in red and blue) with similar behaviours were identified. For the first group (in red), the rush hours in the morning and in the evening can be clearly identified. The occurrence of the peaks varies from one city to another due to different sociodemographic patterns. In the second group (in blue), a maximum level of activity is reached in the morning (between 07:00 and 08:00 h) that largely

remains for the rest of the day-time period (i.e. 07:00 to 19:00 h) and through part of the night time (i.e. 19:00 to 21:00 h). The hour when traffic activity reaches the maximum level also varies from one city to the other. Besides the potential effect of different social habits, the difference between the two groups of profiles could be also associated with differences in the vehicle densities. For instance, Oslo, Utrecht and Berlin (first group) have vehicle densities of 0.6, 1.1 and 1.3 veh/km$^2$ (in thousands), whereas in Barcelona, Madrid and Milano (second group) the densities are much higher (i.e. 5.2, 2.2 and 3.9, respectively)

(AB, 2017).

Figure 4d shows the constructed Saturdays hourly profiles for selected cities. As before, two groups of profiles showing similar patterns are highlighted. The profiles related to the first group (in red) tend to present larger activity levels during day time (between 09:00 h and 18:00 h), whereas in the second group (in blue) weight factors are higher during night time (between

21:00 and 03:00 h). A similar pattern is observed with Sunday hourly profiles (not shown).

The resulting profiles were assigned equally to all the traffic-related categories with the exception of NMVOC evaporative emissions (*GNFR_F4*), in which we use a specific hourly profile based on HERMESv3 (Sect. 2.5.2). The resulting profile is shown in Fig. 4c and d (yellow lines) and Tables A.3 and A.4.


For European and North American countries without any available temporal factors, assumptions were made considering geographical proximity as described in Sect. 2.5.1. For China, the profiles were computed as an average of the weight factors reported for Beijing and Guangzhou. For all the rest of the cases (mainly Africa, Latin America and Asia), the urban/rural profiles developed for Germany were assumed as default, as they were based on the largest number of traffic count stations

(more than 1500). This approach may, of course, be improved but was constrained in this study by the traffic count data availability.

## 2.6    Aviation

The temporal profiles developed for air traffic emissions during Landing and Take-Off cycles (LTO) in airports are reported under the *GNFR_H* category in the CAMS-REG-TEMPO dataset. Country-dependent monthly temporal profiles were

constructed using airport traffic data, as described below. Due to the lack of country-specific data, a fixed hourly temporal profile is proposed. We could not consider this sector for CAMS-GLOB-TEMPO since it is excluded in the CAMS-

GLOB_ANT inventory. Aviation emissions are reported in a separate inventory called CAMS-GLOB-AIR, in which emission from LTO cycles are reported together with climbing, descent and cruise airplane operations.

### 2.6.1 Monthly profiles

We collected monthly airport traffic data by reporting airport for the years 2011 to 2017 from the Eurostat statistics (Eurostat, 2019). Year 2010 was excluded from the data gathering process due to the air travel disruption in North and Central Europe caused by the Eyjafjallajökull eruption. An analysis of the seasonality observed in several airports for each individual year allowed to confirm a low interannual variability (Fig. S10). Consequently, the constructed temporal profiles were based on the average data of all the available years. Country-dependent monthly profiles were derived by aggregating the respective national

airports available in the Eurostat dataset.

### 2.6.2 Weekly and hourly profiles

We assumed flat weekly profiles for this sector as no clear patterns could be found in the available datasets. We use a fixed hourly profile based on airport traffic from the Madrid-Barajas and Barcelona-El Prat airports (AENA, personal communication). The computed fixed profile (Tables A.3 and A.4) was found to be broadly consistent with the hourly

variations reported by other studies (e.g. Unal et al., 2005; Zhou et al., 2019).

### 2.7 Agriculture

Global and regional temporal profiles for the agricultural emissions are reported in two separate sectors: *mma* and *agr* in CAMS-GLOB-TEMPO, and *GNFR_K* and *GNFR_L* in CAMS-REG-TEMPO. In both cases, the former category only includes emissions from livestock (enteric fermentation, manure management), whereas the latter reports emissions from

several activities but mainly fertilizer applications and agricultural waste burning. For both sectors, monthly and daily region-dependent profiles were constructed considering specific meteorological-parametrizations (Sect. 2.7.1 to 2.7.3). For the hourly profiles, only fixed weight factors are proposed due to a data limitation issue (Sect. 2.7.4).

For the livestock sector (*mma* and *GNFR_K*), both in CAMS-GLOB-TEMPO and the CAMS-REG-TEMPO we assumed $NH_3$

and $NO_x$ to arise from the excreta of the animals and follow a meteorology-dependent temporal profile. The rest of pollutants are considered to depend upon of the animal activity (e.g. emissions of PM arise mainly from feed, $CH_4$ from enteric fermentation) and therefore we assumed a flat temporal profile. In the other agricultural categories (*agr* and *GNFR_L*), $NH_3$ emissions are assumed to be mainly related to fertilizer application, while the other criteria pollutants (i.e. $NO_x$, $SO_x$, NMVOC, CO, PM10, PM2.5) and $CO_2$ are dominated by agricultural waste burning. For the particular case of $CH_4$, which is mostly

emitted from rice fields, no particular profiles are proposed yet, this will be addressed in future works. All in all, the profiles created for agricultural emissions depend on the pollutant, year and specific sector.

### 2.7.1 Monthly and daily profiles for livestock emissions

For the livestock sector (*mma* and *GNFR_K*), the temporal variation of $NH_3$ and $NO_x$ emissions are assumed to be depend on temperature and ventilations rates (Gyldenkærne et al., 2005; Skjøth et al., 2011) (Eq. 9):


$$FD_m(x,d) = \frac{T_m(x,d)^{0.89} * V_m(x,d)^{0.26}}{\sum_{t=1}^{365} T_m(x,d)^{0.89} * V_m(x,d)^{0.26}} \qquad (9)$$

where $FD_m(x,d)$ is the daily temporal profile for manure management practice *m*, grid cell *x* and day *d*; $T_m(x,d)$ is the average daily temperature associated to manure management practice *m*, grid cell *x* and day *d* (in °C), and $V_m(x,d)$ is the

average daily ventilation rate associated to manure management practice *m*, grid cell *x* and day *d* (in m·s⁻¹). The manure management practices considered include housing in open barns, housing in closed barns and storage. For each category, the values of $T_m(x,d)$ and $V_m(x,d)$ are estimated following the expressions reported by Gyldenkærne et al. (2005) and Skjøth et al. (2011). For instance, in the case of storage, $V_m(x,d)$ and $T_m(x,d)$ are assumed to be equal to the 10 m outdoor wind speed and 2 m outdoor temperature, respectively. The meteorological information needed to compute $V_m(x,d)$ and $T_m(x,d)$ values

was obtained from ERA5 for the period 2010 – 2017 (CS3, 2017).

Since both the CAMS-GLOB_ANT and CAMS-REG_AP/GHG inventories report livestock emissions under a unique category, with no discrimination by manure management practice, the computed $FD_m(x,d)$ values had to be averaged in order to derive general daily factor values ($FD(x,d)$). The averaging was performed considering country- and grid-dependent weight

factors following Eq. 10:

$$FD(x,d) = \sum_{m=1}^{3} FD_m(x,d) * f(x)_m \qquad (10)$$

The weight factors ($f(x)_m$) were constructed using as a basis: (i) the national EMEP emissions reported by the Centre on

Emission Inventories and Projections (EMEP/CEIP, 2019) for countries that are Members of the EMEP programme and (ii) the global bottom-up inventory of $NH_3$ emissions (MASAGE_NH3) (Paulot et al., 2014) for the rest of the world. Table S1 summarises the weight factors used for each country. For countries not included in the list, an average weight factor is used.

The resulting gridded daily temporal profiles were developed for eight years (2010 to 2017). Using these time series as a basis,

and following with the procedure described in Sect. 2.3.1, a climatological daily profile and monthly gridded factors were also produced.

### 2.7.2 Monthly and daily profiles for fertilizer-related emissions

The seasonality of $NH_3$ emissions from fertilizer application depends mainly on the magnitude and timing of fertilizer application over different crop categories (i.e. planting schedule for each crop). The proposed gridded monthly profiles for this pollutant sector are based on a mosaic of multiple bottom-up agricultural emission inventories, which include information on local crop calendars in their emission estimations. The datasets included in the mosaic are: (i) the global bottom-up gridded MASAGE_NH3 inventory (Paulot et al., 2014), (ii) the regional gridded Chinese emission inventory reported by Zhang et al. (2018), (iii) the regional gridded North American National Emission Inventory (NEI) reported by US EPA (2019b) and (iv) the regional European emission inventories reported for Denmark, Germany (Skjoth et al., 2011), (v) Poland (Werner et al., 2015) and (vi) Netherlands, France and Belgium (Backes et al., 2016). From the MASAGE_NH3 and Zhang et al. (2018) inventories, we considered the original monthly $NH_3$ emissions reported under the use of fertilizers emission categories. From the NEI inventory, we used as a basis the original hourly gridded emissions reported under the species "NH3_FERT", which includes the amount of $NH_3$ from fertilizer sources, and aggregated them to the monthly level. For all cases, the monthly and gridded emissions were first normalised to sum 12 and then remapped onto the CAMS-GLOB_ANT and CAMS-REG_AP grids applying a nearest neighbour approach.

With the objective of computing daily variations, the gridded monthly profiles obtained using the aforementioned mosaic approach were combined with the daily meteorological parametrizations reported by Gyldenkærne et al. (2005) and Skjøth et al. (2011) (Eq. 11):

$$FD(x,d) = \frac{e^{0.0223 * T_{2m}(x,d)} * e^{0.0419 * W_{10m}(x,d)}}{\sum_{t=1}^{365} e^{0.0223 * T_{2m}(x,d)} * e^{0.0419 * W_{10m}(x,d)}} \tag{11}$$

where $T_{2m}(x,d)$ is the daily mean 2 m outdoor temperature for grid cell $x$ and day $d$ [°C]; and $W_{10m}(x,d)$ is the daily 10 m wind speed for grid cell $x$ and day $d$ [m·s$^{-1}$]. Both parameters were derived from ERA5 for the period 2010 – 2017 (C3S, 2017). The resulting gridded daily temporal profiles includes eight years of data (2010 to 2017) and a climatological profile.

### 2.7.3 Monthly profiles for agricultural waste burning emissions

For all the other criteria pollutants (i.e. $NO_x$, $SO_x$, NMVOC, CO, $PM_{10}$, $PM_{2.5}$) included in the *agr* and *GNFR_L* sectors, we used the monthly gridded profiles reported by Klimont et al. (2016) for the agricultural waste burning category. This temporal representation was developed based on the timing and location of active fires on agricultural land in the Global Fire Database (GFEDv3.1) combined with annual emissions from the Greenhouse Gas and Air Pollution Interactions and Synergies (GAINS) model. The original monthly weights were remapped from the ECLIPSE source grid (0.5x0.5 deg) onto the CAMS-GLOB_ANT and CAMS-REG_AP grids applying a nearest neighbour approach.

#### 2.7.4 Hourly profiles

The hourly distribution of agricultural emissions has typically relied on the profile reported by both TNO and EMEP datasets
(Fig. 2,b). The profile is constructed on the idea that $NH_3$ emission rates from agricultural practices (fertilizer application and
manure management) tend to vary with temperature, usually showing a peak in the middle of the day (Tables A.3 and A.4).
Figure 2b compares this profile with the hourly distributions derived from flux measurements performed in a fertilized corn
canopy in North Carolina (Walker et al., 2013) and from direct measurements performed in a mechanically ventilated swine
barn (James et al., 2012). The similarity observed between the TNO profile and the two measurement-based temporal factors
is significantly high (i.e. emissions peak during the afternoon in each case, starting from 13:00 to 15:00 h Local Time). We
therefore maintained the TNO profile for describing the hourly variation of $NH_3$ agricultural emissions (both for livestock and
fertilizer application).

Regarding the other criteria pollutants (i.e. $NO_x$, $SO_x$, NMVOC, CO, $PM_{10}$, $PM_{2.5}$) included in the *agr* and *GNFR_L* sectors, a
new hourly fixed temporal profile for agricultural waste burning emissions is proposed based on the work by Mu et al. (2011),
where climatological mean hourly cycles were constructed using GOES WF_ABBA active fire satellite observations from full
hemisphere scans during 2007–2009. The constructed profile (Fig. 2b and Tables A.3 and A.4) shows a maximum peak at
midday, which is consistent with high levels of midday fire emissions.

### 3 Results and discussion

In this section, we discuss the obtained temporal profiles for CAMS-GLOB-TEMPO and CAMS-REG-TEMPO. In Sect. 3.1
the profiles are compared to independent observational datasets and in Sect. 3.2 to other existing sets of temporal profiles
currently used under the framework of CAMS.

Figure 5 shows the 0.1x0.1 deg CAMS-GLOB-TEMPO gridded January and August profiles constructed for NMVOC and
$SO_x$ energy industry emissions and for manufacturing industry emissions. For the energy sector, several countries such as
Spain and UK show how the seasonality significantly varies as a function of the pollutant. It is also observed that in many
European countries the manufacturing industrial activity decreases during August due to summer holidays, while an increase
is observed in other countries such as China.

Figure 6 shows two examples of CAMS-GLOB-TEMPO gridded daily profiles for the residential/commercial sector along
with the times series at four geographically or climatically different locations (i.e. Athens; Barcelona; Buenos Aires and Oslo)
for years 2010 and 2017. As expected, the largest factors occur in winter and the lowest ones in summer at all four locations.
According to the results, emissions in Athens, Barcelona or Buenos Aires can be 3 to 5 times higher during the cold periods
(i.e. January in Barcelona and Athens, June in Argentina) than during warm periods (August in Barcelona and Athens, January

in Argentina). In Oslo both the seasonal cycle and daily variability are less pronounced than in the other locations because the differences between daily and annual mean temperatures are generally lower. There is a large inter-annual variability in the four locations. In winter of 2010, Barcelona experienced three cold outbreaks of similar intensity (in January, February and March), whereas in 2017 only one significant episode can be observed (mid-January). Similarly, in 2010 three major peaks are observed in Athens in mid-January, beginning of February and mid-December, whereas in 2017 only one episode stands out above the rest. Results clearly highlight that extreme weather events can strongly affect the temporal profiles and thereby the resulting emissions.

Figure 7 shows examples of the 0.1x0.05 deg CAMS-REG-TEMPO gridded profiles constructed for the different road transport categories, including January and August (monthly) weights for gasoline CO exhaust emissions, and both Friday and Sunday (weekly) and 09:00 and 20:00 h (hourly weekday) weights for all traffic sources except NMVOC gasoline evaporation. The monthly gridded profiles clearly show the influence of outdoor temperature, with weight factors up to 2.5 times higher in January than in August in Eastern Europe and Russia. The weekly profile maps clearly show the decrease in emissions from traffic during the weekend, especially in urban areas. The hourly profiles show the high levels of traffic activity during night time in some Mediterranean countries compared of central and North European ones, and also a clear distinction between urban and rural areas.

Figure 8 shows examples of CAMS-GLOB-TEMPO profiles for the different agricultural emission sources, including April and November (monthly) weights for fertilizer $NH_3$ emissions, daily (i.e. 2017/03/06 and 2017/07/16) weights for livestock $NH_3$ and $NO_x$ emissions, and daily time series for livestock emissions over Spain and Argentina for 2010 and 2017 (Note the country daily times series are spatially aggregated as the actual patterns are variable spatially.). Concerning the fertilizer monthly factors, it is observed that the spatial resolution is not homogeneous across the domain. This is due to the different spatial resolutions of the original datasets used to construct this gridded profile: The MASAGE_NH3 emission inventory is available at a resolution of 2x2.5 degree, while the datasets used for China and USA are reported at resolutions of 0.25x0.25 deg and 12kmx12km, respectively. Despite the coarse resolution, the profiles broadly capture the different seasonality in different parts of the world, with a peak of emissions occurring in April in Europe and in November in Latin America. For the livestock sector, the largest weight factors occur during the summer and the lowest ones during the winter due to the temperature dependence. Nevertheless, and in contrast to what is observed with the residential/commercial daily factors, the signal for livestock emissions is mostly seasonal and the daily fluctuations are relatively small, which makes the interannual variability small.

## 3.1    Comparison to independent observational datasets

We compared the CAMS-TEMPO profiles to independent observational datasets. The comparison is mainly performed at the European level as sufficient data could not be gathered to provide a robust comparison for other regions.

Figure 9 shows the CAMS-TEMPO monthly weight factors for $SO_x$ energy industry in Spain and manufacturing industry in Italy compared against two independent temporal profiles based on $SO_x$ measured emissions from all Spanish power plants (CIEMAT, personal communication) and real-world measurements of industrial natural gas consumption provided by the European Network of Transmission System Operators for Gas (ENTSOG, 2020), respectively. Correlations between the two time series are high in both cases (i.e. r= 0.84 and 0.61, respectively). In the case of energy industry in Spain both profiles show minimum levels during spring and an increase during summer due to the intensive use of air conditioning systems. In Italy, both manufacturing industry profiles present a strong decrease during August and December due to summer and Christmas holidays.

In Fig. 10 we compare our profiles for the residential sector with daily factors derived from minute-resolved measurements of natural gas from a residential house in Canada during 2013 (Makonin et al., 2016). We compared our daily factors estimated using the HDD approach for the grid cell closer to the house and considering two different values of $T_b$ (18 and 15.5 °C). In both cases, CAMS-TEMPO reproduces the temporal variation of the locally measured profile. Both at the beginning and the end of the year, the measured and HDD-based profiles show similar maximum values, which correspond to the periods when outdoor temperature reached the minimum levels (not shown). The results also show that using $T_b = 15.5 \, °C$ we obtain a slightly higher correlation than when assuming $T_b = 18 \, °C$ (0.81 versus 0.76).

In Fig. 11 we compare the CAMS-REG-TEMPO road transport profiles to the temporal variation of air pollutant concentrations measured in Madrid, Milano, Barcelona and and Berlin. CO and $NO_2$ hourly concentrations were obtained from the European Environmental Agency (EEA) download service of validated and official air quality data (EEA, 2019). Four urban traffic stations highly influenced by road transport emissions were considered for the analysis: Escuelas Aguirre (Madrid), Milano-Senato (Milano), Eixample (Barcelona) and B Neukölln-Karl-Marx-Str. 76 (Berlin).

The monthly variability of CO and $NO_2$ measured concentrations in Milano and Madrid were compared against the meteorology-dependent gridded monthly profiles created for CO GNFR_F1 (gasoline exhaust) and $NO_x$ GNFR_F2 (diesel exhaust) at these locations. Weight factors from the closest grid cells closest to each station were selected for the comparison. As shown in Fig. 11a and b, the traffic decrease occurring during summertime in Milano and Madrid is also reproduced in the CO and $NO_2$ levels. At the same time, both the CAMS-REG-TEMPO profiles and the measurement-based profiles show a V shape, indicating higher emissions during winter due to the low temperatures. The correlation between the two times series is of r= 0.82 for Milano and of r= 0.74 for Madrid. we note that the seasonality of the CO and $NO_2$ measurements may be also influenced by other factors, including meteorology and the influence of other emission sources such as residential combustion, and subsequently a perfect correlation between the two datasets cannot be expected.

Similarly, the hourly profiles proposed for urban locations in Spain and Germany during weekdays and Sundays were compared to the hourly variation of $NO_2$ concentrations for Barcelona and Berlin (Fig. 11c and d). In all cases the hourly evolution of $NO_2$ is mostly driven by variations accounted for in the CAMS-TEMPO profiles. For weekdays, the shift of the traffic morning/evening peaks observed between cities is also reproduced in $NO_2$ concentrations. Traffic and $NO_2$ levels in Berlin reach maximum levels between 07:00 and 08:00 h LT, whereas in Barcelona the peak occurs between 08:00 and 09:00 h LT. On the other hand, Sunday night time and morning $NO_2$ concentrations (between 23:00 and 08:00 h LT) are relative higher in Barcelona than in Berlin (in relative terms), which is related to the more intense traffic activity registered in the streets.

## 3.2    Comparison to other temporal profile datasets

We compared the CAMS-TEMPO profiles against the profiles reported by other existing datasets. The comparison is focussed on the profiles that are currently being used for air quality modelling purposes under the framework of CAMS, namely: The global EDGARv4.3.2 (Janssens-Maenhout et al., 2019) monthly profiles (used in the CAMS global production) and the European EMEP (Simpson et al., 2012) and TNO (Denier van der Gon et al., 2011) profiles (used in CAMS regional production).

The comparison of monthly, weekly and hourly profiles for the energy industry sector in selected countries is shown in Fig. 12. It is worth noting that both TNO and EDGAR report the same profile for all countries. At the monthly level, significant differences are observed between CAMS-TEMPO and EDGAR in China and USA. In both cases, the summer peak observed in CAMS-TEMPO (presumably due to the intensive use of air conditioning systems) does not show up in EDGAR profile. In the case of Romania, all profiles show important decreases but at different times of the year: April in CAMS-TEMPO (presumably due to low use of heating/cooling devices), July in TNO/EDGAR and September in EMEP. In the UK, the patterns between the different datasets are more similar, all of them reproducing a V shape, except for the relatively flatter NMVOC emissions profiles in CAMS-TEMPO. Concerning the weekly and hourly profiles, important discrepancies are observed between CAMS-TEMPO and the factors proposed by TNO and EMEP for certain countries. In the CAMS-TEMPO weekly profiles for Austria the intensity of the weekend decrease is relatively higher, while the hourly profiles for Spain are flatter than the ones reported by TNO and EMEP.

The comparison of monthly weight factors for the residential/combustion sector indicates generally low discrepancies between the different datasets (Fig. 13). The largest difference is observed in Greece, where both CAMS-TEMPO and EMEP allocate most of the emissions during wintertime while TNO and EDGAR propose a smoother transition between this season of the year and spring/fall. Both EDGAR and CAMS-TEMPO datasets propose an almost flat profile for India, where residential fuel is mainly used for cooking activities, an activity that can be considered constant throughout the year. For the daily temporal

disaggregation of emissions, TNO and EMEP propose a fixed weekly profile (not shown), which disregards the daily dynamics
inferred by the heating degree day approach considered in CAMS-TEMPO (Fig. 6).

For road transport, the differences between CAMS-TEMPO and other datasets are quite significant. The monthly profiles reported by TNO, EMEP and EDGAR are almost flat, while the pollutant and meteorological dependent profiles developed in the present work suggest important decreases during summertime, especially for the case of CO (Fig. 14a). At the weekly
level, the TNO profile is in line with most of the city-level constructed profiles, although in some cases differences in the intensity of the weekend decrease are observed (Fig. 4b). At the hourly level, the main discrepancies are observed when comparing TNO with the Saturday and NMVOC evaporative profiles of CAMS-TEMPO (Fig. 4d). It is worth noting that the hourly weekday profile constructed for Utrecht is almost perfectly correlated with TNO ($R^2 = 0.97$). This is explained by the fact that the TNO profile was estimated using Dutch traffic data.


Figure 15 shows a comparison of the monthly profiles for the agricultural sector. All the datasets selected for comparison, except for TNO, propose a unique profile for all the different agricultural activities (i.e. livestock, use of fertilizers, agricultural waste burning), which is equally applied to all emissions. The CAMS-TEMPO profiles constructed for $NH_3$ fertilizer emissions show how the peak significantly varies among countries (i.e. May in China, June in India, April in Spain). In the case of India
and China, these peaks are consistent with the $NH_3$ satellite-derived seasonality shown by Warner et al. (2017). It is also worth noting the triple peak in Poland, which is related to the application timings of fertilizers (spring) and manure (summer and autumn) on crops (Werner et al., 2017). In contrast, the profiles reported by TNO and EMEP show a unique peak in April in all countries. In the case of $NH_3$ from livestock, the profiles show a weaker seasonality, but relevant discrepancies in the intensities and occurrences of the emission peaks are also observed (i.e. EDGAR-India, EMEP-Spain). In the case of $PM_{10}$
emissions related to agricultural waste burning activities, the results suggest a good agreement between EMEP and CAMS-TEMPO, with most emissions occurring during fall, after the harvesting period. On the other hand, the profile proposed by TNO presents a double peak, a first one occurring during spring (March and April) and a second one during summer (August).

## 4    Data availability

Gridded maps with all the temporal factors (monthly, weekly, daily, hourly) per sector and year are available as NetCDF files
for the global domain at a resolution of 0.1x0.1 deg (CAMS-GLOB-TEMPOv2.1, https://doi.org/10.24380/ks45-9147, Guevara et al., 2020a) and the European regional domain (30° W – 60° E and 30° N – 72°N) at a resolution of 0.1x0.05 (CAMS-REG-TEMPOv2.1, https://doi.org/10.24380/1cx4-zy68, Guevara et al., 2020b) and can be accessed through the Emissions of atmospheric Compounds and Compilation of Ancillary Data (ECCAD) system with a login account (https://eccad.aeris-data.fr/). For review purposes, ECCAD has set up an anonymous repository where subsets of the CAMS-
GLOB-TEMPOv2.1 and CAMS-REG-TEMPOv2.1 data can be accessed directly (https://www7.obs-mip.fr/eccad/essd-surf-

emis-cams-tempo/). In addition, constructed fixed temporal profiles are available per sector and substance in Appendix A of this work.

## 5    Conclusions

This paper presents the CAMS-TEMPO dataset, a collection of monthly, weekly, daily and hourly emission temporal profiles
for the priority air pollutants ($NO_x$, $SO_x$, NMVOC, $NH_3$, CO, $PM_{10}$, $PM_{2.5}$) and the greenhouse gases ($CO_2$ and $CH_4$) and each of the following anthropogenic source categories: energy industry, residential combustion, manufacturing industry, road transport (exhaust and non-exhaust processes), aviation (LTO cycles) and agriculture (i.e. use of fertilizers, livestock and agricultural waste burning). Depending on the pollutant source and temporal resolution, the resulting profiles are reported as spatially invariant (i.e. a unique set of temporal weights for all the domain) or gridded values (i.e. temporal weights vary per
grid cell). Multiple sources of information – including energy statistics and measured activity data, among others - and meteorology-dependent parametrizations have been collected and adapted to construct the profiles.

The CAMS-TEMPO profiles were designed to be combined with the global and regional anthropogenic emission inventories developed under the framework of Copernicus (CAMS-GLOB_ANT and CAMS-REG_AP/GHG, respectively), and to break
down the original aggregated annual emissions to finer temporal resolutions (up to hourly). In order to ensure this combination, the developed temporal weight factors were constructed at a global 0.1x0.1 deg and regional 0.1x0.05 deg resolution following the domain descriptions and emission sector classification system defined in the each of the inventories.

There are several features that makes the CAMS-TEMPO profiles a major step when compared to the datasets currently being
used for air quality modelling under the framework of CAMS:

- •    Pollutant-dependency: For some sectors, profiles were computed for all species independently in order to account for the variability of the activity patterns and the specific processes that drive their release to the atmosphere. As an example, a distinction was made between the temporal distribution of traffic emissions from non-exhaust PM mainly
driven by traffic activity and non-exhaust NMVOC evaporation mainly driven by outdoor temperature.
- •    Spatial variability: For nearly all sectors, the temporal profiles are made country or even country and region-specific in order to take into account the effects of e.g., different sociodemographic patterns and climatological conditions. For instance, differences between urban and rural traffic activity patterns are considered in the profiles constructed for road transport emissions.
•    Meteorological influence: For the residential/commercial combustion, road transport and agriculture sector, the profiles were constructed using meteorology-dependent parametrizations (e.g. heating degree day, temperature effect

on exhaust traffic emissions) that account for the emissions variability driven by temperature or wind speed. The resulting profiles are year-dependent and cover a timespan that goes from 2010 to 2017.

Several CAMS-TEMPO profiles are analyzed in this paper to illustrate their main characteristics and potential. Moreover, an inter-comparison exercise against independent observational datasets (e.g. real-world measurements of natural gas consumptions, emissions and pollutant concentrations) and other existing sets of temporal profiles was also performed. The comparison between CAMS-TEMPO temporal weight factors and the measurement-based profiles showed in general a high degree of correlation. Despite the scarcity of independent measurements, our comparison suggests a high level of 875     representativeness of the developed profiles. On the other hand, the comparison against other sets of temporal profiles showed important discrepancies, especially for the traffic and agricultural sectors. This comparison highlights some shortcomings of the global and regional profiles currently used in the framework of CAMS, namely the omission of meteorological influences and the neglection of the temporal variation of emissions across sectors, species and/or countries/regions.

It is important to highlight that the continuous growth of the open data movement has been a key element for the successful development of the CAMS-TEMPO data. Services such as the European Open Data Portal (https://data.europa.eu/euodp/en/data) - and similar initiatives at the city level - or the ENTSO-E Transparency platform (ENTSO-E, 2018) are delivering very valuable data for the development of emission-related databases.

**5.1     Limitations of the dataset**

The CAMS-TEMPO dataset provides an updated global and regional (European) picture of the temporal characterization of emissions by aggregated sectors. Despite all the efforts, there are, however, some limitations associated to the current version of the dataset that potential users of the CAMS-TENPO should take into account.

First, the specificity of the computed profiles depends upon the degree of sectoral disaggregation used to report the original CAMS inventories. For instance, monthly profiles per industrial divisions could not be considered, since all manufacturing emissions are reported under a unique sector in both CAMS-GLOB_ANT and CAMS-REG_AP/GHG. Similarly, fuel-weighted profiles that are spatially aggregated to the national-level were had to be considered for the energy industry sector, as original emissions are not split by fuel type. On the other hand, the split by fuel type of traffic emissions in the CAMS-895     REG_AP/GHG inventory allowed to consider meteorological influences associated to e.g. CO gasoline and $NO_x$ diesel emissions, but specific traffic activity dynamics associated to light-duty and heavy-duty vehicles could not be accounted for.

A second limitation is related to the assumptions made when applying the Heating Degree Day approach for the residential/commercial combustion sector. The computation of daily temporal factors was done considering a threshold

temperature and a fraction of non-space heating activities homogenous for all the world (15.5 ℃ and 0.2, respectively). Several studies have highlighted that these two values can vary across regions due to changes in local climate, building characteristics and socio-demographic aspects. For instance, Grythe et al. (2019) concluded that a lower threshold temperature value (i.e. 10 ℃) may be more representative for Scandinavian countries, whereas Daioglou et al. (2012) indicated that in rural areas of developing regions (e.g. India, China, Brazil) the share of biomass used for cooking purposes can be larger than 80%. Region-

dependent HDD parameters should be considered in order to overcome this limitation.

Another important shortcoming of the current CAMS-TEMPO dataset is related to the scarcity of available information in developing regions (i.e. Africa, South America and Asia) to construct detailed profiles for the energy industry, manufacturing industry and road transport sectors. In the current version of the CAMS-TEMPO dataset, a simple gap-filling method has been

implemented, which consist of mainly using the TNO European-based profiles when no national or local datasets are available. The rationale behind this choice is that the TNO profiles have been largely used and tested over the last decade in multiple international modelling exercises such as the Air Quality Modelling Evaluation International Initiative (AQMEII; Pouliot et al., 2015). However, TNO profiles are mostly based on Western European data and therefore the degree of representativeness for other world regions is a source of uncertainty. To address this constraint, the current gap-filling procedure will be reviewed

in the future by constructing world region profiles for countries with geographical, climatological and/or behavioural similarities, following the approach presented by Crippa et al. (2020).

## 5.2    Future perspective

The CAMS-TEMPO dataset represents an effort to improve the temporal characterisation of emission data to be used for

atmospheric chemistry modelling. Future works will include the evaluation of these temporal profiles when used for modelling activities. Through close cooperation with air quality modelers, we expect to obtain feedback on the dataset as well as suggestions for future improvements. Besides that, a number of future updates have also been identified during the present work, including:

• The extension of the year dependent temporal profiles to be in line with the current timespan considered in the CAMS-GLOB_ANT (2000-2018) and CAMS-REG_AP/GHG (2000-2017) inventories.
• The development of new temporal profiles for certain sectors/pollutants that have not currently been considered, including $CH_4$ emissions from agriculture (rice fields) and waste management (landfills).
• The refinement of the heating degree day approach considering region-dependent threshold temperature values.


Besides the identified updates, the investigation of monthly and seasonal changes of emissions using satellite data will be also explored in future works. Global and high-resolution observations of the atmospheric composition provided by missions such

as the Copernicus Sentinel-5 Precursor (S5P) can be of great value to improve the description of the spatio-temporal distribution of emissions (Lorente et al., 2019). Finally, understanding and quantifying the impact of the COVID-19 lockdowns
upon the temporal distribution of emissions will be also an important topic to be studied.

The CAMS-TEMPO profiles are distributed free of charge through the Emissions of atmospheric Compounds and Compilation of Ancillary Data (ECCAD) system (https://eccad.aeris-data.fr/).


**Appendix A**

**Table A1: Fixed monthly temporal profiles per domain, sector and pollutant. The definition of the sectors can be found in Table 2 and Table 3**

| Domain | Sector | Pollutant | 1 | 2 | 3 | 4 | 5 | 6 | 7 | 8 | 9 | 10 | 11 | 12 |
|---|---|---|---|---|---|---|---|---|---|---|---|---|---|---|
| CAMS-GLOB | fef | All | 1.2 | 1.2 | 1.2 | 0.8 | 0.8 | 0.8 | 0.8 | 0.8 | 0.8 | 1.2 | 1.2 | 1.2 |
| CAMS-GLOB | slv | All | 0.95 | 0.96 | 1.02 | 1 | 1.01 | 1.03 | 1.03 | 1.01 | 1.04 | 1.03 | 1.01 | 0.91 |
| CAMS-GLOB | tnr | All | 1 | 1 | 1 | 1 | 1 | 1 | 1 | 1 | 1 | 1 | 1 | 1 |
| CAMS-GLOB | swd | All | 1 | 1 | 1 | 1 | 1 | 1 | 1 | 1 | 1 | 1 | 1 | 1 |
| CAMS-GLOB | mma | $PM_{10}$, $PM_{2.5}$, CO, $SO_x$, NMVOC, $CO_2$, $CH_4$ | 1 | 1 | 1 | 1 | 1 | 1 | 1 | 1 | 1 | 1 | 1 | 1 |
| CAMS-GLOB | agr | $CH_4$ | 1 | 1 | 1 | 1 | 1 | 1 | 1 | 1 | 1 | 1 | 1 | 1 |
| CAMS-REG | GNFR_D | All | 1.2 | 1.2 | 1.2 | 0.8 | 0.8 | 0.8 | 0.8 | 0.8 | 0.8 | 1.2 | 1.2 | 1.2 |
| CAMS-REG | GNFR_E | All | 0.95 | 0.96 | 1.02 | 1 | 1.01 | 1.03 | 1.03 | 1.01 | 1.04 | 1.03 | 1.01 | 0.91 |
| CAMS-REG | GNFR_F4 | NMVOC | 0.74 | 0.69 | 0.74 | 0.84 | 1 | 1.2 | 1.54 | 1.5 | 1.23 | 1.03 | 0.77 | 0.72 |
| CAMS-REG | GNFR_K | $PM_{10}$, $PM_{2.5}$, CO, $SO_x$, NMVOC, $CO_2$, $CH_4$ | 1 | 1 | 1 | 1 | 1 | 1 | 1 | 1 | 1 | 1 | 1 | 1 |
| CAMS-REG | GNFR_L | $CH_4$ | 1 | 1 | 1 | 1 | 1 | 1 | 1 | 1 | 1 | 1 | 1 | 1 |
| CAMS-REG | GNFR_I | All | 1 | 1 | 1 | 1 | 1 | 1 | 1 | 1 | 1 | 1 | 1 | 1 |
| CAMS-REG | GNFR_J | All | 1 | 1 | 1 | 1 | 1 | 1 | 1 | 1 | 1 | 1 | 1 | 1 |


**Table A2: Fixed weekly temporal profiles per domain, sector and pollutant. The definition of the sectors can be found in Table 2 and Table 3**

| Domain | Sector | Pollutant | 1 | 2 | 3 | 4 | 5 | 6 | 7 |
|--------|--------|-----------|-----|-----|-----|-----|-----|-----|-----|
| CAMS-GLOB | ind | All | 1.08 | 1.08 | 1.08 | 1.08 | 1.08 | 0.8 | 0.8 |
| CAMS-GLOB | agr | $PM_{10}$, $PM_{2.5}$, CO, $NO_x$, NMVOC, $CO_2$, $CH_4$ | 1 | 1 | 1 | 1 | 1 | 1 | 1 |
| CAMS-GLOB | fef | All | 1 | 1 | 1 | 1 | 1 | 1 | 1 |
| CAMS-GLOB | slv | All | 1.2 | 1.2 | 1.2 | 1.2 | 1.2 | 0.5 | 0.5 |
| CAMS-GLOB | tnr | All | 1 | 1 | 1 | 1 | 1 | 1 | 1 |
| CAMS-GLOB | swd | All | 1 | 1 | 1 | 1 | 1 | 1 | 1 |
| CAMS-REG | GNFR_B | All | 1.08 | 1.08 | 1.08 | 1.08 | 1.08 | 0.8 | 0.8 |
| CAMS-REG | GNFR_L | $PM_{10}$, $PM_{2.5}$, CO, $NO_x$, NMVOC, $CO_2$, $CH_4$ | 1 | 1 | 1 | 1 | 1 | 1 | 1 |
| CAMS-REG | GNFR_H | All | 1 | 1 | 1 | 1 | 1 | 1 | 1 |
| CAMS-REG | GNFR_D | All | 1 | 1 | 1 | 1 | 1 | 1 | 1 |
| CAMS-REG | GNFR_E | All | 1.2 | 1.2 | 1.2 | 1.2 | 1.2 | 0.5 | 0.5 |
| CAMS-REG | GNFR_I | All | 1 | 1 | 1 | 1 | 1 | 1 | 1 |
| CAMS-REG | GNFR_J | All | 1 | 1 | 1 | 1 | 1 | 1 | 1 |

**Table A3: Fixed hourly temporal profiles per domain, sector and pollutant (Part 1: from 1 to 12h). The definition of the sectors can be found in Table 2 and Table 3.**

| Domain | Sector | Pollutant | 1 | 2 | 3 | 4 | 5 | 6 | 7 | 8 | 9 | 10 | 11 | 12 |
|--------|--------|-----------|---|---|---|---|---|---|---|---|---|----|----|----|
| CAMS-GLOB | ind | All | 0.75 | 0.75 | 0.78 | 0.82 | 0.88 | 0.95 | 1.02 | 1.09 | 1.16 | 1.22 | 1.28 | 1.3 |
| CAMS-GLOB | agr | $NH_3$, $CH_4$ | 0.6 | 0.6 | 0.6 | 0.6 | 0.6 | 0.65 | 0.75 | 0.9 | 1.1 | 1.35 | 1.45 | 1.6 |
| CAMS-GLOB | agr | $PM_{10}$, $PM_{2.5}$, CO, $SO_x$, $NO_x$, NMVOC, $CO_2$ | 0.06 | 0.06 | 0.06 | 0.07 | 0.07 | 0.07 | 0.2 | 0.2 | 0.2 | 1.82 | 1.82 | 1.82 |
| CAMS-GLOB | mma | All | 0.6 | 0.6 | 0.6 | 0.6 | 0.6 | 0.65 | 0.75 | 0.9 | 1.1 | 1.35 | 1.45 | 1.6 |
| CAMS-GLOB | fef | All | 1 | 1 | 1 | 1 | 1 | 1 | 1 | 1 | 1 | 1 | 1 | 1 |
| CAMS-GLOB | slv | All | 0.5 | 0.35 | 0.2 | 0.1 | 0.1 | 0.2 | 0.75 | 1.25 | 1.4 | 1.5 | 1.5 | 1.5 |
| CAMS-GLOB | tnr | All | 1 | 1 | 1 | 1 | 1 | 1 | 1 | 1 | 1 | 1 | 1 | 1 |
| CAMS-GLOB | swd | All | 1 | 1 | 1 | 1 | 1 | 1 | 1 | 1 | 1 | 1 | 1 | 1 |
| CAMS-REG | GNFR_B | All | 0.75 | 0.75 | 0.78 | 0.82 | 0.88 | 0.95 | 1.02 | 1.09 | 1.16 | 1.22 | 1.28 | 1.3 |
| CAMS-REG | GNFR_F4 | NMVOC | 0.48 | 0.36 | 0.29 | 0.25 | 0.21 | 0.25 | 0.4 | 0.74 | 1.01 | 1.38 | 1.57 | 1.71 |
| CAMS-REG | GNFR_K | All | 0.6 | 0.6 | 0.6 | 0.6 | 0.6 | 0.65 | 0.75 | 0.9 | 1.1 | 1.35 | 1.45 | 1.6 |
| CAMS-REG | GNFR_L | $NH_3$, $CH_4$ | 0.6 | 0.6 | 0.6 | 0.6 | 0.6 | 0.65 | 0.75 | 0.9 | 1.1 | 1.35 | 1.45 | 1.6 |
| CAMS-REG | GNFR_L | $PM_{10}$, $PM_{2.5}$, CO, $SO_x$, $NO_x$, NMVOC, $CO_2$ | 0.06 | 0.06 | 0.06 | 0.07 | 0.07 | 0.07 | 0.2 | 0.2 | 0.2 | 1.82 | 1.82 | 1.82 |
| CAMS-REG | GNFR_H | All | 0.28 | 0.13 | 0.1 | 0.08 | 0.09 | 0.09 | 0.44 | 0.91 | 1.18 | 1.43 | 1.6 | 1.61 |
| CAMS-REG | GNFR_D | All | 1 | 1 | 1 | 1 | 1 | 1 | 1 | 1 | 1 | 1 | 1 | 1 |
| CAMS-REG | GNFR_E | All | 0.5 | 0.35 | 0.2 | 0.1 | 0.1 | 0.2 | 0.75 | 1.25 | 1.4 | 1.5 | 1.5 | 1.5 |
| CAMS-REG | GNFR_I | All | 1 | 1 | 1 | 1 | 1 | 1 | 1 | 1 | 1 | 1 | 1 | 1 |
| CAMS-REG | GNFR_J | All | 1 | 1 | 1 | 1 | 1 | 1 | 1 | 1 | 1 | 1 | 1 | 1 |

**Table A4: Fixed hourly temporal profiles per domain, sector and pollutant (Part 2: 13 to 24h). The definition of the sectors can be found in Table 2 and Table 3.**

| Domain | Sector | Pollutant | 13 | 14 | 15 | 16 | 17 | 18 | 19 | 20 | 21 | 22 | 23 | 24 |
|---|---|---|---|---|---|---|---|---|---|---|---|---|---|---|
| CAMS-GLOB | ind | All | 1.22 | 1.24 | 1.25 | 1.16 | 1.08 | 1.01 | 0.95 | 0.9 | 0.85 | 0.81 | 0.78 | 0.75 |
| CAMS-GLOB | agr | $NH_3$, $CH_4$ | 1.65 | 1.75 | 1.7 | 1.55 | 1.35 | 1.1 | 0.9 | 0.75 | 0.65 | 0.6 | 0.6 | 0.6 |
| CAMS-GLOB | agr | $PM_{10}$, $PM_{2.5}$, CO, $SO_x$, $NO_x$, NMVOC, $CO_2$ | 3.39 | 3.39 | 3.39 | 1.68 | 1.68 | 1.68 | 0.56 | 0.56 | 0.56 | 0.22 | 0.22 | 0.22 |
| CAMS-GLOB | mma | All | 1.65 | 1.75 | 1.7 | 1.55 | 1.35 | 1.1 | 0.9 | 0.75 | 0.65 | 0.6 | 0.6 | 0.6 |
| CAMS-GLOB | fef | All | 1 | 1 | 1 | 1 | 1 | 1 | 1 | 1 | 1 | 1 | 1 | 1 |
| CAMS-GLOB | slv | All | 1.5 | 1.5 | 1.5 | 1.5 | 1.5 | 1.4 | 1.25 | 1.1 | 1 | 0.9 | 0.8 | 0.7 |
| CAMS-GLOB | tnr | All | 1 | 1 | 1 | 1 | 1 | 1 | 1 | 1 | 1 | 1 | 1 | 1 |
| CAMS-GLOB | swd | All | 1 | 1 | 1 | 1 | 1 | 1 | 1 | 1 | 1 | 1 | 1 | 1 |
| CAMS-REG | GNFR_B | All | 1.22 | 1.24 | 1.25 | 1.16 | 1.08 | 1.01 | 0.95 | 0.9 | 0.85 | 0.81 | 0.78 | 0.75 |
| CAMS-REG | GNFR_F4 | NMVOC | 1.8 | 1.86 | 1.86 | 1.79 | 1.6 | 1.43 | 1.24 | 1.03 | 0.86 | 0.7 | 0.61 | 0.57 |
| CAMS-REG | GNFR_K | All | 1.65 | 1.75 | 1.7 | 1.55 | 1.35 | 1.1 | 0.9 | 0.75 | 0.65 | 0.6 | 0.6 | 0.6 |
| CAMS-REG | GNFR_L | $NH_3$, $CH_4$ | 1.65 | 1.75 | 1.7 | 1.55 | 1.35 | 1.1 | 0.9 | 0.75 | 0.65 | 0.6 | 0.6 | 0.6 |
| CAMS-REG | GNFR_L | $PM_{10}$, $PM_{2.5}$, CO, $SO_x$, $NO_x$, NMVOC, $CO_2$ | 3.39 | 3.39 | 3.39 | 1.68 | 1.68 | 1.68 | 0.56 | 0.56 | 0.56 | 0.22 | 0.22 | 0.22 |
| CAMS-REG | GNFR_H | All | 1.66 | 1.58 | 1.47 | 1.43 | 1.42 | 1.46 | 1.46 | 1.4 | 1.39 | 1.25 | 0.96 | 0.58 |
| CAMS-REG | GNFR_D | All | 1 | 1 | 1 | 1 | 1 | 1 | 1 | 1 | 1 | 1 | 1 | 1 |
| CAMS-REG | GNFR_E | All | 1.5 | 1.5 | 1.5 | 1.5 | 1.5 | 1.4 | 1.25 | 1.1 | 1 | 0.9 | 0.8 | 0.7 |
| CAMS-REG | GNFR_I | All | 1 | 1 | 1 | 1 | 1 | 1 | 1 | 1 | 1 | 1 | 1 | 1 |
| CAMS-REG | GNFR_J | All | 1 | 1 | 1 | 1 | 1 | 1 | 1 | 1 | 1 | 1 | 1 | 1 |


## 6    Authors contribution

Marc Guevara conceived and coordinated the development of the CAMS-TEMPO profiles. Carles Tena helped constructing the CAMS-TEMPO data files. Nellie Elguindi and Claire Granier provided feedback for the construction of the global profiles and its combination with the CAMS-GLOB-ANT inventory. Hugo Denier van der Gon and Jeroen Kuenen provided feedback for the construction of the European regional profiles and its combination with the CAMS-REG-AP/GHG inventory. Sabine Darras processed and prepared the CAMS-TEMPO data files to make them available through the ECCAD system. Oriol Jorba and Carlos Pérez García-Pando helped conceiving the CAMS-TEMPO dataset and supervised the work. Marc Guevara prepared the manuscript with contributions from all co-authors.

## 7    Acknowledgements

The present work was funded through the CAMS_81 (CAMS global and regional emissions) contract, coordinated by the Centre National de la Recherche Scientifique (CNRS, Claire Granier). The Copernicus Atmosphere Monitoring Service (CAMS, https://atmosphere.copernicus.eu/) is operated by the European Centre for Medium-Range Weather Forecasts on behalf of the European Commission as part of the Copernicus Programme. The study has also received support from the Ministerio de Ciencia, Innovación y Universidades (MICINN) as part of the BROWNING project RTI2018-099894-B-I00 and NUTRIENT project CGL2017-88911-R and from the Agencia Estatal de Investigacion (AEI) as part of the VITALISE project (PID2019-108086RA-I00 / AEI /10.13039/501100011033). Carlos Pérez García-Pando acknowledges the European Research Council (grant no. 773051, FRAGMENT), the long-term support from the AXA Research Fund, as well as the support received through the Ramón y Cajal programme (grant RYC-2015-18690) of the MICINN. We acknowledge PRACE and RES for awarding access to MareNostrum at Barcelona Supercomputing Center. The authors are thankful to the Spanish Research Centre for Energy, Environment and Technology (CIEMAT) for sharing the databases of power plant emissions.

## 8    Competing interests

The authors declare that they have no conflict of interest.

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

| Sector | Dataset | Spatial coverage | Temporal resolution | Spatial resolution | Yearly/Meteorology dependency |
|---|---|---|---|---|---|
| energy industry | this work | Global, EU | monthly, weekly, hourly | Country-dependent | no |
| | TNO | EU | monthly, weekly, hourly | Fixed | no |
| | EMEP | EU | monthly, weekly, hourly | Country-dependent | no |
| | EDGARv4.3.2 | Global | monthly | 3 geo-regions | no |
| | EDGARv5 | Global | monthly, weekly, hourly | Country-dependent | yes (monthly profiles) |
| residential combustion | this work | Global, EU | monthly, weekly, daily, hourly | Grid cell level (monthly, daily profiles) | yes (monthly, daily profiles) |
| | TNO | EU | monthly, weekly, hourly | Fixed | no |
| | EMEP | EU | monthly, weekly, hourly | Country-dependent | no |
| | EDGARv4.3.2 | Global | monthly | 3 geo-regions | no |
| | EDGARv5 | Global | monthly, weekly, hourly | Country-dependent | yes (monthly profiles) |
| manufacturing industry | this work | Global, EU | monthly, weekly, hourly | Country-dependent (monthly profiles) | no |
| | TNO | EU | monthly, weekly, hourly | Fixed | no |
| | EMEP | EU | monthly, weekly, hourly | Country-dependent | no |
| | EDGARv4.3.2 | Global | monthly | Fixed | no |
| | EDGARv5 | Global | monthly, weekly, hourly | 23 world regions | no |
| road transport | this work | Global, EU | monthly, weekly, hourly | Grid cell level | yes (EU monthly profiles) |
| | TNO | EU | monthly, weekly, hourly | Fixed | no |
| | EMEP | EU | monthly, weekly, hourly | Country-dependent | no |
| | EDGARv4.3.2 | Global | monthly | Fixed | no |
| | EDGARv5 | Global | monthly, weekly, hourly | 23 world regions | no |
| aviation | this work | EU | monthly, weekly, hourly | Country-dependent (monthly profiles) | no |
| | TNO | EU | monthly, weekly, hourly | Fixed | no |
| | EMEP | EU | monthly, weekly, hourly | Country-dependent | no |
| | EDGARv4.3.2 | Global | monthly | Fixed | no |
| | EDGARv5 | Global | monthly, weekly, hourly | Fixed | no |
| agriculture (livestock) | this work | Global, EU | monthly, weekly, daily, hourly | Grid cell level (monthly, daily profiles) | yes (monthly, daily profiles) |
| | TNO | EU | monthly, weekly, hourly | Fixed | no |
| | EMEP | EU | monthly, weekly, hourly | Country-dependent | no |
| | EDGARv4.3.2 | Global | monthly | 3 geo-regions | no |

| | | | | | |
|---|---|---|---|---|---|
| | EDGARv5 | Global | monthly, weekly, hourly | Fixed | no |
| | this work | Global, EU | monthly, weekly, daily, hourly | Grid cell level (monthly, daily profiles) | yes (monthly, daily profiles) |
| agriculture (fertilizers) | TNO | EU | monthly, weekly, hourly | Fixed | no |
| | EMEP | EU | monthly, weekly, hourly | Country-dependent | no |
| | EDGARv4.3.2 | Global | monthly | 3 geo-regions | no |
| | EDGARv5 | Global | monthly, weekly, hourly | Fixed | no |
| | this work | Global, EU | monthly, weekly, hourly | Grid cell level (monthly profiles) | no |
| agriculture (agricultural waste burning) | TNO | EU | monthly, weekly, hourly | Fixed | no |
| | EMEP | EU | monthly, weekly, hourly | Country-dependent | no |
| | EDGARv4.3.2 | Global | monthly | 3 geo-regions | no |
| | EDGARv5 | Global | monthly, weekly, hourly | 23 world regions | no |

Table 2: Main characteristics of the CAMS global temporal profiles (CAMS-GLOB-TEMPO) reported by sector and temporal resolution (monthly, daily, weekly, hourly). The text between brackets gives information on the spatial resolution and pollutant and yearly dependency of each profile. Gridded: indicates that the profile varies per grid cell within a country; per country: indicates that the profile varies only per country; fixed: indicates that the profile is spatially invariant, year-independent: indicates that the profiles does not vary per year; year-dependent: indicates that the profile varies per year; pollutant-independent: indicates that the same profile is proposed for all pollutants ($NO_x$, $SO_x$, NMVOC, $NH_3$, CO, $PM_{10}$, $PM_{2.5}$, $CO_2$ and $CH_4$); pollutant-dependent: indicates that the profile varies per pollutant; per day type: indicates that the profile varies as a function of the day (weekday, Saturday and Sunday). The symbol "-" denotes that no profile is proposed.

| Sector | Description | Monthly (∑=12) | Daily (∑=365/366) [1] | Weekly (∑=7) | Hourly (∑=24) |
|---|---|---|---|---|---|
| ene | Power generation (power and heat plants, refineries, others) | (per country, year-independent, pollutant-dependent) | - | (per country, year-independent, pollutant-dependent) | (per country, year-independent, pollutant-dependent) |
| ind | Industrial process | (per country, year-independent, pollutant-independent) | - | (fixed, year-independent, pollutant-independent) [2] | (fixed, year-independent, pollutant-independent) [2] |
| res | Other stationary combustion | (gridded, year-dependent, pollutant-independent) | (gridded, year-dependent, pollutant-independent) | (fixed, year-independent, pollutant-independent) [2] | (gridded, year-independent, pollutant-dependent) |
| fef | Fugitives | (fixed, year-independent, pollutant-independent) [2] | - | (fixed, year-independent, pollutant-independent) [2] | (fixed, year-independent, pollutant-independent) [2] |
| slv | Solvents | (fixed, year-independent, pollutant-independent) [2] | - | (fixed, year-independent, pollutant-independent) [2] | (fixed, year-independent, pollutant-independent) [2] |
| tro | Road transportation | (gridded, year-independent, pollutant-independent) | - | (gridded, year-independent, pollutant-independent) | (year-independent, pollutant-independent, per day type) |
| shp | Ships | (fixed, year-independent, pollutant-independent) [2] | - | (fixed, year-independent, pollutant-independent) [2] | (fixed, year-independent, pollutant-independent) [2] |

| | | | | | |
|---|---|---|---|---|---|
| tnr | Off road transportation | (fixed year-independent, pollutant-independent) [2] | - | (fixed year-independent, pollutant-independent) [2] | (fixed year-independent, pollutant-independent) [2] |
| swd | Solid waste and waste water | (fixed year-independent, pollutant-independent) [2] | - | (fixed year-independent, pollutant-independent) [2] | (fixed year-independent, pollutant-independent) [2] |
| mma | Agriculture (livestock) | $NH_3$ and $NO_x$ (gridded, year-dependent) Others (fixed, year-independent) | $NH_3$ and $NO_x$ (gridded, year-dependent) | (fixed, year-independent, pollutant-independent) [2] | (fixed, year-independent, pollutant-independent) [2] |
| agr | Agriculture (fertilizers and agricultural waste burning) | $NH_3$ (gridded, year-dependent) Others (gridded, year-independent) | $NH_3$ (gridded, year-dependent) | (fixed, year-independent, pollutant-independent) [2] | (fixed, year-independent, pollutant-dependent) |

[1] Leap or non-leap years
[2] Same profiles as the ones reported by the TNO dataset (Denier van der Gon et al., 2011)

**Table 3: Main characteristics of the CAMS regional temporal profiles (CAMS-REG-TEMPO) reported by sector and temporal resolution (monthly, daily, weekly, hourly). The text between brackets gives information on the spatial resolution and pollutant and yearly dependency of each profile. Gridded: indicates that the profile varies per grid cell within a country; per country: indicates that the profile varies only per country; fixed: indicates that the profile is spatially invariant, year-independent: indicates that the profiles does not vary per year; year-dependent: indicates that the profile varies per year; pollutant-independent: indicates that the same profile is proposed for all pollutants ($NO_x$, $SO_x$, NMVOC, $NH_3$, CO, $PM_{10}$, $PM_{2.5}$, $CO_2$ and $CH_4$); pollutant-dependent: indicates that the profile varies per pollutant; per day type: indicates that the profile varies as a function of the day (weekday, Saturday and Sunday). The symbol "-" denotes that no profile is proposed.**

| Sector | Description | Monthly ($\sum$=12) | Daily ($\sum$=365/366) [1] | Weekly ($\sum$=7) | Hourly ($\sum$=24) |
|---|---|---|---|---|---|
| GNFR_A | Public Power (power and heat plants) | (per country, year-independent, pollutant-dependent) | - | (per country, year-independent, pollutant-dependent) | (per country, year-independent, pollutant-dependent) |
| GNFR_B | Industry | (per country, year-independent, pollutant-independent) | - | (fixed, year-independent, pollutant-independent) [2] | (fixed, year-independent, pollutant-independent) [2] |
| GNFR_C | Other stationary combustion | (gridded, year-dependent, pollutant-independent) | (gridded, year-dependent, pollutant-independent) | (fixed, year-independent, pollutant-independent) [2] | (gridded, year-independent, pollutant-dependent) |
| GNFR_D | Fugitive | (fixed, year-independent, pollutant-independent) [2] | - | (fixed, year-independent, pollutant-independent) [2] | (fixed, year-independent, pollutant-independent) [2] |
| GNFR_E | Solvents | (fixed, year-independent, pollutant-independent) [2] | - | (fixed, year-independent, pollutant-independent) [2] | (fixed, year-independent, pollutant-independent) [2] |
| GNFR_F1 | Road transport exhaust gasoline | CO and NMVOC (gridded, year-dependent) Others (gridded, year-independent) | - | (gridded, year-independent, pollutant-independent) | (gridded, year-independent, pollutant-independent, day type) |
| GNFR_2 | Road transport exhaust diesel | $NO_x$ (gridded, year-dependent) Others (gridded, year-independent) | - | (gridded, year-independent, pollutant-independent) | (gridded, year-independent, pollutant-independent, day type) |
| GNFR_F3 | Road transport exhaust LPG | (gridded, year-independent, pollutant-independent) | - | (gridded, year-independent, pollutant-independent) | (gridded, year-independent, pollutant-independent, day type) |

| GNFR | Sector | | | | |
|---|---|---|---|---|---|
| GNFR_F4 | Road transport non-exhaust (wear and evaporative) | PM (gridded, year-independent) NMVOC (fixed, year-independent) | - | PM (gridded, year-independent) NMVOC (fixed, year-independent) | PM (gridded, year-independent) NMVOC (fixed, year-independent) |
| GNFR_G | Shipping | (fixed, year-independent, pollutant-independent) [2] | - | (fixed, year-independent, pollutant-independent) [2] | (fixed, year-independent, pollutant-independent) [2] |
| GNFR_H | Aviation | (per country, year-independent, pollutant-independent) | - | (fixed, year-independent, pollutant-independent) [2] | (fixed, year-independent, pollutant-independent) |
| GNFR_I | Off road transport | (fixed, year-independent, pollutant-independent) [2] | - | (fixed, year-independent, pollutant-independent) [2] | (fixed, year-independent, pollutant-independent) [2] |
| GNFR_I | Waste management | (fixed, year-independent, pollutant-independent) [2] | - | (fixed, year-independent, pollutant-independent) [2] | (fixed, year-independent, pollutant-independent) [2] |
| GNFR_K | Agriculture (livestock) | $NH_3$ and $NO_x$ (gridded, year-dependent) Others (fixed, year-independent) | $NH_3$ and $NO_x$ (gridded, year-dependent) | (fixed, year-independent, pollutant-independent) [2] | (fixed, year-independent, pollutant-independent) [2] |
| GNFR_L | Agriculture (fertilizers, agricultural waste burning) | $NH_3$ (gridded, year-dependent) Others (gridded, year-independent) | $NH_3$ (gridded, year-dependent) | (fixed, year-independent, pollutant-independent) [2] | (fixed, year-independent, pollutant-dependent) |

[1] Leap or non-leap years

[2] Same profiles as the ones reported by the TNO dataset (Denier van der Gon et al., 2011)


**Table 4: Emission factors [kg·TJ⁻¹] related to the energy industry per fuel type and pollutant. Values obtained from the EMEP/EEA 2016 emission inventory guidebook for the priority air pollutants (1.A.1 Energy industries, Table 3-2, 3-3, 3-4, 3-5 and 3-7) and from the IPPC guidelines (Volume 2: Energy, Table 2.2) for greenhouse gases.**

| Fuel / EF [kg·TJ⁻¹] | $NO_x$ | $SO_x$ | CO | NMVOC | $PM_{10}$ | $PM_{2.5}$ | $CO_2$ | $CH_4$ |
|---|---|---|---|---|---|---|---|---|
| Hard Coal | 209 | 820 | 8.7 | 1.0 | 7.7 | 3.4 | 98,300 | 1 |
| Brown Coal (Lignite) | 247 | 1680 | 8.7 | 1.4 | 7.9 | 3.2 | 101,000 | 1 |
| Gaseous fuels (natural gas) | 89 | 0.281 | 39 | 2.6 | 0.89 | 0.89 | 56,100 | 1 |
| Heavy Fuel Oil | 142 | 495 | 15.1 | 2.3 | 25.2 | 19.3 | 77,400 | 1 |
| Biomass | 81 | 10.8 | 90 | 7.31 | 155 | 133 | 112,000 | 30 |


**Table 5: Share of final energy consumption in the residential sector by fuel and type of end-use in Europe (Eurostat, 2018)**

| Fuel | Space heating | Water heating | Cooking | Other end uses |
|---|---|---|---|---|
| Gas | 77.7% | 17.2% | 5.1% | - |
| Solid fuels (coal products) | 90.7% | 8.2% | 1.1% | - |
| Petroleum products (LPG, fuel oil) | 81.1% | 12.9% | 6.0% | 0.1% |
| Renewables and wastes (solid biofuels) | 89.1% | 8.8% | 1.7% | 0.4% |
| **All** | **81.8%** | **13.9%** | **4.2%** | **0.1%** |


**Table 6: List of traffic activity datasets and corresponding sources of information compiled**

| Dataset | Description | Source of Information |
|---|---|---|
| Paris city | Hourly traffic counts registered at the permanent stations for the years 2013 - 2017 | Paris data (2018) |
| Madrid city | Hourly traffic counts registered at the permanent stations for the years 2013 - 2017 | Madrid data (2018) |
| Barcelona city | Hourly traffic counts registered at the permanent stations for the year 2015 | Courtesy from Barcelona city council |
| Valencia city | Hourly traffic counts registered at the permanent stations for the year 2009 | Courtesy from Valencia city council |
| Milano city | Hourly traffic counts registered at the permanent stations for the year 2013 | Milano data (2018) |
| Utrecht city | Hourly traffic counts registered at the permanent stations for the years 2014-2015 | Utrecht data (2018) |
| Copenhagen city | Hourly traffic counts registered at the permanent stations for the year 2014 | Copenhagen data (2018) |
| Oslo city | Hourly traffic counts registered at the permanent stations for the years 2008, 2010 and 2012 | Courtesy from Norwegian Road Administration |
| London city | Hourly traffic counts registered at the station in Marylebone Road for the years 2013 to 2016 | Courtesy from Transport for London |
| Berlin city | Monthly and hourly profiles derived from 2014 traffic counts in the main road network | VLB (2018) |
| Greater Athens | Monthly, weekly and hourly profiles derived from 2006-2010 traffic counts in the main road network | Fameli and Assimakopoulos (2015) |
| New York City | Hourly traffic counts registered at portable stations for the years 2014 - 2018 | New York City data (2019) |
| Mexico City | Hourly traffic counts registered at permanent stations for the year 2015 | Courtesy from Secretary of Environment of Mexico City |
| City of Melbourne | Hourly traffic counts registered at portable stations for the years 2014 - 2017 | Melbourne data (2019) |
| Beijing | Hourly profile derived from the traffic flows reported by the Intelligent Traffic System of Beijing | Cai and Xie (2011) |
| Guangzhou | Weekly and hourly profiles derived from the field investigations in Guangzhou urban/rural areas | Zheng et al. (2009) |
| UK | Average daily traffic flows by month, day and hour in the UK road network (2012-2016 average) | GovUK (2018) |
| Germany | Hourly traffic counts registered at the highways and federal highways stations for the year 2016 | BASt (2018) |
| Spain | Hourly traffic counts registered at the national transport network for the year 2009 | Courtesy from Spanish Ministry Transport |
| New South Wales (Australia) | Hourly traffic counts registered at permanent and sample stations for the years 2015 to 2017 | TfNSW (2019) |
| California | Monthly, weekly and hourly profiles derived from weigh-in-motion traffic counts from 2010 | McDonald et al. (2014) |

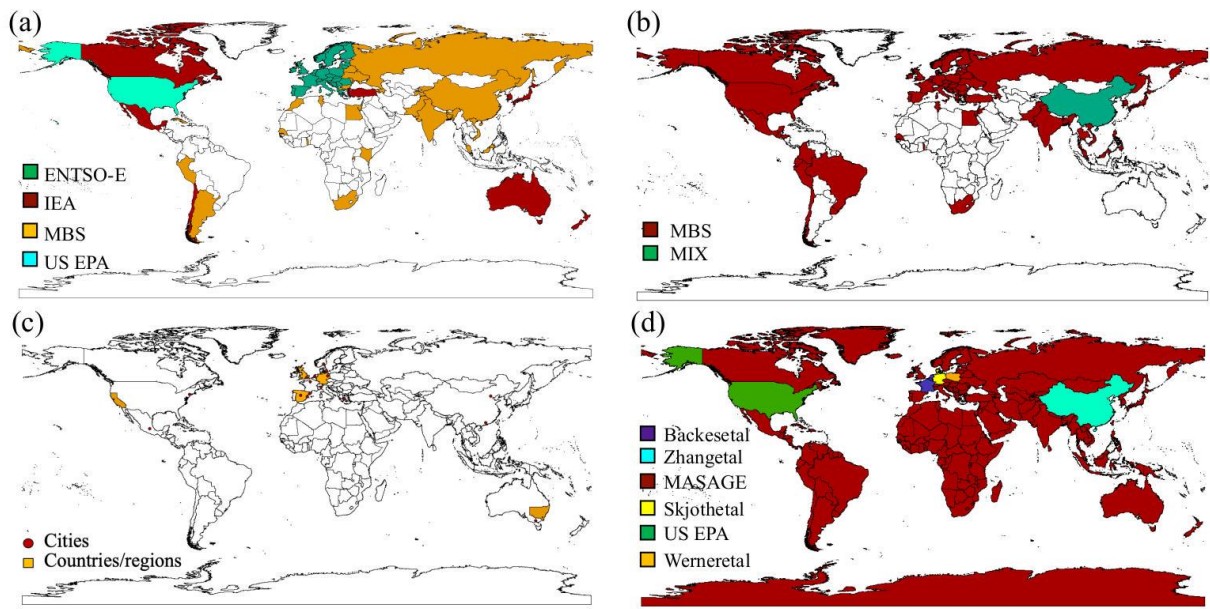


**Figure 1: Representation of the spatial coverage of the datasets used to derive temporal profiles for energy industry (a), manufacturing industry (b), road transport (c) and agriculture (use of fertilizers) (d). For energy industry, the legend indicates the different sources of information used: The European Network of Transmission System Operators for Electricity (ENTSO-E), the United States Environmental Protection Agency (US EPA), The International Energy Agency (IEA) and the Monthly Bulletin of**
**Statistics (MBS). For manufacturing industry, the legend indicates the sources of information used: the MBS and the MIX inventory (Li et al., 2017). For agriculture, sources of information are also highlighted: Backes et al. (2016), Zhang et al., (2018), the MASAGE inventory (Paulot et al., 2014), US EPA (2019), Skjoth et al. (2011), and Werner et al. (2015). Administrative boundaries are derived from GADM (2020).**


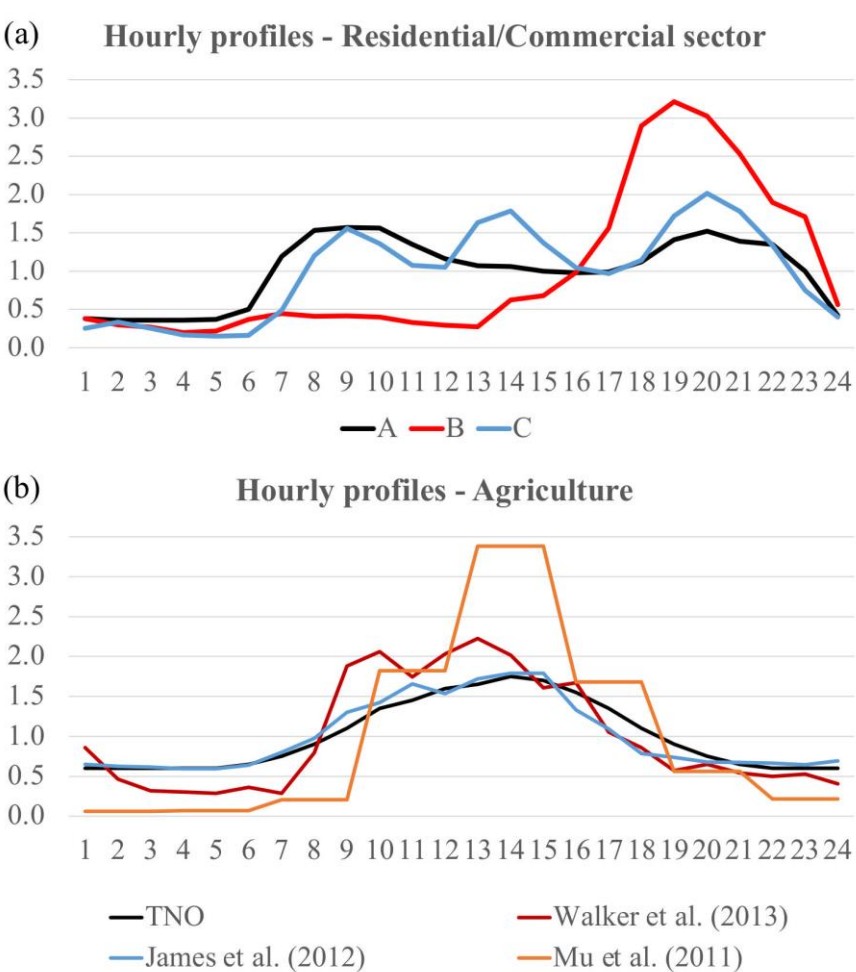

**Figure 2: (a) Proposed hourly temporal profiles for the residential/commercial combustion sector where profile A refers to $NO_x$, $SO_x$ and $CH_4$ emissions in urban/rural areas of developed and developing countries, profile B refers to $PM_{10}$, $PM_{2.5}$, CO, $CO_2$, NMVOC and $NH_3$ in urban/rural areas of developed countries, and profile C refers to all pollutants in rural areas of developing countries. (b) Comparison between the hourly temporal profile proposed by TNO for agricultural emissions (Denier van der Gon et al., 2011) and the three measurement-based temporal profiles reported by Walker et al. (2013) derived from $NH_3$ flux measurements performed in a fertilized corn canopy in North Carolina, James et al. (2012) based on direct $NH_3$ measurements performed in a mechanically ventilated swine barn, and Mu et al. (2011) derived from active fire satellite observations.**

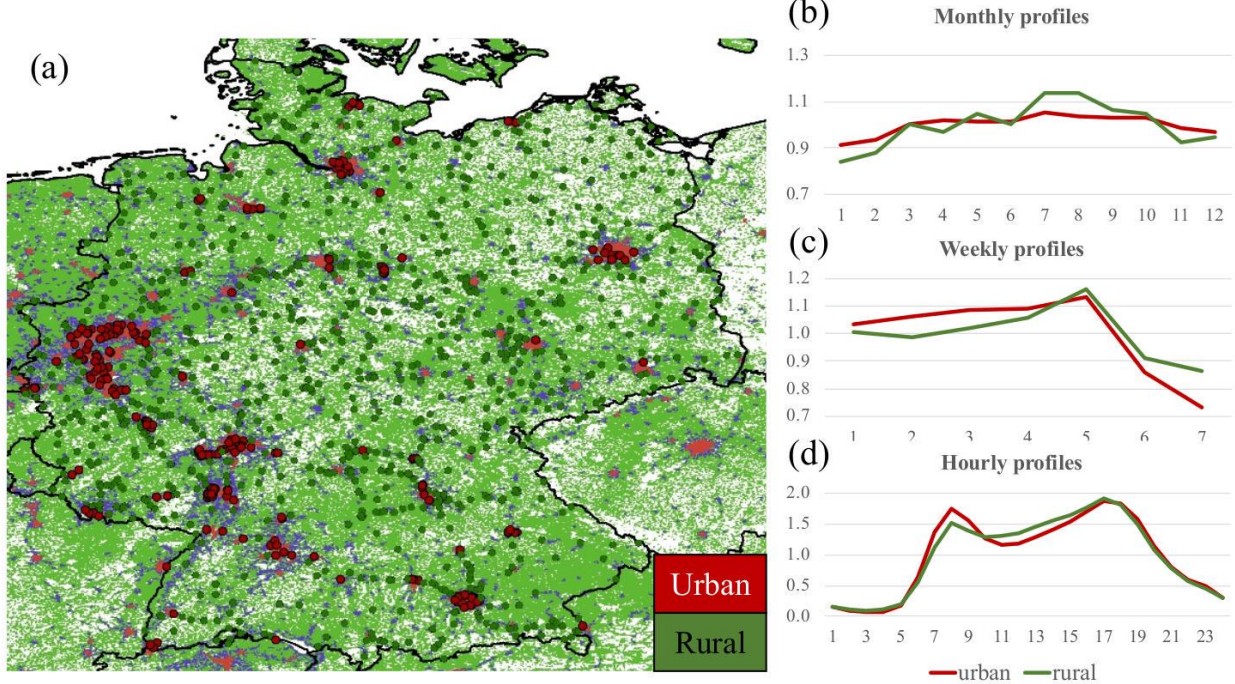

**Figure 3: Map representing urban and rural human settlements in Germany as reported by the Global Human Settlement Layer (GHSL; Pesaresi et al., 2019) and the location of urban and rural German traffic count stations (a) and monthly, weekly and hourly temporal profiles derived from averaging the original traffic counts (BASt, 2018) by type of station (b, c and d). Administrative boundaries are derived from GADM (2020).**


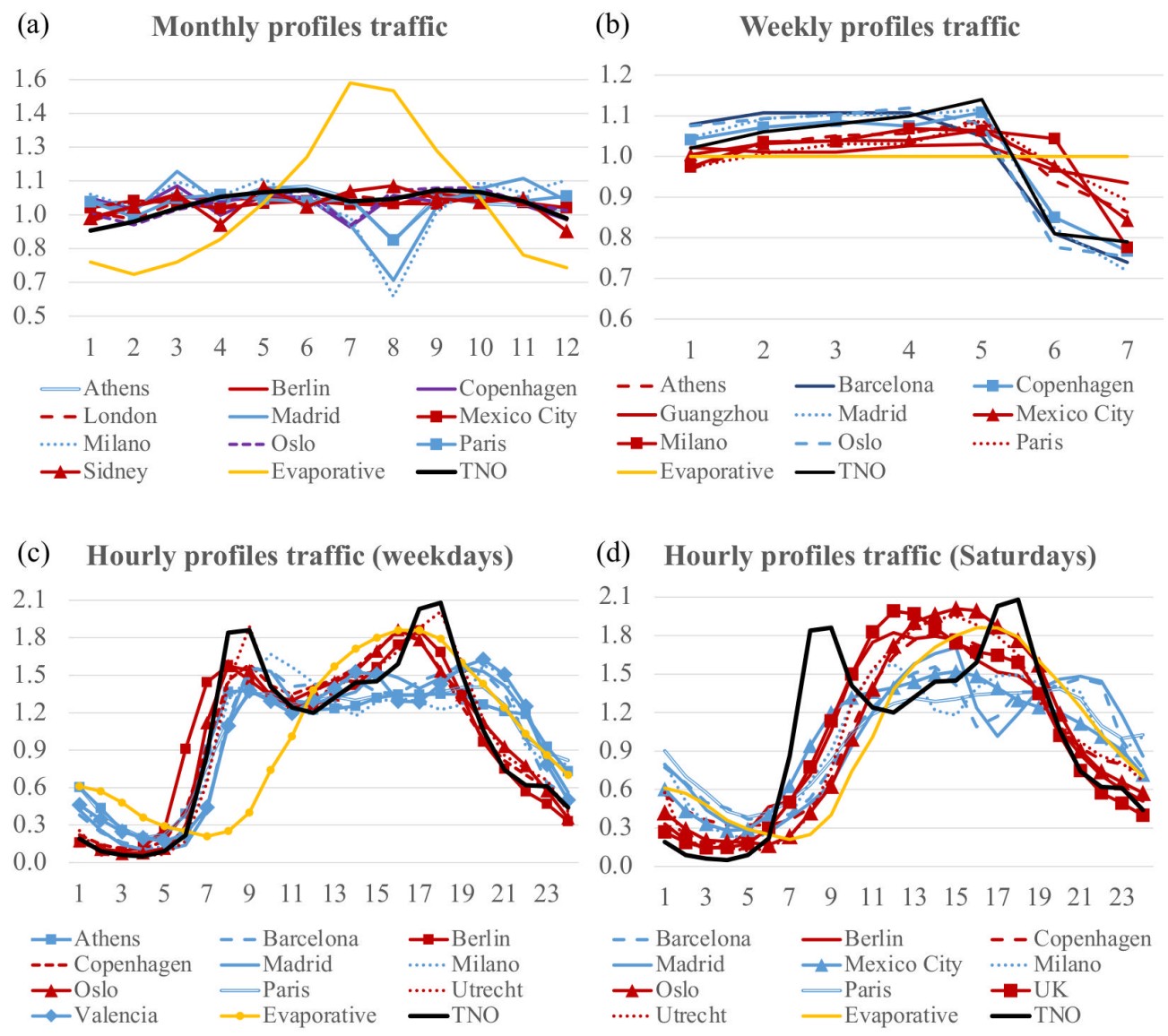

**Figure 4: Monthly (a), weekly (b) and hourly (c, for weekdays, d for Saturdays) temporal profiles derived from measured traffic counts in selected cities (see Table 6 for references). The profile reported by the TNO dataset is plotted for comparison purposes (Denier van der Gon et al., 2011). The monthly, weekly and hourly profiles proposed for the gasoline evaporative emissions (Evaporative, yellow line) is also shown.**


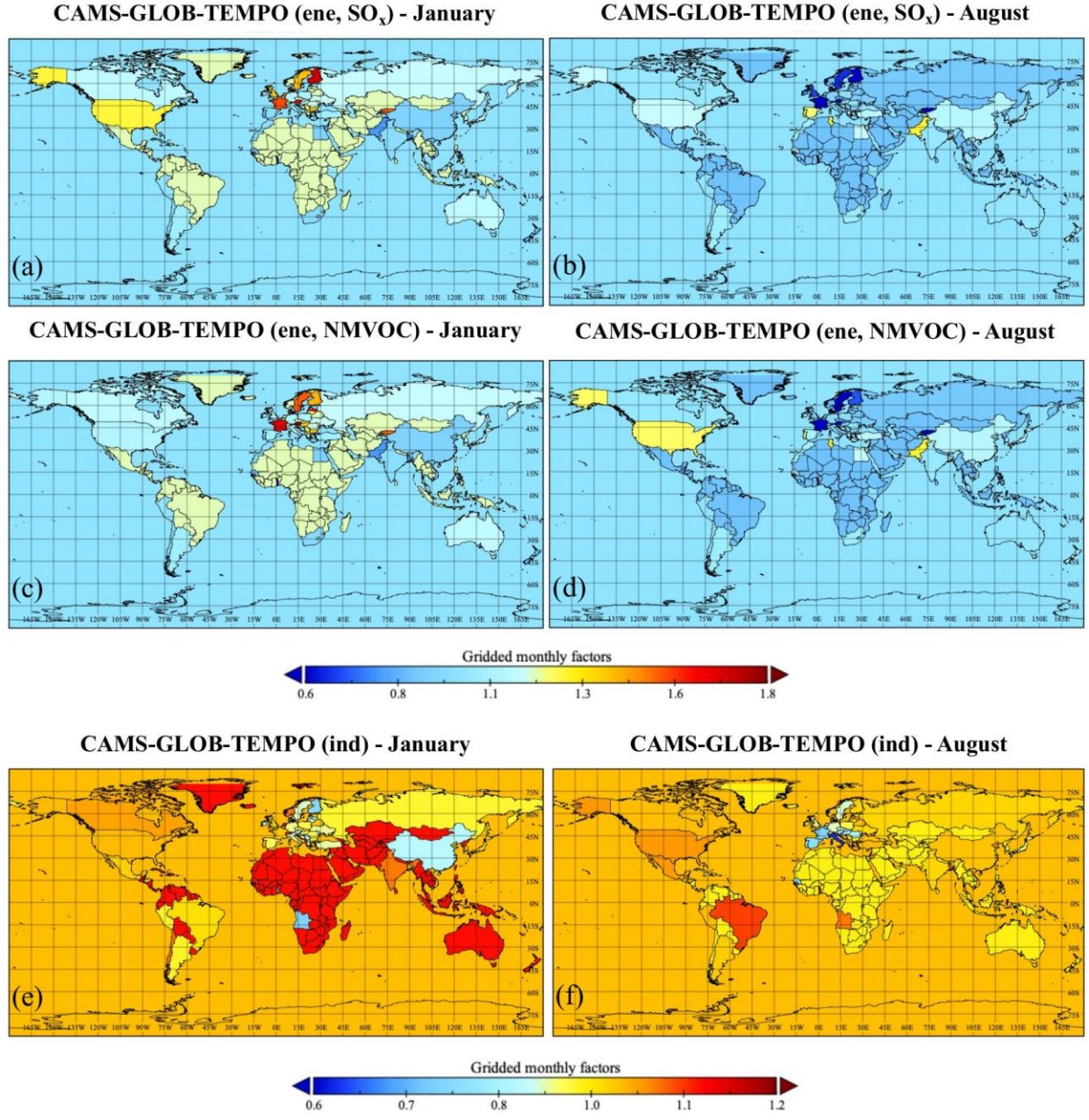

**Figure 5: CAMS-GLOB-TEMPO (0.1x0.1 deg) monthly scale factor maps for SO$_x$ (a, b) and NMVOC (c, d) energy industry (ene) emissions and for manufacturing industry (ind) emissions (e, f) for the months of January and August. Administrative boundaries are derived from the Micro World Data Bank (MWDB2, 2011).**


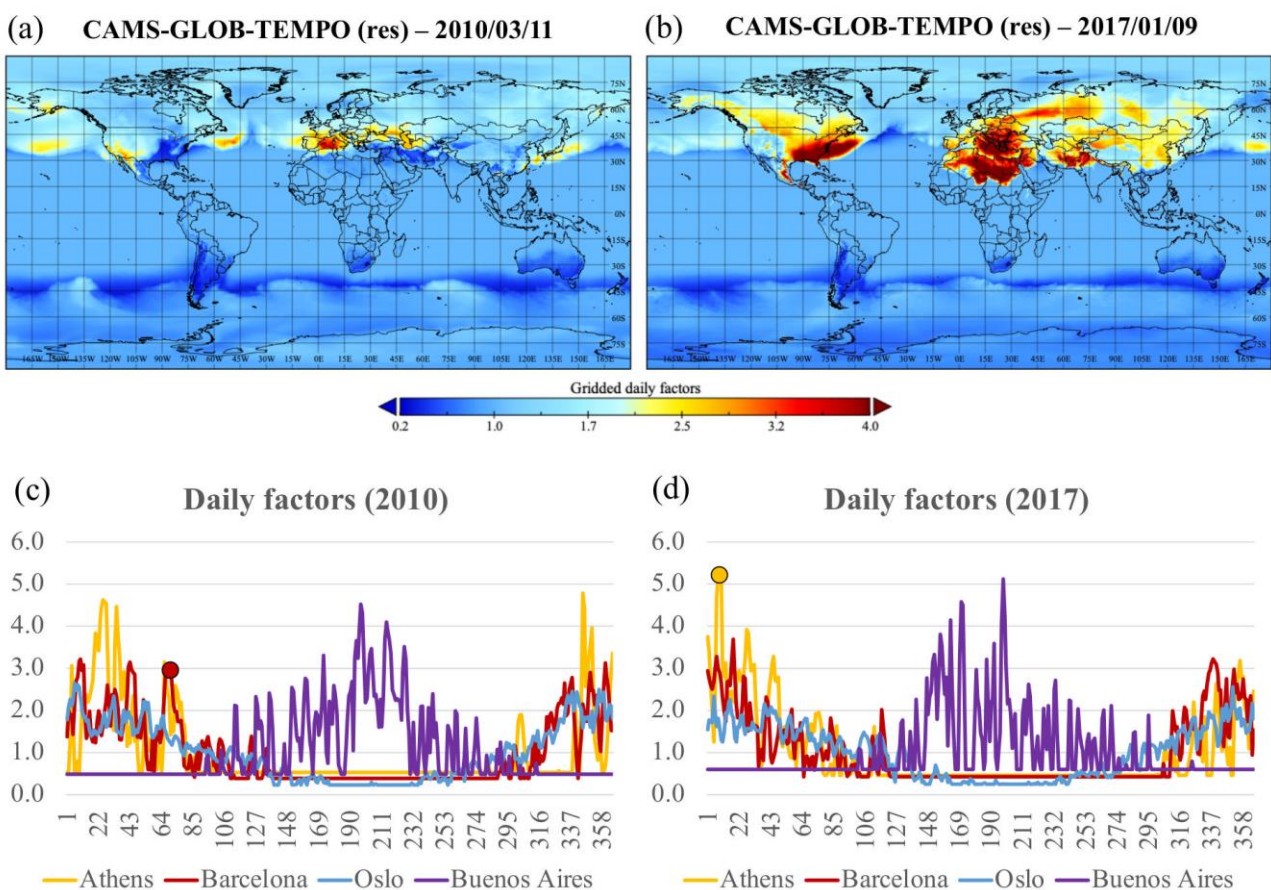

Figure 6 CAMS-GLOB-TEMPO (0.1x0.1 deg) daily scale factor maps for residential/commercial (res) emissions for 2010/03/11 (a) and 2017/01/09 (b). Daily factors obtained over the cities of Athens, Barcelona, Buenos Aires and Oslo for 2010 (c) and 2017 (d). The red and yellow circles on the plots indicate the cold outbreaks experienced in Barcelona (2010/03/11) and Athens (2017/01/09), respectively. Administrative boundaries are derived from the Micro World Data Bank (MWDB2, 2011).

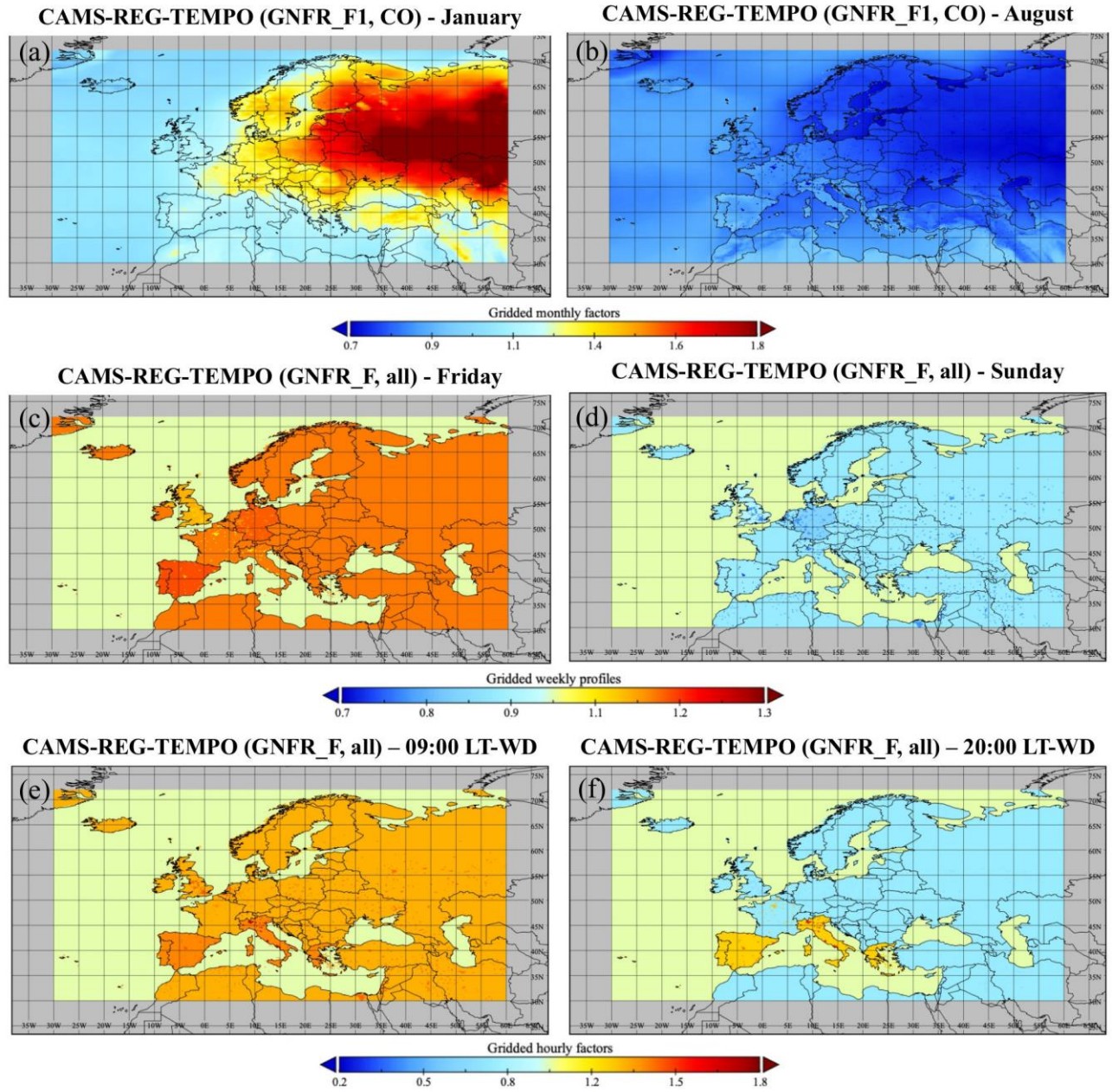

**Figure 7: CAMS-REG-TEMPO (0.1x0.05 deg) monthly (a, b), weekly (c, d) and hourly (e, f) scale factor maps. Monthly factors correspond to the meteorological dependent profiles computed for CO exhaust gasoline emissions (GNFR_F1) for January and August (climatology of 2010-2017). Weekly factors are represented for Friday and Sunday. Hourly factors are represented for weekdays (WD) at 09:00 and 20:00 local time (LT). Administrative boundaries are derived from the Micro World Data Bank (MWDB2, 2011).**


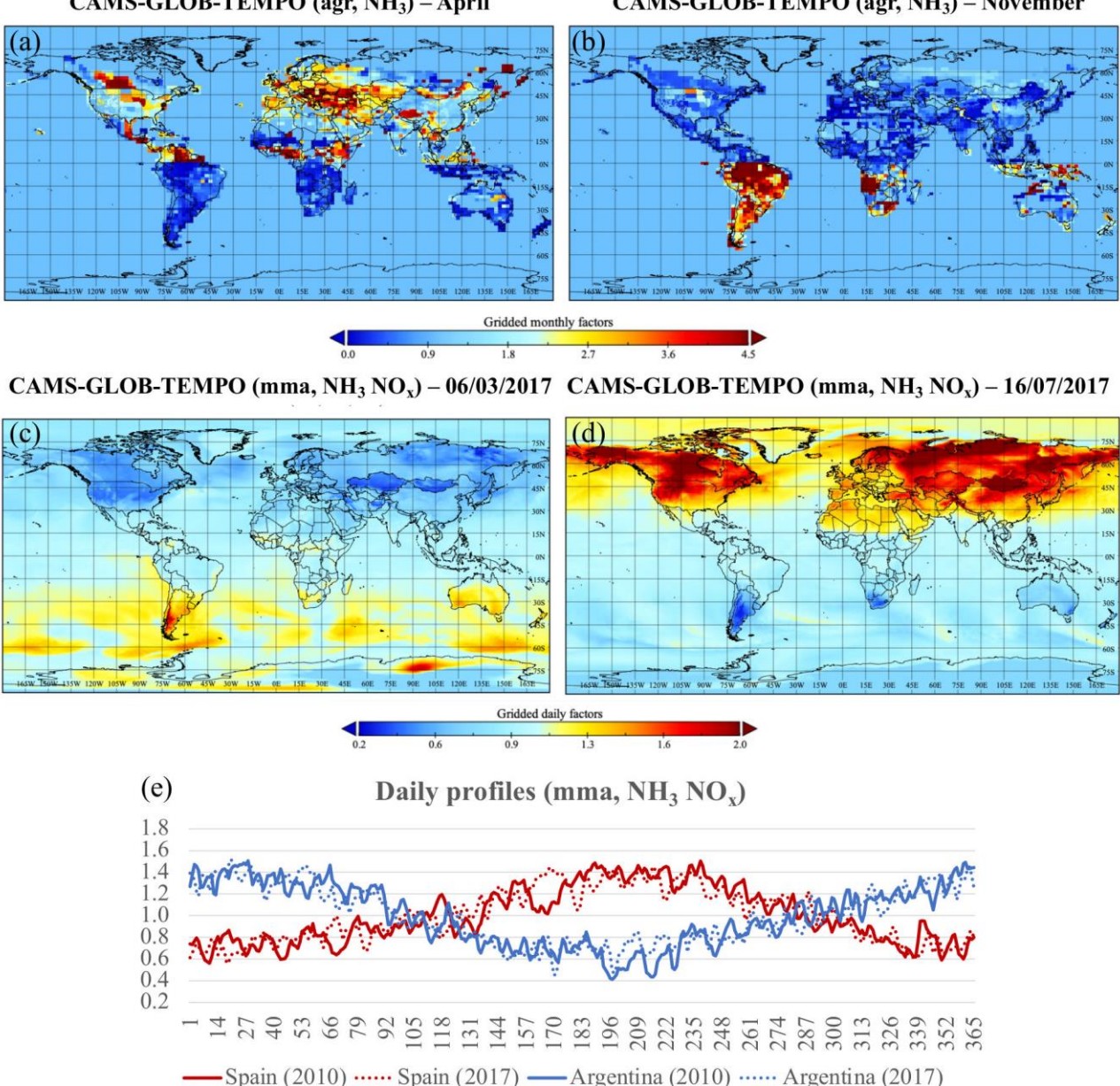

**Figure 8: CAMS-GLOB-TEMPO (0.1x0.1 deg) monthly (a, b) and daily (c, d) scale factor maps for agriculture (agr) NH3 emissions and livestock (mma) NH3 and NOx emissions for April and November and 2017/03/06 and 2017/07/16, respectively. Daily factors for livestock emissions (mma) obtained over Spain and Argentina for 2010 and 2017 (e). Administrative boundaries are derived from the Micro World Data Bank (MWDB2, 2011).**


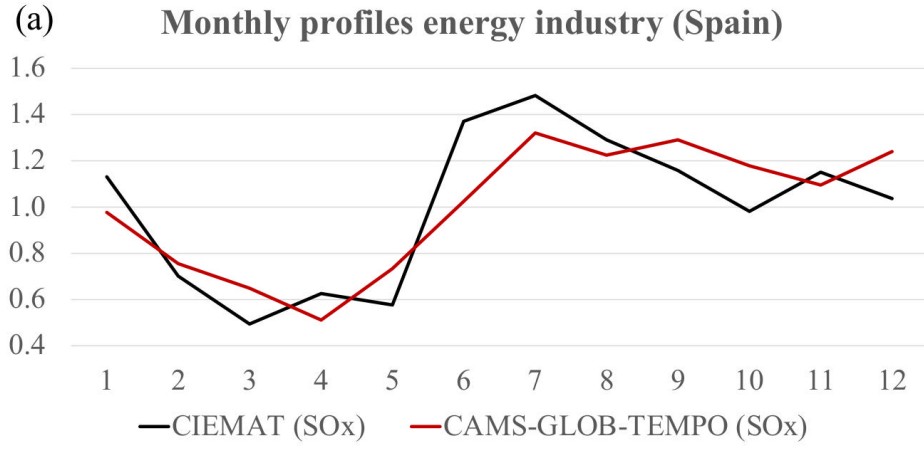

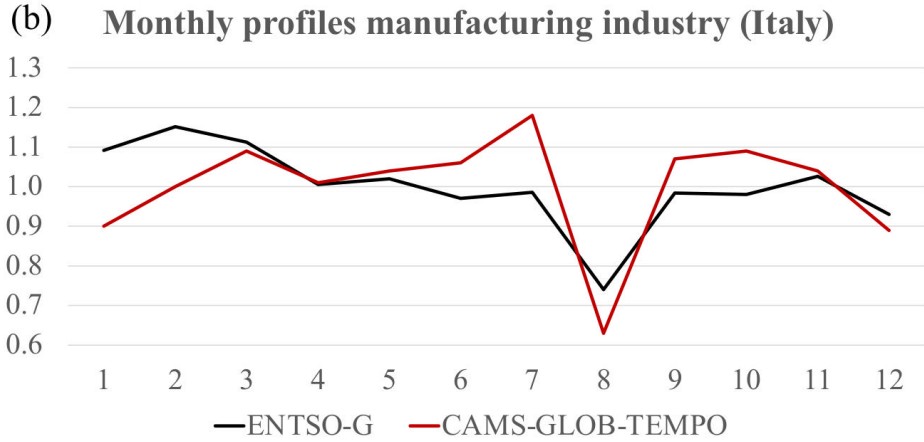

Figure 9: (a) Comparison between monthly temporal factors for $SO_x$ energy industry emissions derived from power plant emission measurements (CIEMAT, personal communication) and reported by CAMS-GLOB-TEMPO for Spain. (b) Comparison between monthly temporal factors for manufacturing industry emissions derived from real-world measurements of industrial natural gas consumption (ENTSOG, 2020) and reported by CAMS-GLOB-TEMPO for Italy.

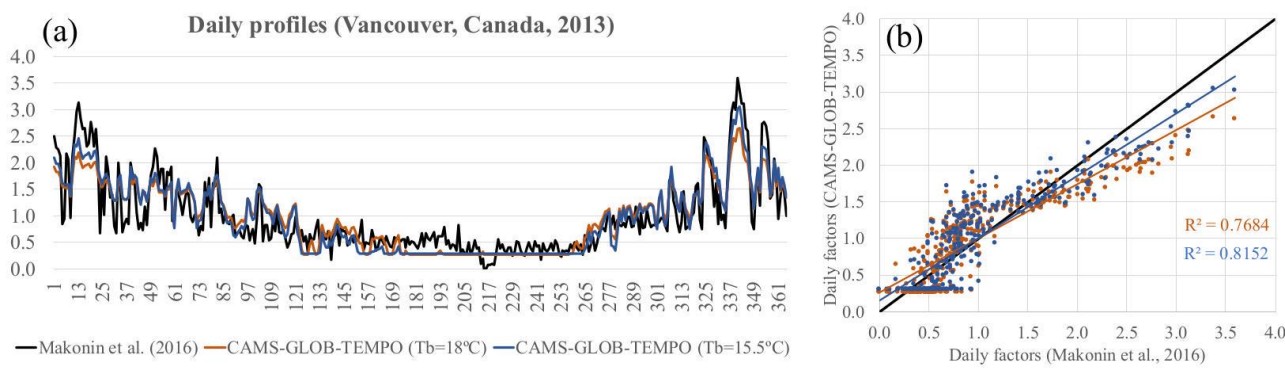

**Figure 10: (a)** Comparison between daily temporal factors obtained in a location in Vancouver (Canada) for 2013 from real-world measurements of residential natural gas consumption (Makonin et al., 2016) and from the residential/commercial gridded temporal profiles reported by CAMS-GLOB-TEMPO. Two versions of the CAMS-GLOB-TEMPO temporal factors are shown as a function of the base temperature ($T_b$) assumed when applying the Heating Degree Day approach (18 ºC in orange and 15.5ºC in blue). **(b)** scatter plot showing the trend lines and $R^2$ values obtained from the comparison between measurement-based and CAMS-GLOB-TEMPO daily factors.

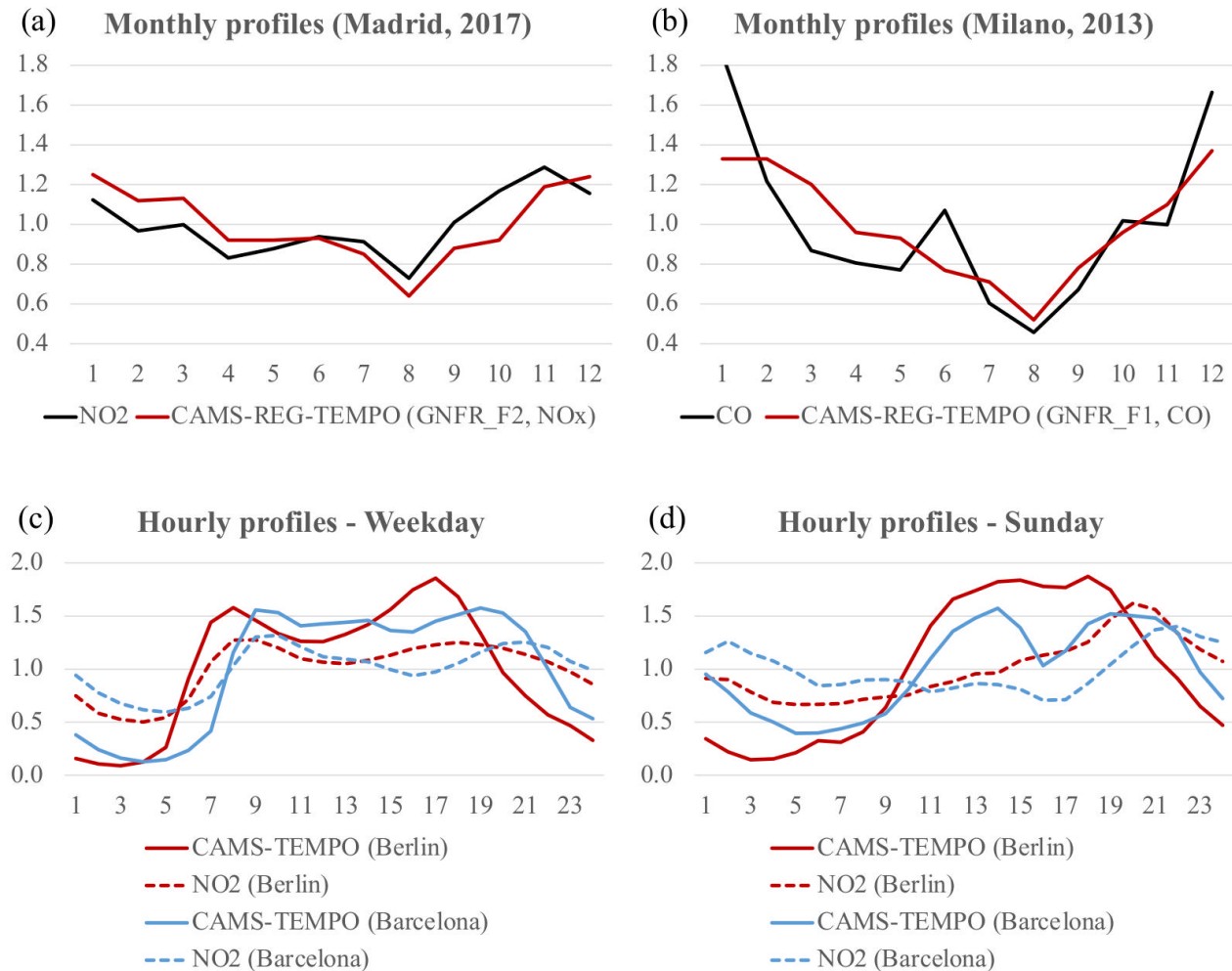

**Figure 11: Comparison between monthly variations of CAMS-REG-TEMPO profiles for NO$_x$ diesel exhaust emissions (GNFR_F2) and measured NO$_2$ concentrations for the city of Madrid during 2017 (a). Comparison between monthly variations of CAMS-REG-TEMPO profile for CO gasoline exhaust emissions (GNFR_F1) and measured CO concentrations for the city of Milano during 2013 (b). Hourly variation of CAMS-TEMPO GNFR_F profiles and NO$_2$ concentrations during weekdays (c) and Sundays (d) for the cities of Barcelona and Berlin.**


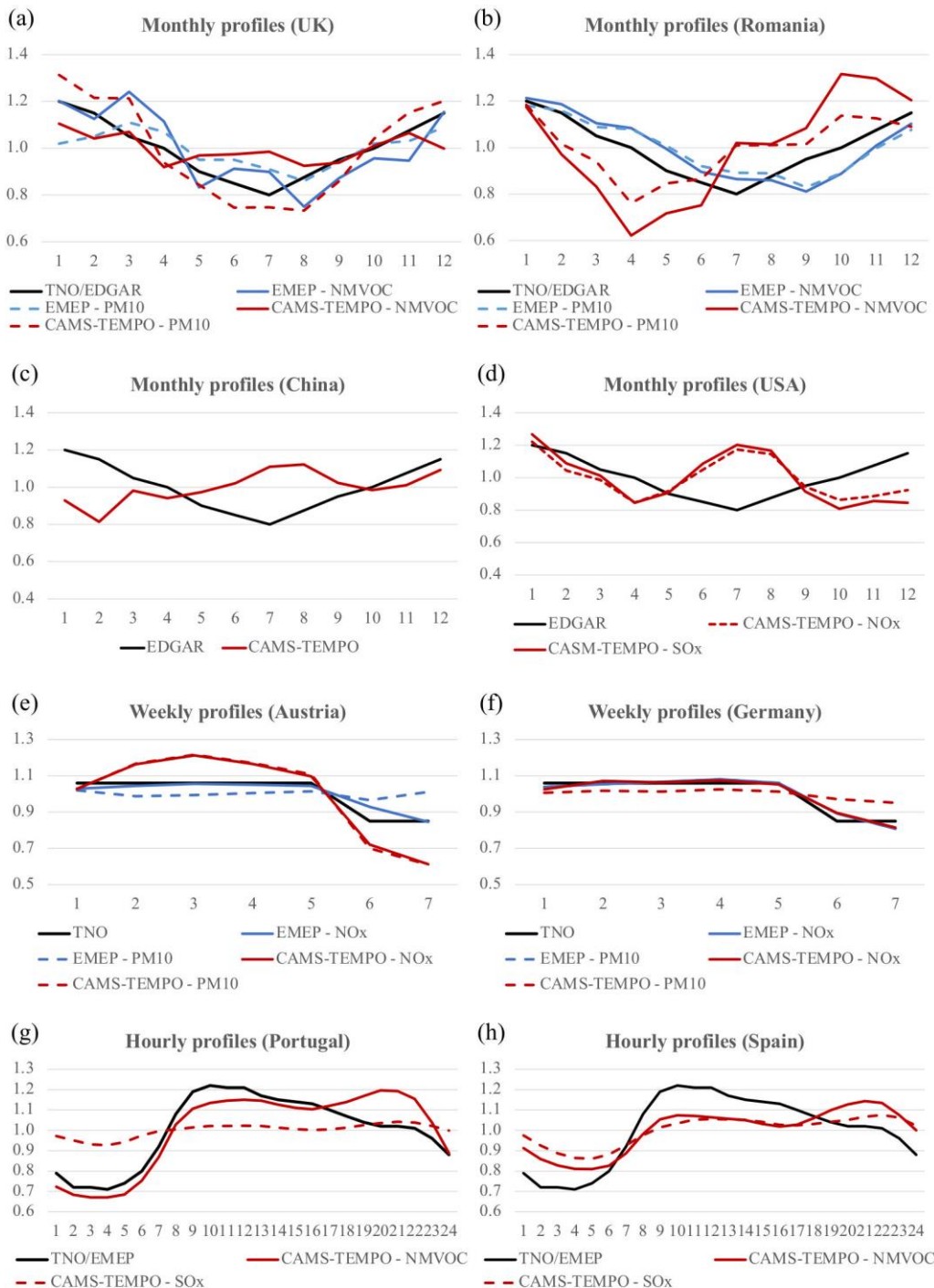

**Figure 12: Comparison of monthly (a, b), weekly (c, d) and hourly (e, f) profiles for energy industry emissions developed in the present work (CAMS-TEMPO) with profiles from EDGAR, EMEP and TNO for selected countries.**

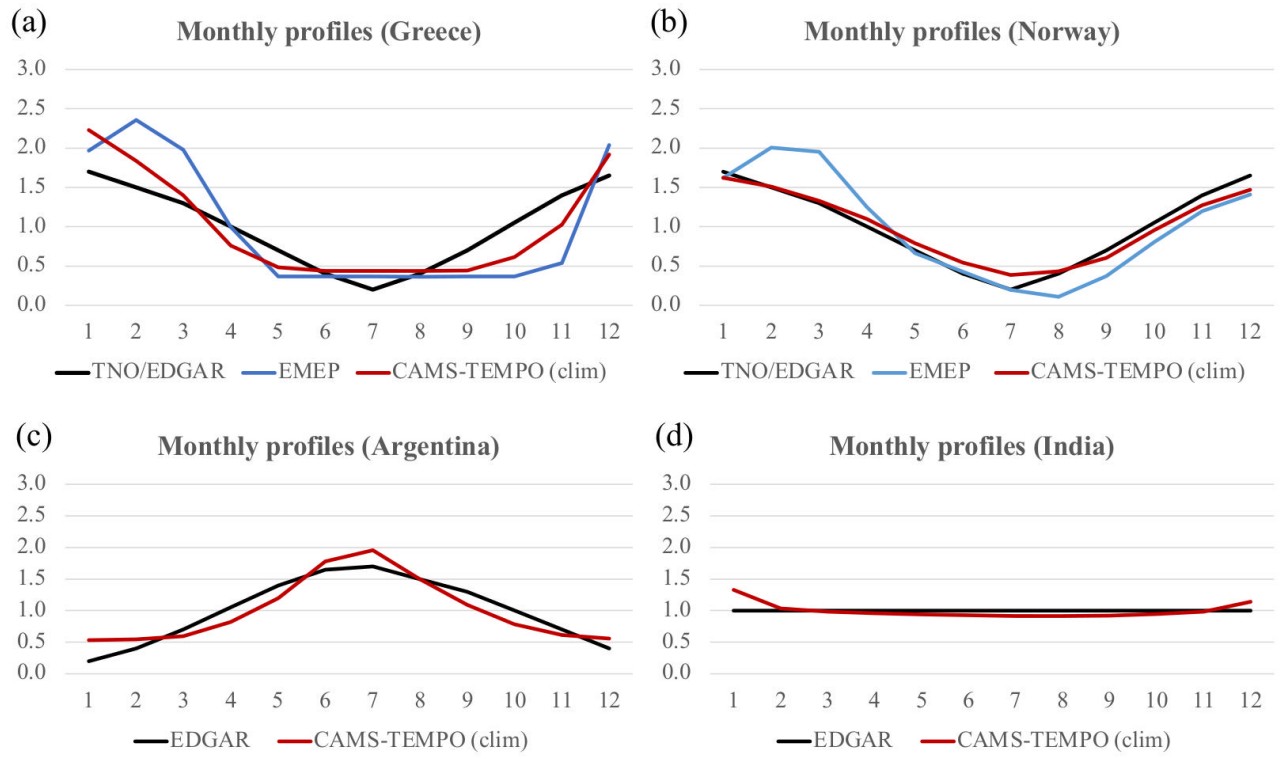

**Figure 13: Comparison of monthly profiles for residential/commercial combustion emissions developed in the present work (CAMS-TEMPO) with profiles from EDGAR, EMEP and TNO for selected countries. The CAMS-TEMPO profiles represent the climatological weight factors (clim) based on the average of each month over all the available years (2010-2017).**

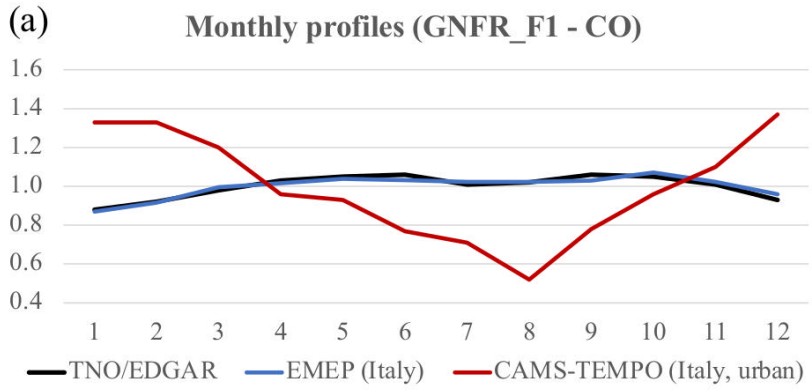

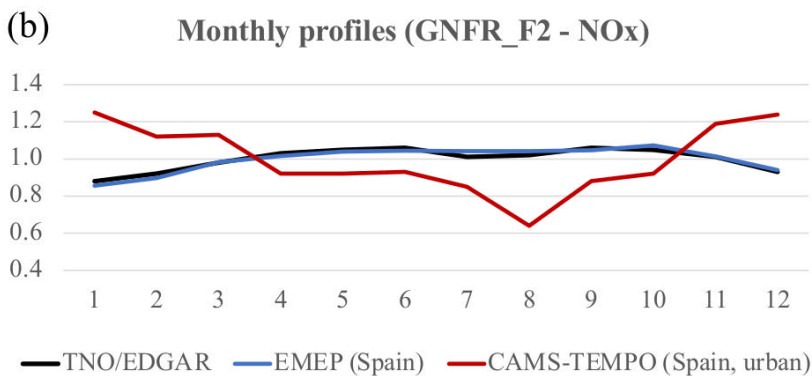

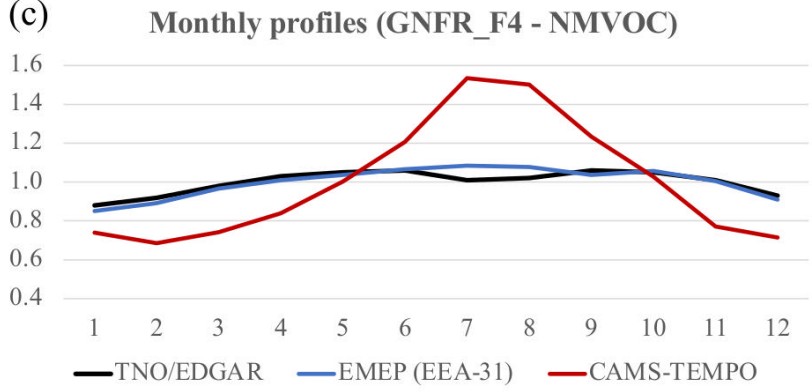

**Figure 14 Comparison of monthly profiles for road transport emissions developed in the present work (CAMS-TEMPO) with profiles from EDGAR, EMEP and TNO for selected countries and categories: CO gasoline exhaust (GNFR_F1, a), NOx diesel exhaust (GNFR_F2, b) and NMVOC non-exhaust (GNFR_F4, c).**


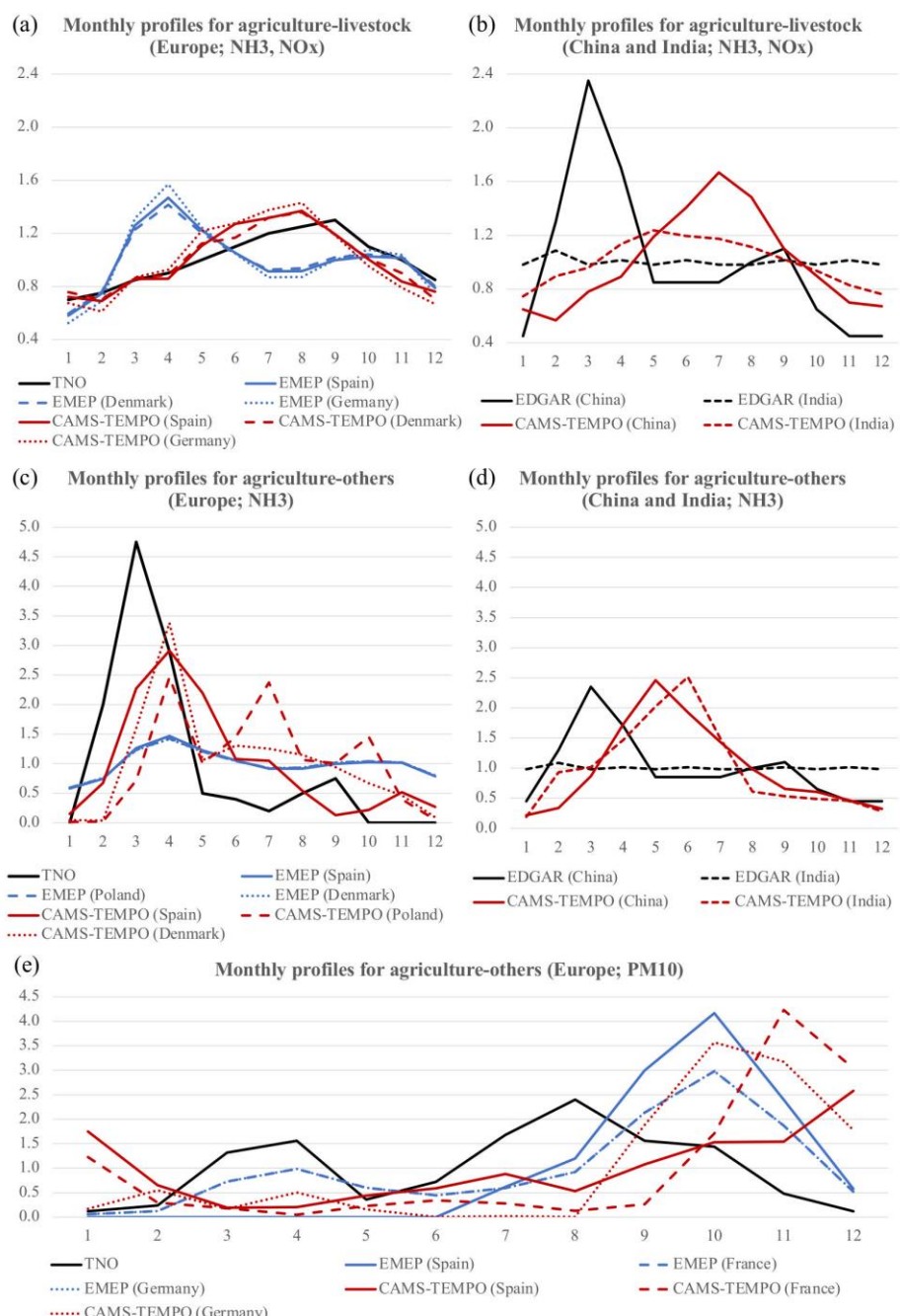

**Figure 15: Comparison of monthly profiles for agricultural emissions developed in the present work (CAMS-TEMPO) with profiles from EDGAR, EMEP and TNO for selected countries and categories: NH3 livestock emissions in Europe (a) and China and India (b), NH3 other agricultural emissions in Europe (c) and China and India (d), PM10 other agricultural emissions in Europe (e).**