# Peer review of "CAMS-TEMPO: global and European emission temporal profile maps for atmospheric chemistry modelling"

_Earth System Science Data, 2020_

## Referee Comment (RC1) · Anonymous Referee #1 · 24 Aug 2020

M. Guevara and co-authors present a manuscript that discusses the development of a new CAMS-TEMPO dataset, which provides temporal profiles for seven atmospheric pollutants and two greenhouse gases for use in 3D atmospheric chemical transport models. Depending on the sector and compound, the dataset provides compound-specific profiles with monthly, weekly, daily, and hourly temporal resolution and reports profiles as a function of emission sector. Two final data products are publicly available, which provide gridded data at 0.1x0.1 degree spatial resolution at the global scale and 0.1x0.05 degrees for European countries, and are consistent with sectoral definitions of emission inventories developed under the CAMS program.

[Figure]

This manuscript describes much-needed updates to emission temporal profiles and does a fairly thorough jobs of documenting the development process. The majority of the manuscript is devoted to the methods section, while the results and discussion section provides select comparisons with available profiles for select sectors, cities, and time periods. The manuscript describes a large amount of excellent work and is generally well-written. However, my first main comment is that the authors should spend additional time in the introduction clarifying exactly how these profiles are an improvement upon past profiles. It is clear that the development of this dataset is well thought-out and has taken a monumental effort, the authors just need to further distinguish the novel aspect of their profiles.

To aid in this clarification, the authors could include a summary table in the introduction or in the supplement that explicitly states which sectors, species, and temporal profiles were developed in this work. While this information is described in the text, a summary table in the main text or supplement would be helpful for modelers seeking to incorporate these updated profiles in their work. While Tables 1 and 2 do provide most of this information, it would be helpful to extend these tables to also include explicit information about the profiles available for each compound in each sector and their spatial resolution. As the CAMS-TEMPO data are provided on a grid, readers may initially think that each of the profiles vary with the same spatial resolution, however, many sectoral profiles are only available at the country level.

My second main comment is that this manuscript would be improved if the authors provided additional discussions (where relevant) about the applicability of their assumptions in regions with limited data, such as Asia, Africa, and South America, which may have different profiles than European countries. This manuscript does a great job of discussing and presenting the regional European data used to derive the CAMS-REG-TEMPO dataset (plus the European countries in the global dataset) but is limited in its discussion of other regions. Similarly, Section 3.1 only provides an evaluation of the CAMS-TEMPO profiles by comparing the data to independent observations in Europe.
If sufficient data does not exist to provide a robust comparison for other regions, this needs to be clearly stated.

Along these lines, the Editorial on ESSD goals, practices, and recommendations by D. Carlson and T. Oda state that authors must describe and document the uncertainties in their datasets. While uncertainties are very difficult to quantify for datasets such as CAMS-TEMPO, the authors should include an additional section where they explicitly discuss the sources of uncertainties in their temporal profiles and how these uncertainties may impact their dataset. Some of this information is already provided in a discussion of limitations on lines 820 – 833, which should be moved to a separate section.

Other relatively minor comments are below and are meant to help improve the clarity of the manuscript. Most comments are requesting additional discussion about the assumptions made in the derivation of profiles for select sectors. For the road sector, the authors generally do the most thorough job of explaining the applicability and impact of their assumptions to regions outside of Europe.

Specific Comments: Introduction – The authors provide a thorough description of widely used global and regional temporal profiles that are currently available. The paragraph starting on line 71 then goes on to discuss some of the limitations in current profiles, including that the same profiles are applied across multiple countries and sectors. The remainder of the introduction indicates that the CAMS-TEMPO dataset is an improvement upon these past datasets as it is derived with more updated input datasets and includes unique profiles across countries, sectors, pollutants, and year, depending on the given profile. However, this is where a summary table would be helpful to highlight exactly which profiles (e.g., which resolution, sectors, and species) are improved over past work and what the basis of the improvements are (e.g., meteorologically -dependent, finer spatial resolution, etc). Since some of the CAMS-TEMPO profiles are taken from the TNO dataset and some are derived by applying the data from the most spatially similar country, it is unclear in the introduction exactly what the

improvements are for each sector and compound compared to previous profiles, particularly since the EDGAR and GAINS profiles are relatively-recently released. This could be clarified with a table in the introduction that compares the years, temporal resolution, and spatial resolution of previous datasets compared to this one.

In addition, it would be helpful to add a few sentences at the end of the introduction about the structure of the manuscript. State that the first few sections describe the development of the sector-specific profiles and that the remaining sections compare the profiles to currently available datasets.

Section 2 – Methodology General comments - Since the availability of monthly, weekly, daily, and hourly profiles, as well as the degree of spatial information is different for each sector and compound, it would be helpful if the authors clarified early on in each section why only certain profiles were derived and why certain sector and compound profiles are spatially invariant (similar to line 324). For example, are the limited profiles due to a data limitation issue (as for the industry sector) or due to the characteristics of the emission source?

It would also be helpful to briefly describe the data used to develop the TNO profiles so that the readers have a sense for how contemporary the TNO profiles are.

Could the authors also comment on the applicability of fuel-weighted profiles that are spatially aggregated to the national-level? For instance, the use of coal in power plants in the US is not evenly distributed geographically across the country. What impact does the spatial variation of fuel use have on the derived temporal profiles (for the energy sector specifically)?

Line 120 – If aviation is discussed in the methods, it should be included in the lists of sectors in the abstract and introduction.

Line 161 – The IEA data is available for a large range of both OECD and non-OECD countries. Could the authors comment on their decision to use TNO profiles for some

countries that have available IEA data?

Line 188 – It is unclear why the fuel-specific monthly factors for countries with IEA data could not be derived. IEA data is provided for each of the fuel types described above from the ENTSO-E dataset.

Line 192 – Provide a reference link to the EPA's recommendations.

Line 199-201– It is not clear what 'methodology' this line is precisely referring to. Are the authors referring to Eq. (1)? (Same comment for line 208)

Line 258 – 267 – While the authors provide sufficient evidence for the choice of f = 0.2 for European countries, they also note that there are large differences in the relative use of biofuel for home heating vs. cooking in developing regions. The authors note that an investigation of the choice of different values of f and Td will be presented in future work. As global emissions from the residential sectors are largest throughout Asian countries, such as India and China (Hoesly et al., (GMD) 2014), the authors should provide examples of monthly and daily profiles under a range of f and Td values and discuss how the current assumptions based on European statistics may impact the derived profiles in other regions that produce a large share of the global total residential emissions.

Line 301 – typo – assign

Line 302 – did the authors mean to include CH4 in the list with NOx and SOx?

Line 340 – 347 – This paragraph discusses some of the assumptions of low inter-annual variability and relatively constant monthly profiles across different industrial sub-sectors. These assumptions are supported by evidence in European countries in Figures S3 and S4, however, there is no discussion on the applicability of these assumption to other large emission regions such as the U.S. and China. Select monthly profiles from non-European countries should be discussed here as well.

Section 2.5 – General comments In Table 2, is appears that unique profiles are reported

for different vehicle fuel types in the CAMS-TEMPO regional dataset. However, there does not seem to be a discussion in this section on differences in profiles from different fuel types. Can the authors clarify whether the CAMS-REG-TEMPO profiles are the same for the GNFR_F1, GNFR_2, GNFR_F3, and GNFR_F4 sectors?

Paragraph on line 382 – Could the authors clarify whether a vehicle type-weighting was applied to derive the temporal profiles for regions with vehicle-type emissions? How were the temporal profiles derived in these cases?

Line 451 – typo, missing period

Figure 4 – The authors should include the flat weekly profile for evaporative NMVOCs to highlight that the profile is flat. As is, it looks as if weekly data for those emissions is unavailable.

Lines 519 – 525 – This text is a repeat of the text from line 409 – 415 and could be replaced by a reference to the previous section. It should also be noted whether the profiles for China were derived from the Guangzhou study and the rationale behind applying factors from Germany to other countries throughout Asia and Africa.

Line 625 – remove one of the instances of 'hourly' in this sentence.

Line 773 – The tables in the appendix should be referenced throughout the main text as they are helpful for the reader.
* * *

---

## Referee Comment (RC2) · Anonymous Referee #2 · 19 Sep 2020

This manuscript covers one of most important key aspects in emissions modeling for air quality and climate modeling communities. While climate modeling community does not need the emission input in hourly level, air quality modeling community does requires temporally highly resolved hourly emission inputs which play a key role in its modeling performance. This manuscript descries all the updates and enhancements made to most latest temporal profiles available and does a fairly great jobs on documenting those processes. However, sometime it lacks on describing the details on how those latst region-specific activities/emissions get used to develop the updated temporal profiles. As an example, it does not describe well on how does the U.S. EPA's NEI emissions were used to develop the county-level temporal monthly, weekly,,

profiles while there are so much of variation between the source sectors and regional temporal variations. I am aware of that it will be impossible to cover the details but at least it can describes higher level of statistical methods and assumptions to meet its requirements. My second comment is that I understand that it has to cover quite range of information and data to describe its detail methodologies. However, right after reading this manuscript into this, I realized that I do not have enough based knowledge of CAMS-related emissions inventory development. This manuscript expects some level of technical understanding prior to what this manuscript can offer. I felt sometime that this manuscript was written as a project technical report originally and then converted to this manuscript. I hope this manuscript can cover some base knowledge for readers who are not so familiar with CAMS_GLOB_ANT and CMAS_REG_AP/GHG inventory development, and many other studies used in this manuscripts. Other than these two comments, there are a few minor comments listed below: Line 63: Full description of TNO Line 67: Full description of EPA Line 84: Add any reference(s) for gridded temporal data development statement. Line 107: "all the domain" to "entire domain" Line 107: " or gridded values" to "temporal-meteorology" (?) Line: 109: "yearly" to "year" (?) Line: 122: "power and heat plants and refineries" to "power/heat plants, and refineries" Line 124: Full description of GNFR_A category Line 146: "includes, among other, processed„" to "includes, ???? high resolution temporal profiles by source category???? , processed„" Line 151: "„hydro, geothermal/other) and country" to "„hydro, and geothermal/other) and country" Line 188: Please explain the details on what is MBS. . . . . .

Line 195: Fig.S1.. Question: How did you come up with US energy sector weekly by fuel type? How did you overcome the regional differences between the CEMS locations? What were the assumptions made for this approach? . . . . . . . . . . . . Overall, I would like for authors to go over the manuscripts more thoroughly and describes underline knowledge, and generalize many technical information based on the readers with minimum knowledge stand point.

---

## Referee Comment (RC3) · Anonymous Referee #3 · 20 Sep 2020

This paper presents and describes the CAMS-TEMPO, a dataset of global and European emission temporal profiles providing gridded monthly, daily, weekly and hourly weight factors for atmospheric chemistry modelling. It represents a very important dataset for the air quality modelling community covering both regional and global applications. The building of such kind of datasets including different activity sectors and pollutants requires a huge and very detailed research and data treatment, which is extremely well reported in this manuscript. Another relevant aspect of this work is the fact that the produced datasets are freely available to be used by modellers.

The manuscript is well structured, each section is adequately divided into subsections

which facilitates the reading of this kind of detailed and long paper. Notwithstanding, it is advisable to end the introduction with a description of the paper structure, which would help to get a first idea of what it presents and would help a reader that is for example just interested on the methods or on the details of a couple of activity sectors.

The methods section starts by presenting and discussing tables 1 and 2 that are in the fact a summary of the product described in the paper; however, this is not clear as one starts reading it. An introductory paragraph before line 101 explaining how the methods section was planned and structures would help.

Since the produced temporal profiles have a global coverage, it would be useful to refer hoe the authors dealt with time zones, assuming that the profiles consider UTC time, which is not referred either.

The paper is very well written, but I would suggest a careful reading in the revision process to correct typos, missing spaces when referring values (e.g. 100 m instead of 100m), missing punctuation, . . .

In terms of Figures, they are well presented and readable, but I would suggest to use the method a), b), etc. . . instead of top, left, right. . . in all the figures.

---

## Author Comment (AC1) · 9 Nov 2020

We would like to thank the reviewers for their positive and constructive feedback, which helped improving the quality of the paper. The reviewers have pointed out issues that required further improvements or explanations. Below we address each specific issue and the manuscript has been updated accordingly.

Please also note the supplement to this comment:
https://essd.copernicus.org/preprints/essd-2020-175/essd-2020-175-AC1-supplement.pdf

[Figure]

[Figure]

**Supplement:**

We would like to thank the reviewers for their positive and constructive feedback, which helped improving the quality of the paper. The reviewers have pointed out issues that required further improvements or explanations. Below we address each specific issue and the manuscript has been updated accordingly.

**Anonymous Referee #1**

M. Guevara and co-authors present a manuscript that discusses the development of a new CAMS-TEMPO dataset, which provides temporal profiles for seven atmospheric pollutants and two greenhouse gases for use in 3D atmospheric chemical transport models. Depending on the sector and compound, the dataset provides compound specific profiles with monthly, weekly, daily, and hourly temporal resolution and reports profiles as a function of emission sector. Two final data products are publicly available, which provide gridded data at 0.1x0.1 degree spatial resolution at the global scale and 0.1x0.05 degrees for European countries, and are consistent with sectoral definitions of emission inventories developed under the CAMS program.

This manuscript describes much-needed updates to emission temporal profiles and does a fairly thorough jobs of documenting the development process. The majority of the manuscript is devoted to the methods section, while the results and discussion section provides select comparisons with available profiles for select sectors, cities, and time periods. The manuscript describes a large amount of excellent work and is generally well-written.

However, my first main comment is that the authors should spend additional time in the introduction clarifying exactly how these profiles are an improvement upon past profiles. It is clear that the development of this dataset is well thought-out and has taken a monumental effort, the authors just need to further distinguish the novel aspect of their profiles.

We agree with the reviewer. The following two sentences has been added in the introduction section to clearly state the novels aspects of the CAMS-TEMPO profiles:

"*The development of CAMPS-TEMPO comes from the need of overcoming the aforementioned limitations of current profiles (i.e. use of outdated source of information and neglection of the temporal variation of emissions across species and countries/regions (⋯)"* lines 93 to 95 of the revised manuscript)

"*The CAMS-TEMPO profiles introduce multiple novel aspects when compared to the current profiles used for air quality modelling, including: (i) pollutant-dependency, (ii) spatial variability and (iii) meteorological influence."* lines 103 to 105 of the revised manuscript)

Moreover, a new table (table 1 of the revised manuscript) has been introduced to compare the main characteristics of the temporal profiles developed in this work with the ones reported in previous datasets (detailed further in a comment below).

To aid in this clarification, the authors could include a summary table in the introduction or in the supplement that explicitly states which sectors, species, and temporal profiles were developed in this work. While this information is described in the text, a summary table in the main text or supplement would be helpful for modelers seeking to incorporate these updated profiles in their work. While Tables 1 and 2 do provide most of this information, it would be helpful to extend these tables to also include explicit information about the profiles available for each compound in each sector and their spatial resolution. As the CAMS-TEMPO data are provided on a grid, readers may initially think that each of the profiles vary with the same spatial resolution, however, many sectoral profiles are only available at the country level.

We agree with the reviewer. Tables 1 and 2 have been updated to include explicit information about the profiles available for each compound in each sector and their spatial resolution and year dependency. For each profile, text between brackets has been added to give information on the

spatial resolution and pollutant and yearly dependency of each profile. The following information is provided; Gridded: indicates that the profile varies per grid cell within a country; per country: indicates that the profile varies only per country; fixed: indicates that the profile is spatially invariant, year-independent: indicates that the profiles does not vary per year; year-dependent: indicates that the profile varies per year; pollutant-independent: indicates that the same profile is proposed for all pollutants (NOx, SOx, NMVOC, NH3, CO, PM10, PM2.5, CO2 and CH4); pollutant-dependent: indicates that the profile varies per pollutant; per day type: indicates that the profile varies as a function of the day (weekday, Saturday and Sunday).

My second main comment is that this manuscript would be improved if the authors provided additional discussions (where relevant) about the applicability of their assumptions in regions with limited data, such as Asia, Africa, and South America, which may have different profiles than European countries. This manuscript does a great job of discussing and presenting the regional European data used to derive the CAMS-REG-TEMPO dataset (plus the European countries in the global dataset) but is limited in its discussion of other regions.

We agree with the reviewer that the discussions of the results in Asia, Africa, and South America is limited, specifically for the energy, manufacturing and road transport sectors. This is due to the lack of sufficient data to provide robust profiles and comparisons. We have clearly stated this fact in the conclusions section of the revised manuscript:

"*Another important shortcoming of the current CAMS-TEMPO dataset is related to the scarcity of available information in developing regions (i.e. Africa, South America and Asia) to construct temporal profiles for the energy industry, manufacturing industry and road transport sectors.*" (lines 1086 to 1088 of the revised manuscript)

In parallel, we have introduced additional discussions on the limitations of the gap-filling methods implemented in CAMS-TEMPO to report temporal profiles in those regions, and how these could be improved in future versions of the dataset:

"*In the current version of the CAMS-TEMPO dataset, a simple gap-filling method has been implemented, which consist of mainly using the TNO European-based profiles when no national or local datasets are available. The rationale behind this choice is that the TNO profiles have been largely used and tested over the last decade in multiple international modelling exercises such as the Air Quality Modelling Evaluation International Initiative (AQMEII; Pouliot et al., 2015). However, TNO profiles are mostly based on Western European data and therefore the degree of representativeness for other world regions is a source of uncertainty. To address this constraint, the current gap-filling procedure will be reviewed in the future by constructing world region profiles for countries with geographical, climatological and/or behavioral similarities, following the approach presented by Crippa et al. (2020).*" (lines 1088 to 1095 of the revised manuscript)

In the particular case of residential combustion, we have added a discussion on how the selection of different Tb and f factors can affect the computation of daily temporal profiles in China and India, two of the largest contributors to total emissions from this sector (further details in a comment below).

Crippa, M., Solazzo, E., Huang, G., Guizzardi, D., Koffi, E., Muntean, M., Schieberle, C., Friedrich, R., and Janssens-Maenhout, G.: High resolution temporal profiles in the Emissions Database for Global Atmospheric Research. Scientific Data 7, 121, doi:10.1038/s41597-020-0462-2, 2020.

Pouliot, G., Denier van der Gon, H. A. D., Kuenen, J., Zhang, J., Moran, M. D., and Makar, P. A.: Analysis of the emission inventories and model-ready emission datasets of Europe and North America for phase 2 of the AQMEII project, J. Atmos. Environ., 115, 345–360, https://doi.org/10.1016/j.atmosenv.2014.10.061, 2015.

Similarly, Section 3.1 only provides an evaluation of the CAMS-TEMPO profiles by comparing the data to independent observations in Europe. If sufficient data does not exist to provide a robust comparison for other regions, this needs to be clearly stated.

We agree with the reviewer. The following sentence was added at the beginning of the section:

"*We compared the CAMS-TEMPO profiles to independent observational datasets. The comparison is mainly performed at the European level as sufficient data could not be gathered to provide a robust comparison for other regions.*" (lines 900 to 901 of the revised manuscript)

Along these lines, the Editorial on ESSD goals, practices, and recommendations by D. Carlson and T. Oda state that authors must describe and document the uncertainties in their datasets. While uncertainties are very difficult to quantify for datasets such as CAMS-TEMPO, the authors should include an additional section where they explicitly discuss the sources of uncertainties in their temporal profiles and how these uncertainties may impact their dataset. Some of this information is already provided in a discussion of limitations on lines 820 − 833, which should be moved to a separate section.

We agree with the reviewer. A new section (5.1 Limitations of the dataset) has been added to the manuscript. The discussion of limitations has been expanded compared to its previous version, taking into account specific issues highlighted by the reviewers.

Other relatively minor comments are below and are meant to help improve the clarity of the manuscript. Most comments are requesting additional discussion about the assumptions made in the derivation of profiles for select sectors. For the road sector, the authors generally do the most thorough job of explaining the applicability and impact of their assumptions to regions outside of Europe. Specific Comments:

Introduction – The authors provide a thorough description of widely used global and regional temporal profiles that are currently available. The paragraph starting on line 71 then goes on to discuss some of the limitations in current profiles, including that the same profiles are applied across multiple countries and sectors. The remainder of the introduction indicates that the CAMS-TEMPO dataset is an improvement upon these past datasets as it is derived with more updated input datasets and includes unique profiles across countries, sectors, pollutants, and year, depending on the given profile. However, this is where a summary table would be helpful to highlight exactly which profiles (e.g., which resolution, sectors, and species) are improved over past work and what the basis of the improvements are (e.g., meteorologically-dependent, finer spatial resolution, etc). Since some of the CAMS-TEMPO profiles are taken from the TNO dataset and some are derived by applying the data from the most spatially similar country, it is unclear in the introduction exactly what the improvements are for each sector and compound compared to previous profiles, particularly since the EDGAR and GAINS profiles are relatively-recently released. This could be clarified with a table in the introduction that compares the years, temporal resolution, and spatial resolution of previous datasets compared to this one.

A new table has been introduced to compare the main characteristics of the temporal profiles developed in this work with the ones reported in other datasets (TNO, Denier van det Gon et al. (2011); EMEP, Simpson et al. (2012); EDGARv4.3.2, Janssens-Maenhout et al. (2019) and EDGARv5, Crippa et al., (2020)) regarding spatial coverage, temporal and spatial resolution and yearly/meteorology dependency.

**Table 1: Main characteristics of the temporal profiles developed in this work compared to those reported in other datasets including TNO (Denier van det Gon et al., 2011), EMEP (Simpson et al., 2012), EDGARv4.3.2 (Janssens-Maenhout et al., 2019) and EDGARv5 (Crippa et al., 2020) regarding spatial coverage, temporal and spatial resolution and yearly/meteorology dependency.**

| Sector | Dataset | Spatial coverage | Temporal resolution | Spatial resolution | Yearly/Meteorology dependency |
|---|---|---|---|---|---|

| | | | | | |
|---|---|---|---|---|---|
| **energy industry** | this work | Global, EU | monthly, weekly, hourly | Country-dependent | no |
| | TNO | EU | monthly, weekly, hourly | Fixed | no |
| | EMEP | EU | monthly, weekly, hourly | Country-dependent | no |
| | EDGARv4.3.2 | Global | monthly | 3 geo-regions | no |
| | EDGARv5 | Global | monthly, weekly, hourly | Country-dependent | yes (monthly profiles) |
| **residential combustion** | this work | Global, EU | monthly, weekly, daily, hourly | Grid cell level (monthly, daily profiles) | yes (monthly, daily profiles) |
| | TNO | EU | monthly, weekly, hourly | Fixed | no |
| | EMEP | EU | monthly, weekly, hourly | Country-dependent | no |
| | EDGARv4.3.2 | Global | monthly | 3 geo-regions | no |
| | EDGARv5 | Global | monthly, weekly, hourly | Country-dependent | yes (monthly profiles) |
| **manufacturing industry** | this work | Global, EU | monthly, weekly, hourly | Country-dependent (monthly profiles) | no |
| | TNO | EU | monthly, weekly, hourly | Fixed | no |
| | EMEP | EU | monthly, weekly, hourly | Country-dependent | no |
| | EDGARv4.3.2 | Global | monthly | Fixed | no |
| | EDGARv5 | Global | monthly, weekly, hourly | 23 world regions | no |
| **road transport** | this work | Global, EU | monthly, weekly, hourly | Grid cell level | yes (EU monthly profiles) |
| | TNO | EU | monthly, weekly, hourly | Fixed | no |
| | EMEP | EU | monthly, weekly, hourly | Country-dependent | no |
| | EDGARv4.3.2 | Global | monthly | Fixed | no |
| | EDGARv5 | Global | monthly, weekly, hourly | 23 world regions | no |
| **aviation** | this work | EU | monthly, weekly, hourly | Country-dependent (monthly profiles) | no |
| | TNO | EU | monthly, weekly, hourly | Fixed | no |
| | EMEP | EU | monthly, weekly, hourly | Country-dependent | no |
| | EDGARv4.3.2 | Global | monthly | Fixed | no |
| | EDGARv5 | Global | monthly, weekly, hourly | Fixed | no |
| **agriculture (livestock)** | this work | Global, EU | monthly, weekly, daily, hourly | Grid cell level (monthly, daily profiles) | yes (monthly, daily profiles) |
| | TNO | EU | monthly, weekly, hourly | Fixed | no |
| | EMEP | EU | monthly, weekly, hourly | Country-dependent | no |
| | EDGARv4.3.2 | Global | monthly | 3 geo-regions | no |
| | EDGARv5 | Global | monthly, weekly, hourly | Fixed | no |
| **agriculture (fertilizers)** | this work | Global, EU | monthly, weekly, daily, hourly | Grid cell level (monthly, daily profiles) | yes (monthly, daily profiles) |
| | TNO | EU | monthly, weekly, hourly | Fixed | no |
| | EMEP | EU | monthly, weekly, hourly | Country-dependent | no |
| | EDGARv4.3.2 | Global | monthly | 3 geo-regions | no |
| | EDGARv5 | Global | monthly, weekly, hourly | Fixed | no |
| | this work | Global, EU | monthly, weekly, hourly | Grid cell level (monthly profiles) | no |

| agriculture (agricultural waste burning) | TNO | EU | monthly, weekly, hourly | Fixed | no |
| | EMEP | EU | monthly, weekly, hourly | Country-dependent | no |
| | EDGARv4.3.2 | Global | monthly | 3 geo-regions | no |
| | EDGARv5 | Global | monthly, weekly, hourly | 23 world regions | no |

The table has been introduced in the introduction section of the manuscript as follows:

"*Table 1 summarizes and compares the main characteristics of the CAMS-TEMPO profiles with the ones reported in other datasets including TNO (Denier van det Gon et al., 2011), EMEP (Simpson et al., 2012), EDGARv4.3.2 (Janssens-Maenhout et al., 2019) and EDGARv5 (Crippa et al., 2020) regarding spatial coverage, temporal and spatial resolution and yearly/meteorology dependency.*" lines 105 to 108 of the revised manuscript)

Moreover, and as mentioned in a previous comment, Tables 1 and 2 have been updated to include explicit information about the CAMS-TEMPO profiles available for each compound in each sector and their spatial resolution and yearly dependency.

In addition, it would be helpful to add a few sentences at the end of the introduction about the structure of the manuscript. State that the first few sections describe the development of the sector-specific profiles and that the remaining sections compare the profiles to currently available datasets

We agree with the reviewer. The following paragraph has been added at the end of the introduction:

"*The paper is organized as follows. Section 2 describes, for each sector, the approaches and sources of information used to develop the CAMS-TEMPO profiles. Section 3 discusses the obtained temporal profiles and compares them to currently available datasets. Sections 4 provides a description of the data availability and finally Sect. 5 presents the main conclusions of this work.*" (lines 141 to 144 of the revised manuscript)

Section 2 – Methodology General comments - Since the availability of monthly, weekly, daily, and hourly profiles, as well as the degree of spatial information is different for each sector and compound, it would be helpful if the authors clarified early on in each section why only certain profiles were derived and why certain sector and compound profiles are spatially invariant (similar to line 324). For example, are the limited profiles due to a data limitation issue (as for the industry sector) or due to the characteristics of the emission source?

We have clarified in the beginning of each section the profiles that were constructed and why certain sector and compound profiles are spatially invariant. For instance, in the case of agriculture, we added the following sentence:

"*For both sectors, monthly and daily region-dependent profiles were constructed considering specific meteorological-parametrizations (Sect. 2.7.1 to 2.7.3). For the hourly profiles, only fixed weight factors are proposed due to a data limitation issue (Sect. 2.7.4).*" (lines 771 to 773 of the revised manuscript)

It would also be helpful to briefly describe the data used to develop the TNO profiles so that the readers have a sense for how contemporary the TNO profiles are.

The data used to develop the TNO profiles is already described in the introduction section:

"*TNO and EDGAR share the same monthly profiles for residential combustion and road transport (Friedrich and Reis, 2004), as well as for the energy industry (Veldt, 1992) and agriculture (Asman, 1992).*" (lines 79 to 80 of the revised manuscript)

We have added information on the weekly and hourly profiles for road transport:

"*Similarly, weekly and hourly factors reported by TNO are based on long time series of Dutch data registering the traffic intensity between 1985 and 1998.*" (lines 86 to 87 of the revised manuscript)

Extra information is also included in the new Table 1 of the manuscript.

Could the authors also comment on the applicability of fuel-weighted profiles that are spatially aggregated to the national-level? For instance, the use of coal in power plants in the US is not evenly distributed geographically across the country. What impact does the spatial variation of fuel use have on the derived temporal profiles (for the energy sector specifically)?

As mentioned in the manuscript, the specificity of CAMS-TEMPO profiles depends upon the degree of sectoral disaggregation used to report the original CAMS inventories. For the energy sector, temporal profiles per fuel type could not be considered as all the emissions are reported under a unique sector. Having said that, we agree with the reviewer that considering national-level fuel-weighted profiles can lead to a certain level of uncertainty as power plants are usually not evenly distributed geographically across the countries. With our current approach, we assume that e.g. all SO$_x$ emissions emitted by power plants are following the same temporal pattern, regardless of the type of fuel used by each facility. In order to quantify the potential uncertainty behind the current approach, we compared the CAMS-TEMPO monthly profile proposed for the US SOx energy industry emissions against the individual monthly profiles obtained for the top 50 US emitting power plants, which represent approximately 40% of total emissions. The facility-level monthly profiles were computed considering the CEMS data. The results of the comparison are shown in the figure below. It is observed that the majority of the facilities follow a quite homogeneous temporal pattern, which is in line with the seasonality proposed by CAMS-TEMPO.

[Figure]

We highlighted the limitation of using national aggregated and fuel-weighted profiles on the conclusions section of the revised manuscript:

*"Similarly, fuel-weighted profiles that are spatially aggregated to the national-level were had to be considered for the energy industry sector, as original emissions are not split by fuel type."* (lines 1064 to 1066 of the revised manuscript)

Line 120 – If aviation is discussed in the methods, it should be included in the lists of sectors in the abstract and introduction

Already included in the abstract, i.e. "transport (road traffic and air traffic in airports)"

Added in the introduction as follows:

*"energy industry, residential combustion, manufacturing industry, transport (road traffic and air traffic in airports) and agriculture"* (lines 101 to 102 of the revised manuscript)

Line 161 – The IEA data is available for a large range of both OECD and non-OECD countries. Could the authors comment on their decision to use TNO profiles for some countries that have available IEA data?

At the time of using the IEA electricity data, the information was only reported for OECD countries. As stated in the new IEA website, "since April 2020 this report also includes electricity production data for a selection of IEA Association Countries and other economies". We take note of this update and we will consider it in futures updates of the CAMS-TEMPO dataset.

Line 188 – It is unclear why the fuel-specific monthly factors for countries with IEA data could not be derived. IEA data is provided for each of the fuel types described above from the ENTSO-E dataset.

Similar to the previous point, at the time of using the IEA electricity data, the information was provided by generation type (i.e. combustible fuels, nuclear, hydro, geothermal/other) but not by fossil fuel type (all fossil fuels are reported under the category "combustible fuels"). The new IEA database uses a "new detailed breakdown" in which, compared to the so-called "old breakdown", the electricity data is provided by fuel type (coal, oil, natural gas, renewable and other combustibles). The comparisons between data provided using the "new detailed breakdown" and "old breakdown" can be found in this link: https://www.iea.org/reports/monthly-electricity-statistics. We take note of this update and we will consider it in futures updates of the CAMS-TEMPO dataset.

Line 192 – Provide a reference link to the EPA's recommendations.

The following reference is provided:

Stella, G.: Development of Hourly Inventories Utilizing CEM-Based Data. 14th emission inventory conference of the US Environmental Protection Agency. Available at: https://citeseerx.ist.psu.edu/viewdoc/download?doi=10.1.1.543.5410&rep=rep1&type=pdf (last accessed, October 2020), 2005.

Line 199-201– It is not clear what 'methodology' this line is precisely referring to. Are the authors referring to Eq. (1)? (Same comment for line 208)

Yes, we refer to Eq. (1), we have clarified this point in the manuscript.

Line 258 – 267 – While the authors provide sufficient evidence for the choice of f = 0.2 for European countries, they also note that there are large differences in the relative use of biofuel for home heating vs. cooking in developing regions. The authors note that an investigation of the choice of different values of f and Td will be presented in future work. As global emissions from the residential sectors are largest throughout Asian countries, such as India and China (Hoesly et al., (GMD) 2014), the authors should provide examples of monthly and daily profiles under a range of f and Td values and discuss how the current assumptions based on European statistics may impact the derived profiles in other regions that produce a large share of the global total residential emissions.

We agree with the reviewer. To illustrate how current assumptions may impact the derived profiles in non-European regions, we computed daily factors for the residential sector over India and China for 2015 using a range of f (0, 0.2 and 0.5) and Tb (15.5, 18 °C) values. A new supplementary figure has been added showing the results.

[Figure]

**Figure S2: Daily factors for residential/commercial combustion emissions computed over China and India for 2015 using the heating degree day approach with a range of threshold temperature (Tb: 15.5, 18 ºC) and non-heating fraction (f: 0, 0.2 and 0.5) values.**

The results are briefly discussed in the revised version of the manuscript:

*"To illustrate how current assumptions may impact the derived profiles in non-European regions, we computed daily factors for the residential sector over India and China for 2015 using a range of f (0, 0.2 and 0.5) and Tb (15.5, 18 °C) values (Fig. S2). We selected these two countries as they produce a large share of the global residential emissions (Hoesly et al., 2018). Differences between the daily factors of up to 55% were found during winter time when comparing the results computed with f = 0.0 and 0.5. The investigation and proposal of different f values (as well as different Tb values) for different regions of the world will be addressed in future works."* (lines 423 to 428 of the revised manuscript)

Line 301 – typo – assign

Corrected

Line 302 – did the authors mean to include CH4 in the list with NOx and SOx?

Yes, $CH_4$ has been added in the list with NOx and SOx

Line 340 – 347 – This paragraph discusses some of the assumptions of low interannual variability and relatively constant monthly profiles across different industrial sub-sectors. These assumptions are supported by evidence in European countries in Figures S3 and S4, however, there is no discussion on the applicability of these assumption to other large emission regions such as the U.S. and China. Select monthly profiles from non-European countries should be discussed here as well.

We agree with the reviewer. The results plotted for Spain in Figure S4 have been replaced by the ones obtained for the US (Board, 2020). Moreover, the result reported for Thailand in Pham et al. (2008) are also mentioned in the manuscript to include an Asia representative, as we did not find specific information for China.

Board: Board of Governors of the Federal Reserve System. Industrial Production: Manufacturing (SIC) [IPMANSICS], retrieved from FRED, Federal Reserve Bank of St. Louis. Available at: https://fred.stlouisfed.org/series/IPMANSICS, (last accessed, October 2020) 2020.

Section 2.5 – General comments In Table 2, is appears that unique profiles are reported for different vehicle fuel types in the CAMS-TEMPO regional dataset. However, there does not seem to be a discussion in this section on differences in profiles from different fuel types. Can the authors clarify whether the CAMS-REG-TEMPO profiles are the same for the GNFR_F1, GNFR_2, GNFR_F3, and GNFR_F4 sectors?

CAMS-REG-TEMPO profiles are the same for the GNFR_F1, GNFR_2, GNFR_F3, and GNFR_F4 sectors except for those pollutant whose temporal emission variability is associated to changes in ambient temperature: CO and NMVOC GNFR_F1, NOx GNFR_F2 and NMVOC GNFR_F4. This has been clarified in the revised version of the manuscript and also in the revised version of the Table (now Table 3).

Paragraph on line 382 – Could the authors clarify whether a vehicle type-weighting was applied to derive the temporal profiles for regions with vehicle-type emissions? How were the temporal profiles derived in these cases?

The dataset of California (McDonald et al., 2014) is the only case where the reported profiles are provided by vehicle-type and not for total vehicles (for Germany, the other country mentioned on line 382, we have both total and vehicle-specific traffic count information).

What we did for the California case (profiles are used for the US) is to consider the original temporal profiles reported for LDV, as this type of vehicle dominates the temporal distribution of total traffic flow and also presents the larger contribution to total emissions. No vehicle type-weighting was applied, as we consider that region-specific weights should be considered (i.e. in urban areas the amount of circulating HDV and associated emissions is generally low, whereas in the interurban areas it is larger) and we do not have this information. As mentioned in the manuscript, the fact that CAMS-GLOB_ANT and CAMS-REG_AP/GHG report LDV and HDV-related emissions under the same pollutant sector limits the possibility of working with vehicle-type profiles.

We clarified in the manuscript that for California we assumed the LDV profiles:

"When only vehicle-type temporal profiles were available (i.e. California), the information reported for LDV is used, as this type of vehicle dominates the temporal distribution of total traffic flow." (lines 574 to 575 of the revised manuscript)

Line 451 – typo, missing period

Changed to "for the period 2010 – 2017"

Figure 4 – The authors should include the flat weekly profile for evaporative NMVOCs to highlight that the profile is flat. As is, it looks as if weekly data for those emissions is unavailable.

The flat weekly profile for evaporative NMVOCs has been included in the revised version of Figure 4.

Lines 519 – 525 – This text is a repeat of the text from line 409 – 415 and could be replaced by a reference to the previous section. It should also be noted whether the profiles for China were derived from the Guangzhou study and the rationale behind applying factors from Germany to other countries throughout Asia and Africa.

The text has been replaced by a reference to the previous section, as proposed by the reviewer. It has been also noted that the profiles for China were derived from the average weight factors reported for Guangzhou and Beijing. The urban/rural profiles developed for Germany were applied to other countries through Asia, Africa and Latin America as they were based on the largest number of traffic count stations (more than 1500). We understand this approach can be improved, but was constrained in this study by the traffic count data availability. We have added a point in the discussion section

about the need of reviewing the current data gap-filling methods implemented in CAMS-TEMPO to report temporal profiles in those countries with little or no information available.

Line 625 – remove one of the instances of 'hourly' in this sentence.

Removed

Line 773 – The tables in the appendix should be referenced throughout the main text as they are helpful for the reader.

We agree with the reviewer. Tables in appendix A have been referenced in the 2.1, 2.4, 2.5, 2.6 and 2.7 sections.

**Anonymous Referee #2**

This manuscript covers one of most important key aspects in emissions modeling for air quality and climate modeling communities. While climate modeling community does not need the emission input in hourly level, air quality modeling community does requires temporally highly resolved hourly emission inputs which play a key role in its modeling performance. This manuscript descries all the updates and enhancements made to most latest temporal profiles available and does a fairly great jobs on documenting those processes.

However, sometime it lacks on describing the details on how those latst region-specific activities/emissions get used to develop the updated temporal profiles. As an example, it does not describe well on how does the U.S. EPA's NEI emissions were used to develop the county-level temporal monthly, weekly profiles while there are so much of variation between the source sectors and regional temporal variations. I am aware of that it will be impossible to cover the details but at least it can describe higher level of statistical methods and assumptions to meet its requirements.

*We have added more details on how the pollutant-dependent temporal profiles for the energy sector in the US were derived using as a basis the hourly measured emissions and heat input reported by the EPA's CEMS data.*

*"For the US, $NO_x$ and $SO_x$ monthly profiles were derived from the corresponding hourly measured emissions reported by the EPA's CEMS data. Measurements from all individual plants were averaged at the monthly level and then normalised to sum 12."* (lines 331 to 335 of the revised manuscript)

*"For the US, the CEMS data was used to compute pollutant-dependent profiles following the same procedure as described in Sect. 2.2.1. Measurements from all individual plants were averaged per day of the week and then normalised to sum 7"* (lines 344 to 345 of the revised manuscript)

*We have also clarified how we used the multiple bottom-up agricultural emission inventories to develop gridded monthly profiles for fertilizer-related emissions.*

*"From the MASAGE_NH3 and Zhang et al. (2018) inventories, we considered the original monthly NH3 emissions reported under the use of fertilizer emission categories. From the NEI inventory, we used as a basis the original hourly emissions reported under the species "NH3_FERT", which includes the amount of NH3 from fertilizer sources, and aggregated them to the monthly level. For all cases, the monthly and gridded emissions were first normalised to sum 12 and then remapped onto the CAMS-GLOB_ANT and CAMS-REG_AP grids applying a nearest neighbour approach."* (lines 809 to 814 of the revised manuscript)

My second comment is that I understand that it has to cover quite range of information and data to describe its detail methodologies. However, right after reading this manuscript into this, I realized that I do not have enough based knowledge of CAMS-related emissions inventory development. This manuscript expects some level of technical understanding prior to what this manuscript can offer. I felt sometime that this manuscript was written as a project technical report originally and then converted to this manuscript. I hope this manuscript can cover some base knowledge for readers who are not so familiar with CAMS_GLOB_ANT and CMAS_REG_AP/GHG inventory development, and many other studies used in this manuscript.

*We agree with the reviewer. A brief description of the CAMS_GLOB_ANT and CMAS_REG_AP/GHG inventories and associated references have been added in the introduction section of the manuscript:*

[revised manuscript text omitted]

Other than these two comments, there are a few minor comments listed below:

Line 63: Full description of TNO

Added: "Netherlands Organisation for Applied Scientific Research (TNO)"

Line 67: Full description of EPA

Added: "Environmental Protection Agency (EPA)"

Line 84: Add any reference(s) for gridded temporal data development statement.

Added

Line 107: "all the domain" to "entire domain"

Changed

Line 107: " or gridded values" to "temporal-meteorology" (?)

We decided to keep the formulation "gridded values" as here what we want to stress is the difference between the profiles that we consider to be spatial invariant (fixed) and the ones that were reported as gridded because they vary either by country or region (not only because of meteorology, but also other factors such as socio-economic aspects)

Line: 109: "yearly" to "year"(?)

Changed

Line: 122: "power and heat plants and refineries" to "power/heat plants, and refineries"

Changed

Line 124: Full description of GNFR_A category

Added

Line 146: "includes, among other, processed,,," to "includes, ???? high resolution temporal profiles by source category????, processed,,,"

The US EPA emission modelling platform includes multiple datasets related to emissions: set of emissions inventories, other data files such as temporal and speciation profiles, software tools, and scripts that process the emissions into the form needed for air quality modelling. Here we just want to mention the information that we used, which is the CEMS data to derive temporal profiles for the energy sector. We removed the "among other" to avoid confusions.

Line 151: ",,hydro, geothermal/other) and country" to ",,hydro, and geothermal/other) and country"

Changed

Line 188: Please explain the details on what is MBS

The following sentence has been added:

"The Monthly Bulletin of Statistics (MBS) is a database of the United Nations with focus on national economic and social statistics"

Line 195: Fig.S1. Question: How did you come up with US energy sector weekly by fuel type? How did you overcome the regional differences between the CEMS locations? What were the assumptions made for this approach?

There was a mistake in the manuscript. US weekly profiles were not derived from hourly electricity production data as originally stated, but from the hourly measured emissions and heat input reported by the EPA's CEMS data. CEMS NOx and SOx hourly measurements from all individual plants were averaged at the weekly level and then normalised to sum 7. For the other pollutants, the measured heat input was used as a proxy to derive the corresponding weekly profiles. In the current approach, we are not considering regional differences between CEMS locations (we are just using national-level averaged temporal profiles). In future works we could check in detail how the profiles from individual plants differ between them and if needed propose regional-dependent profiles for the US. The following sentence has been added to the manuscript:

*"For the US, the CEMS data was used to compute pollutant-dependent profiles following the same procedure as described in Sect. 2.2.1. Measurements from all individual plants were averaged per day of the week and then normalised to sum 7."* (lines 357 to 358 of the revised manuscript)

Overall, I would like for authors to go over the manuscripts more thoroughly and describes underline knowledge, and generalize many technical information based on the readers with minimum knowledge stand point.

The following actions have been taken:

- Added a detailed description of CAMS-REG and CAMS-GLOB emission inventories
- Added a description on how the CEMS data was used to derive pollutant-dependent profiles for the energy sector in the US
- Added a description on how the MASAGE_NH3 and NEI inventories were used to derive monthly profiles for the fertilizer agriculture sector.
- Included a comparison table between the profiles developed in this work and those reported in previous studies.
- Included more details on the characteristics of the CAMS-TEMPO profiles in Tables 2 and 3 of the revised manuscript

**Anonymous Referee #3**

This paper presents and describes the CAMS-TEMPO, a dataset of global and European emission temporal profiles providing gridded monthly, daily, weekly and hourly weight factors for atmospheric chemistry modelling. It represents a very important dataset for the air quality modelling community covering both regional and global applications. The building of such kind of datasets including different activity sectors and pollutants requires a huge and very detailed research and data treatment, which is extremely well reported in this manuscript. Another relevant aspect of this work is the fact that the produced datasets are freely available to be used by modellers. The manuscript is well structured, each section is adequately divided into subsections, which facilitates the reading of this kind of detailed and long paper.

Notwithstanding, it is advisable to end the introduction with a description of the paper structure, which would help to get a first idea of what it presents and would help a reader that is for example just interested on the methods or on the details of a couple of activity sectors.

We agree with the reviewer. The following paragraph has been added at the end of the introduction:

*"The paper is organized as follows. Section 2 describes, for each sector, the approaches and sources of information used to develop the CAMS-TEMPO profiles. Section 3 discusses the obtained temporal profiles and compares them to currently available datasets. Sections 4 provides a description of the data availability and finally Sect. 5 presents the main conclusions of this work."* (lines 141 to 144 of the revised manuscript)

The methods section starts by presenting and discussing tables 1 and 2 that are in the fact a summary of the product described in the paper; however, this is not clear as one starts reading it. An introductory paragraph before line 101 explaining how the methods section was planned and structures would help.

We have rearranged the introductory text of this section so that it is cleared how the methods section is organised.

Since the produced temporal profiles have a global coverage, it would be useful to refer hoe the authors dealt with time zones, assuming that the profiles consider UTC time, which is not referred either.

All the hourly temporal profiles in CAMS-TEMPO are provided in local standard time (LST). The conversion from LST to coordinated universal time (UTC) as a function of the different time zones is a process that needs to be performed by the final user. Time zone adjustments is a process typically performed by the emission processing systems/tools designed to adapt emission data to the air quality modelling requirements. Two examples of processing systems that perform the conversion from LST to UTC are SMOKE (https://www.cmascenter.org/smoke/documentation/2.7/html/ch02s10s04.html) and HERMESv3 (Guevara et al. 2019).

Guevara, M., Tena, C., Porquet, M., Jorba, O., and Pérez García-Pando, C.: HERMESv3, a stand-alone multi-scale atmospheric emission modelling framework – Part 1: global and regional module, Geosci. Model Dev., 12, 1885–1907, https://doi.org/10.5194/gmd-12-1885-2019, 2019.

We have clarified this point in the revised version of the manuscript as follows:

*"Note that the hourly temporal profiles in CAMS-TEMPO are provided in local standard time (LST). The conversion from LST to coordinated universal time (UTC) as a function of time zones is a process that needs to be performed by the final user. Time zone adjustments is a process typically performed by the emission processing systems/tools designed to adapt emission data to the air quality modelling requirements (e.g. Guevara et al., 2019)."* (lines 174 to 178 of the revised manuscript)

The paper is very well written, but I would suggest a careful reading in the revision process to correct typos, missing spaces when referring values (e.g. 100 m instead of 100m), missing punctuation, : : :

We have revised the manuscript and correct typos and missing spaces when referring values

In terms of Figures, they are well presented and readable, but I would suggest to use the method a), b), etc: : : instead of top, left, right: : : in all the figures

We have updated Figures 2 to 15 with the method a), b),… as suggested by the reviewer. We have also updated the corresponding captions of the figures and associated references in the text.

---

## Author Response (AR2)

The only remaining (very small) suggestion I have is for line 311-312. While the authors have conducted an additional sensitivity test, the results of this test indicate that daily factor can have a high level of sensitivity to the Tb and f parameters. Even though the results of the sensitivity test are now included in the main text, I would still suggest stating these implications in the main text. i.e., Change to "Differences between the daily factors of up to 55% were found during winter time when comparing the results computed with f = 0.0 and 0.5, indicating that daily factors are sensitive to these parameters."

We have modified line 311-312 following the referee's suggestion:

"*Differences between the daily factors of up to 55% were found during winter time when comparing the results computed with f = 0.0 and 0.5, indicating that daily factors are sensitive to these parameters*" (lines 311 to 313 of the revised manuscript)